# Benign Overfitting in Out-of-Distribution Generalization of Linear Models

**Shange Tang**[*†]      **Jiayun Wu**[*‡]      **Jianqing Fan**[†]      **Chi Jin**[§]

## Abstract

Benign overfitting refers to the phenomenon where an over-parameterized model fits the training data perfectly, including noise in the data, but still generalizes well to the unseen test data. While prior work provides some theoretical understanding of this phenomenon under the in-distribution setup, modern machine learning often operates in a more challenging Out-of-Distribution (OOD) regime, where the target (test) distribution can be rather different from the source (training) distribution. In this work, we take an initial step towards understanding benign overfitting in the OOD regime by focusing on the basic setup of over-parameterized linear models under covariate shift. We provide non-asymptotic guarantees proving that benign overfitting occurs in standard ridge regression, even under the OOD regime when the target covariance satisfies certain structural conditions. We identify several vital quantities relating to source and target covariance, which govern the performance of OOD generalization. Our result is sharp, which provably recovers prior in-distribution benign overfitting guarantee (Tsigler & Bartlett, 2023), as well as under-parameterized OOD guarantee (Ge et al., 2024) when specializing to each setup. Moreover, we also present theoretical results for a more general family of target covariance matrix, where standard ridge regression only achieves a slow statistical rate of $\mathcal{O}(1/\sqrt{n})$ for the excess risk, while Principal Component Regression (PCR) is guaranteed to achieve the fast rate $\mathcal{O}(1/n)$, where $n$ is the number of samples.

## 1 Introduction

In modern machine learning, distribution shift has become a ubiquitous challenge where models trained on a source data distribution are tested on a different target distribution (Zou et al., 2018; Hendrycks & Dietterich, 2019; Guan & Liu, 2021; Koh et al., 2021). Generalization under distribution shift, known as Out-of-Distribution (OOD) generalization, remains a fundamental issue in the practical application of machine learning (Recht et al., 2019; Hendrycks et al., 2021; Miller et al., 2021; Wenzel et al., 2022). While there has been extensive work on the theoretical understanding of OOD generalization, most of it has focused on under-parameterized models (Shimodaira, 2000; Lei et al., 2021; Ge et al., 2024; Zhang et al., 2022). However, over-parameterized models, such as deep neural networks and large language models (LLMs) in the fine-tuning stage, which have more parameters than training samples, are widely used in modern machine learning. Surprisingly, despite the classic bias-variance tradeoff for under-parameterized models, over-parameterized models tend to overfit the data while still achieving strong in-distribution generalization, a phenomenon known as benign overfitting (Hastie et al., 2022; Shamir, 2023) or harmless interpolation (Muthukumar et al., 2020). Therefore, it is crucial to theoretically understand how benign overfitting shapes OOD generalization in over-parameterized models.

It is established in over-parameterized models that "benign overfitting" occurs when the data essentially resides on a low-dimensional manifold. The manifold assumption (Belkin & Niyogi, 2003) is widely applicable across image, speech, and language data, where although features are embedded

---

[*]They contributed equally and are listed in alphabetical order.

[†]Department of Operations Research and Financial Engineering, Princeton University; {shangetang, jqfan}@princeton.edu

[‡]Department of Computer Science and Technology, Tsinghua University; jiayun.wu.work@gmail.com

[§]Department of Electrical and Computer Engineering, Princeton University; chij@princeton.edu

in a high-dimensional ambient space, their generation is governed by a few degrees of freedom imposed by physical constraints (Niyogi, 2013). Specifically, the covariance matrix of the data should be characterized by several major directions corresponding to large eigenvalues, while the remaining directions are high-dimensional but associated with small eigenvalues. In this setting, even though the estimator may overfit the noise in the low-variance directions, it can still capture the signal along the major directions while the noise is dampened in the minor directions. Recent non-asymptotic analyses have provided upper bounds on the excess risk for the minimum-norm interpolant and over-parameterized ridge estimator under this framework (Bartlett et al., 2020; Hastie et al., 2022; Tsigler & Bartlett, 2023).

However, a theoretical understanding of OOD generalization in over-parameterized models remains elusive. In this paper, we take an initial step towards characterizing OOD generalization in over-parameterized models under *general* covariate shift, a standard assumption in the OOD regime (Ben-David et al., 2006), where the conditional distribution of the outcome given the covariates remains invariant. We derive the first vanishing, non-asymptotic excess risk bound for ridge regression and minimum-norm interpolation, assuming that the source covariance is dominated by a few major eigenvalues, which satisfies the in-distribution benign overfitting condition. Notably, we allow the target covariance to be arbitrary. This result stands in contrast to recent work that either addresses only a restrictive type of target covariance matrices (Hao et al., 2024; Mallinar et al., 2024) or provides excess risk bounds that are non-vanishing (Tripuraneni et al., 2021a; Hao et al., 2024).

In summary, our excess risk bound identifies several key quantities that relate to the source and target covariance. We show that "benign overfitting" occurs in case 1 where the target distribution data lies on the low-dimensional manifold of the source distribution, so that these key quantities are well controlled. In the opposite case 2, where the target distribution data falls outside of the low-dimensional manifold of the source, ridge regression may incur large excess risk, lower bounded by the slow statistical rate of $\mathcal{O}(1/\sqrt{n})$. In contrast, we show that Principal Component Regression (PCR) achieves the fast rate of $\mathcal{O}(1/n)$ in case 2. The non-asymptotic rates of both ridge regression and PCR are validated through simulation experiments on multivariate Gaussian data in Appendix A.

**Our contributions.**

1. We provide a sharp, instance-dependent excess risk bound for over-parameterized ridge regression under covariate shift (Theorem 2). Our result applies to any target distribution, requiring only that the source covariance be dominated by a few major eigenvectors and that the minor components are high-dimensional. Our results shows that when certain key quantities relating to the source and target distributions are bounded (case 1), ridge regression exhibits "benign over-fitting", achieving excess risk comparable to the in-distribution case. Importantly, this condition requires that the *overall magnitude* of the target covariance along the minor directions scales similarly to, or smaller than, that of the source, but it does not depend on the spectral structure of the target covariance on the minor directions. Our results recover the in-distribution bound from Tsigler & Bartlett (2023) when the source and target match and also recover the sharp bound from Ge et al. (2024) for under-parameterized linear regression under covariate shift when the minor components vanish.

2. For case 2 where the "benign-overfitting" of ridge regression is not guaranteed (when target distribution exhibits significant variance in the minor directions), we further show that ridge regression incurs a higher error rate compared to the in-distribution case, lower bounded by the slow statistical rate of $\mathcal{O}(1/\sqrt{n})$ in certain instances (Theorem 4). However, we demonstrate that Principal Component Regression ensures a fast rate of $\mathcal{O}(1/n)$ in these cases, provided that the true signal primarily lies in the major directions of the source (Theorem 5). Additionally, PCR does not rely on the minor directions of the source distribution being high-dimensional, highlighting its advantage over ridge regression in such settings.

## 1.1 RELATED WORK

**Over-parameterization.** The success of over-parameterized models in machine learning has sparked significant research on their theoretical foundations. Harmless interpolation (Muthukumar et al., 2020) or benign overfitting (Shamir, 2023) describes cases where linear models interpolate noise yet still generalize well. Double descent in prediction error is also observed as the ambient dimension surpasses the number of training samples (Nakkiran, 2019; Xu & Hsu, 2019).

Research in this field can be divided into two categories based on assumptions about the spectral structure of the sample covariance. The first category assumes an almost isotropic sample covariance matrix with a bounded condition number or an isotropic prior distribution of parameters (Belkin et al., 2020), allowing for asymptotic risk bounds (Dobriban & Wager, 2018; Richards et al., 2021). However, ridgeless regression is sub-optimal in this setting unless the signal-to-noise ratio is infinite (Wu & Xu, 2020), and non-asymptotic error bounds are lacking. Our work falls into the second category, focusing on the covariance model where a small number of eigenvalues dominate the sample covariance, and the signal is concentrated in the subspace spanned by the leading eigenvectors (Bibas et al., 2019; Chinot & Lerasle, 2022; Hastie et al., 2022; Silin & Fan, 2022). Linear regression can be optimal without regularization under this covariance structure (Kobak et al., 2020), which is of practical interest because ridgeless regression is equivalent as gradient descent from zero initialization (Zhou et al., 2020). Sharp non-asymptotic bounds for variance and bias in ridge and ridgeless regression have been derived (Bartlett et al., 2020; Tsigler & Bartlett, 2023).

Extending the analysis of ridgeless estimators (i.e., minimum norm interpolants), uniform convergence bounds for generalization error have been studied for all interpolants with arbitrary norms. However, uniformly bounding the difference between population and empirical errors generally fails to ensure a consistent predictor (Zhou et al., 2020), necessitating strong assumptions on distributions (Koehler et al., 2021) or hypothesis classes (Negrea et al., 2020). Over-parameterization theory for linear models has also been applied to two-layer neural networks approximated via kernel ridge regression (Liang et al., 2020; Ghorbani et al., 2020; 2021; Bartlett et al., 2021; Mei & Montanari, 2022; Mei et al., 2022; Montanari & Zhong, 2022; Simon et al., 2023), though this lies beyond the scope of the present work.

**Out-of-Distribution generalization.** Out-of-distribution generalization is well studied for under-parameterized models, especially under the setup of covariate shift. Shimodaira (2000) pointed out that vanilla MLE (Empirical Risk Minimization, ERM) is asymptotically optimal among all the weighted likelihood estimators when the model is well-specified. For non-asymptotic results, Cortes et al. (2010); Agapiou et al. (2017), provide risk bounds for importance weighting. Another line of work provides non-asymptotic analyses for covariate shift, focusing on linear regression or a few specific models such as one-hidden layer neural network (Mousavi Kalan et al., 2020; Lei et al., 2021; Zhang et al., 2022). Most recently Ge et al. (2024) provides tight non-asymptotic analysis for general well-specified parametric models, showing that even without target data, vanilla MLE (Empirical Risk Minimization, ERM) is minimax optimal with a sharp $1/n$ excess risk bound based on Fisher information.

Research on over-parameterized models under distribution shift has largely focused on covariate shifts in linear regression. Importance weighting for over-parameterized models (Chen et al., 2024) and general sample reweighting offer no advantage over ERM since both converge to the same estimator via gradient descent (Zhai et al., 2022). Consequently, much literature focuses on minimum-norm interpolation as the natural ERM solution. For isotropic covariance structure, Tripuraneni et al. (2021a) derive an asymptotic generalization bound decreasing with $d/n$ where $d$ represents the data dimension. Most related to our work, a line of work also consider the covariance model dominated by several major eigenvectors, however, they only address a restrictive form of covariate shift or obtaining a non-vanishing bound: Kausik et al. (2024) study a linear model with additive noise on covariates when data strictly lies in a low-dimensional subspace, showing a non-vanishing bound. Hao et al. (2024) give a non-vanishing bound for the case where features are translated by a constant and the covariance matrix is preserved. Mallinar et al. (2024) investigate the special case with independent covariates and simultaneously diagonal source and target covariance matrices, under which the in-distribution analysis of Bartlett et al. (2020); Tsigler & Bartlett (2023) can be directly extended. Still, their estimation bias bound is looser than ours, as it exhibits a gap compared to Tsigler & Bartlett (2023)'s sharp bound even when the source and target distributions are aligned. In contrast, our work achieves the first vanishing non-asymptotic error bound for general covariate shift, assuming only finite second moments for the target covariance matrix.

Another line of research considers non-parametric models under covariate shift (Kpotufe & Martinet, 2018; Hanneke & Kpotufe, 2019; Pathak et al., 2022; Ma et al., 2023), presenting minimax results governed by a transfer-exponent that measures the similarity between source and target distributions. However, this falls outside the scope of our work.

**Principal Component Regression.** Principal Component Regression (PCR) has been designed to treat multicollinearity in high-dimensional linear regression, where the covariates possess a latent, low-dimensional representation (Massy, 1965; Jeffers, 1967; Jolliffe, 1982; Jeffers, 1981). PCR has been widely used in statistics (Liu et al., 2003; Fan et al., 2021; Fan & Gu, 2023), econometrics (Stock & Watson, 2002; Bai & Ng, 2002; Fan et al., 2020), chemometrics (Næs & Martens, 1988; Sun, 1995; Vigneau et al., 1997; Depczynski et al., 2000; Keithley et al., 2009), construction management (Chan & Park, 2005), environmental science (Kumar & Goyal, 2011; Hidalgo et al., 2000), signal processing (Huang & Yang, 2012) and etc.

Regarding the theory of PCR, Hadi & Ling (1998) identify conditions under which PCR will fail. Bair et al. (2006) suggest selecting principal components based on their association with the outcome and provide corresponding asymptotic consistency results. Xu & Hsu (2019) establish asymptotic risk bounds for PCR with varying numbers of selected components $k$. They show that the "double descent" behavior also happens in PCR as $k/d$ increases. Most relevant to our work, Agarwal et al. (2019) derive non-asymptotic error bounds for PCR, and show that the error decays at a rate of $\mathcal{O}(1/\sqrt{n})$ ($n$ is the sample size), assuming all singular values of the data matrix are of similar magnitude. Agarwal et al. (2020) further improves this rate to $\mathcal{O}(1/n)$. However, both results consider a fixed design with strict low-rank assumptions, making them inapplicable to our setting of OOD generalization.

## 2 COVARIATE SHIFT SETUP UNDER OVER-PARAMETERIZATION

### 2.1 DATA WITH COVARIATE SHIFT

We address the out-of-distribution (OOD) generalization of over-parameterized models under covariate shift, where the covariates, denoted by a random vector $x \in \mathbb{R}^d$, follow different distributions during training and evaluation. Specifically, we assume that the training data is sampled from a source distribution $\mathcal{P}_S$, and the learned model is subsequently applied to data from an unknown target distribution $\mathcal{P}_T$. Let the covariates be zero-mean on the source distribution, and define the covariance matrix as $\Sigma_S := \mathbb{E}_{x \sim \mathcal{P}_S}\left[xx^T\right]$. Since we can always choose an orthonormal basis such that $\Sigma_S$ becomes diagonal, we express $\Sigma_S = \mathrm{diag}(\lambda_1, \cdots, \lambda_d)$ without loss of generality, where the eigenvalues are arranged in non-increasing order: $\lambda_1 \geq \cdots \geq \lambda_d \geq 0$. Moreover, we assume sub-gaussianity of the source covariates, i.e., $\Sigma_S^{-1/2}x$ is $\sigma$-sub-gaussian where the precise definition of the sub-Gaussian norm is given in section B. We consider a general covariate distribution for the target, assuming only that it has a finite second moment, denoted by $\Sigma_T := \mathbb{E}_{x \sim \mathcal{P}_T}\left[xx^T\right]$, which is not necessarily diagonal.

We consider a linear response model that remains consistent across the source and target distributions. The outcome follows $y = x^T \beta^* + \epsilon$, where $\beta^* \in \mathbb{R}^d$ represents the true parameter, and $\epsilon$ is an independent noise with zero-mean and variance $v^2$.

### 2.2 LEARNING PROCEDURE AND EVALUATION

The learning procedure involves training a linear model with $n$ i.i.d. samples $\{(x_i, y_i)\}_{i=1}^n$ drawn from the source distribution. Define $X := (x_1, ..., x_n)^T \in \mathbb{R}^{n \times d}$, $Y := (y_1, ..., y_n)^T$ and $\boldsymbol{\epsilon} := (\epsilon_1, ..., \epsilon_n)^T$. We focus on models $\widehat{\beta}(Y)$ that are linear in $Y$, allowing us to write $\widehat{\beta}(Y) = \widehat{\beta}(X\beta^\star) + \widehat{\beta}(\boldsymbol{\epsilon})$. We consider ridge regression and Principal Component Regression to be two examples of such algorithms. With a regularization coefficient $\lambda \geq 0$, the ridge estimator is defined as

$$\widehat{\beta}(Y) = X^T(XX^T + \lambda I_n)^{-1}Y.$$

The estimator is assessed on the target distribution by its excess risk relative to the true model, expressed as the following equation:

$$\mathcal{R}\big(\widehat{\beta}(Y)\big) := \mathbb{E}_{(x,y) \sim \mathcal{P}_T}\left[\left(y - x^T\widehat{\beta}(Y)\right)^2 - \left(y - x^T\beta^*\right)^2\right] = \left\|\widehat{\beta}(Y) - \beta^*\right\|_{\Sigma_T}^2,$$

where we define $\|x\|_A := \sqrt{x^T A x}$ for any positive semi-definite matrix $A$. The metric of interest is the expected excess risk with respect to the noise, given by $\mathbb{E}_{\boldsymbol{\epsilon}}\big[\mathcal{R}\big(\widehat{\beta}(Y)\big)\big]$. Following from the lin-

earity of the model, the expected excess risk can be decomposed into bias and variance components:

$$\mathbb{E}_{\boldsymbol{\epsilon}}\big[\mathcal{R}\big(\widehat{\beta}(Y)\big)\big] = \mathbb{E}_{\boldsymbol{\epsilon}}\big\|\widehat{\beta}(\boldsymbol{\epsilon})\big\|_{\Sigma_T}^2 + \big\|\widehat{\beta}(X\beta^\star) - \beta^\star\big\|_{\Sigma_T}^2,$$

where we define the variance as $V := \mathbb{E}_{\boldsymbol{\epsilon}}\big\|\widehat{\beta}(\boldsymbol{\epsilon})\big\|_{\Sigma_T}^2$ and the bias as $B := \big\|\widehat{\beta}(X\beta^\star) - \beta^\star\big\|_{\Sigma_T}^2$. Here, we assume the shifts only in the distribution of covariates; the regression coefficients remain the same.

## 2.3 THE STRUCTURE OF COVARIANCE IN BENIGN OVERFITTING

Throughout this paper, we follow the convention of Tsigler & Bartlett (2023) and consider the *source* covariance matrix $\Sigma_S$ as characterized by a few numbers of high-variance directions and a large number of low-variance directions of similar magnitude. We refer to the high-variance directions as "major directions" and the low-variance directions as "minor directions". We denote the number of major directions as $k$. For the remaining $d - k$ minor directions, we use the following notions of effective rank to approximate the number of directions with a similar scale. For the ridge regularization coefficient $\lambda \geq 0$, we define:

$$r_k := \frac{\lambda + \sum_{j>k}\lambda_j}{\lambda_{k+1}}, \quad R_k := \frac{\big(\lambda + \sum_{j>k}\lambda_j\big)^2}{\sum_{j>k}\lambda_j^2}.$$

We have $1 \leq r_k \leq R_k$. When $\lambda = 0$, it further holds that $R_k \leq d - k$. We denote the first $k$ columns of $X$ as $X_k$ and the remaining $d - k$ columns as $X_{-k}$. Correspondingly, we partion $\beta^\star$ into $\beta_k^\star$ and $\beta_{-k}^\star$. The covariance matrix blocks along the diagonals are denoted by $\Sigma_{S,k}$, $\Sigma_{S,-k}$, $\Sigma_{T,k}$ and $\Sigma_{T,-k}$. We define the following quantities to facilitate our presentation, which are crucial in our analysis.

$$\mathcal{T} = \Sigma_{S,k}^{-\frac{1}{2}}\Sigma_{T,k}\Sigma_{S,k}^{-\frac{1}{2}}, \quad \mathcal{U} = \Sigma_{S,-k}\Sigma_{T,-k}, \quad \mathcal{V} = \Sigma_{S,-k}^2. \tag{1}$$

## 3 OVER-PARAMETERIZED RIDGE REGRESSION

In the context of in-distribution generalization, where $\Sigma_S = \Sigma_T$, for over-parameterized linear models, Bartlett et al. (2020) and Tsigler & Bartlett (2023) demonstrate that the ridge estimator (with the minimum-norm interpolant as a special case) can effectively learn the signal from the subspace of data spanned by the major eigenvectors, while benignly overfitting the noise in the minor directions under certain scenarios. They argue that benign overfitting occurs when the true signal predominantly lies in the major directions, and the minor directions have a small scale but highly effective rank. This section explores whether this mechanism still holds under covariate shift. We derive upper bounds (Theorem 2) for the excess risk of the over-parameterized ridge estimator in the context of OOD generalization, demonstrating that "benign overfitting" also happens under covariate shift, given that the target distribution's covariance remains dominated by the first $k$ dimensions. Specifically, we show that $\mathcal{T}$ characterizes the shift in the major directions, while the *overall magnitude* of $\Sigma_{T,-k}$, which captures the shift in the minor directions, is crucial for benign overfitting. When the overall magnitude of $\Sigma_{T,-k}$ scales similarly to or smaller than those of the source, ridge regression achieves the same non-asymptotic error rate under covariate shift as in the in-distribution setting. Surprisingly, although a high effective rank in the minor directions of the source is essential for benign overfitting, only the overall magnitude matters for the target distribution.

## 3.1 WARM-UP: IN-DISTRIBUTION BENIGN OVERFITTING

As a warm-up, we introduce Tsigler & Bartlett (2023)'s in-distribution result on benign overfitting in ridge regression. When the data dimension $d$ exceeds the sample size $n$, the ridgeless estimator interpolates the training data, fitting the noise. In this case, the estimator $\widehat{\beta}$ lies within the subspace spanned by the covariates of the $n$ samples. If $d$ is much larger than $n$, a new test point is highly likely to be orthogonal to this subspace, ameliorating noise from affecting the prediction. Therefore, the minor components of the covariance matrix actually provide implicit regularization. Tsigler & Bartlett (2023) assume that the data lies in a space with $k$ major directions and $d - k$ weak

but essentially high-dimensional minor directions, allowing for benign overfitting. This intuition is formalized through an assumption that controls the condition number of the Gram matrix for the remaining $d - k$ dimensions.

**Assumption 1** (CondNum$(k, \delta, L)$ (Tsigler & Bartlett, 2023)). Define a matrix $A_k = \lambda I_n + X_{-k} X_{-k}^T$. With probability at least $1 - \delta$, $A_k$ is positive definite and has a condition number no greater than $L$, i.e.,

$$\frac{\mu_1(A_k)}{\mu_n(A_k)} \leq L,$$

where the $i$-th largest eigenvalue of a matrix is denoted by $\mu_i(\cdot)$.

**Remark 1.** This assumption essentially posits that the minor directions of the source covariance have an effective rank significantly greater than $n$. As evidence, Tsigler & Bartlett (2023) prove that if CondNum holds, the effective rank $r_k$ is lower bounded by $n/L$, up to a constant. Conversely, a lower bound on the effective rank $r_k$ also implies an upper bound of the condition number of $A_k$. For more details, refer to Tsigler & Bartlett (2023, Lemma 3).

Assuming CondNum, Tsigler & Bartlett (2023) obtain sharp upper bounds for both the variance and bias of the ridge estimator, with matching lower bounds (see their Theorem 2). To facilitate the presentation, we define $\widetilde{\lambda} := \lambda + \sum_{j>k} \lambda_j$ to represent the combined regularization term from both ridge and implicit regularization.

**Theorem 1** (Tsigler & Bartlett (2023)). There exists a constant $c$ that only depends on $\sigma, L$, such that for any $n > ck$, if the assumption condNum$(k, \delta, L)$ (Assumption 1) is satisfied, then it holds that $n < cr_k$, and with probability at least $1 - \delta - ce^{-n/c}$,

$$\frac{V}{cv^2} \leq \frac{k}{n} + \frac{n}{R_k}, \qquad \frac{B}{c} \leq B_{\text{ID}} := \|\beta_k^\star\|_{\Sigma_{S,k}^{-1}}^2 \left(\frac{\widetilde{\lambda}}{n}\right)^2 + \|\beta_{-k}^\star\|_{\Sigma_{S,-k}}^2,$$

where $v$ denotes the standard deviation of the noise $\epsilon$.

The first variance term arises from estimating the $k$ major signal dimensions, corresponding to the classic variance in $k$-dimensional ordinary least squares. The second variance term, $n/R_k$, vanishes when the minor directions are sufficiently high-dimensional, i.e., when $R_k \gg n$. However, the signal in the minor directions, $\|\beta_{-k}^\star\|_{\Sigma_{S,-k}}^2$, is nearly lost when projected from the high-dimensional ambient space onto the low-dimensional sample space, contributing to the second bias term. Finally, the first bias term relates to the signal estimation in the first $k$ dimensions and is introduced by the overall regularization from both ridge and implicit regularization imposed by the minor components.

## 3.2 OUT-OF-DISTRIBUTION BENIGN OVERFITTING

We now investigate the out-of-distribution performance of the ridge estimator. Intuitively, when the minor components vanish for both the source and target distributions, over-parameterized ridge regression essentially reduces to under-parameterized ridge regression in the major directions, achieving a rate of $\widetilde{\mathcal{O}}(\text{tr}[\mathcal{T}]/n)$, as demonstrated by Ge et al. (2024). When the minor components do not vanish, a high effective rank of the minor components in the source distribution is essential for "benign overfitting", as shown by Tsigler & Bartlett (2023). However, we argue that for the target distribution, only the *overall magnitude* of the minor components is critical for benign overfitting. This is because when the source's minor directions have an effective rank much larger than $n$, the $n$-dimensional subspace spanned by the training samples is already almost orthogonal to any test point with high probability. As a result, the spectral structure of the target becomes irrelevant–only a small overall magnitude of the target's minor components is required.

We formalize those intuitive claims by deriving upper bounds for both the variance and bias of ridge regression under covariate shift, assuming a source distribution similar to the in-distribution case. Our upper bound is sharp and can be applied to any target distributions, reducing to Tsigler & Bartlett (2023)'s bound (Theorem 1) when the target and source distributions are aligned. Additionally, we recover Ge et al. (2024)'s sharp bound for under-parameterized linear regression under a covariate shift when the high-dimensional minor components vanish.

**Theorem 2.** There exists a constant $c > 2$ depending only on $\sigma, L$, such that for any $cN < n < r_k$, if the assumption condNum$(k, \delta, L)$ (Assumption 1) is satisfied, then with probability at least $1 - 3\delta$,

$$\frac{V}{cv^2} \leq \frac{k}{n} \cdot \frac{\text{tr}[\mathcal{T}]}{k} + \frac{n}{R_k} \cdot \frac{\text{tr}[\mathcal{U}]}{\text{tr}[\mathcal{V}]}.$$

$$\frac{B}{c} \leq B_{\text{ID}} \cdot \left( \|\mathcal{T}\| + \frac{n}{r_k} \frac{\|\Sigma_{T,-k}\|}{\|\Sigma_{S,-k}\|} \right).$$

where $\mathcal{T}, \mathcal{U}, \mathcal{V}$ are defined in Equation (1), $N = \text{Poly}(k + \ln(1/\delta), \lambda_1 \lambda_k^{-1}, 1 + \widetilde{\lambda} \lambda_k^{-1})$, and $\text{Poly}(\cdot)$ denotes a polynomial function.

Recall that $B_{\text{ID}}$ is the bias upper bound from Theorem 1. Theorem 2 establishes an upper bound for the excess risk of ridge regression under general covariate shift, expressed in a multiplicative form based on Theorem 1. This formulation enables a straightforward comparison of the impact of covariate shifts on the bias and variance of ridge estimators relative to the in-distribution case. The first conclusion is that Theorem 2 well reduces to the corresponding result in Theorem 1 when no distribution shift occurs–i.e., $\Sigma_S = \Sigma_T$. This connection follows directly from the condition $n < r_k$.

The second conclusion is that covariate shift in the first $k$ dimensions and last $d - k$ dimensions introduce multiplicative factors of $\frac{\text{tr}[\mathcal{T}]}{k}$, $\|\mathcal{T}\|$ and $\frac{\text{tr}[\mathcal{U}]}{\text{tr}[\mathcal{V}]}$, $nr_k^{-1} \frac{\|\Sigma_{T,-k}\|}{\|\Sigma_{S,-k}\|}$, respectively, on the excess risk. Therefore, as long as these factors are bounded by constants, over-parameterized ridge regression achieves the same non-asymptotic rate of excess risk under covariate shift as the in-distribution setting. This scenario, well addressed by ridge regression, occurs when the target distribution's covariance structure remains dominated by the first $k$ dimensions. In the following, we analyze the impact of the factors introduced by covariate shifts on both the major and minor directions.

1. $\mathcal{T}$ **characterizes the shift in the major directions.** Under covariate shift within the first $k$ dimensions, we obtain the same non-asymptotic error rate as in Theorem 1, only if $\|\mathcal{T}\|$ is bounded by a constant, as $\text{tr}[\mathcal{T}]/k \leq \|\mathcal{T}\|$. The matrix $\mathcal{T}$ plays a central role in Theorem 2 to quantify covariate shift within the first $k$ dimensions, matching our intuition. This echoes with Ge et al. (2024)'s finding that $\text{tr}[\mathcal{T}]$ captures the difficulty of covariate shift for under-parameterized ridgeless regression (MLE). They establish a sharp upper bound on excess risk using Fisher information (see their Theorem 3.1), which simplifies to a rate of $\widetilde{\mathcal{O}}(\text{tr}[\mathcal{T}]/n)$ for linear models. Theorem 2 recovers this result when applied to a $k$-dimensional under-parameterized setting where all high-dimensional minor components vanish, specifically when $\Sigma_{S,-k} = \Sigma_{T,-k} = \mathbf{0}$. Under the same condition as Theorem 2, for a constant $c$ depending only on $\sigma, L$, with high probability, the variance and bias terms are bounded by:

$$\frac{V}{cv^2} \leq \frac{\text{tr}[\mathcal{T}]}{n}, \quad \frac{B}{c} \leq \|\beta_k^\star\|_{\Sigma_{S,k}^{-1}}^2 \left( \frac{\lambda}{n} \right)^2 \|\mathcal{T}\|.$$

The variance bound aligns with Ge et al. (2024)'s result while the bias vanishes as $\lambda \to 0$.

2. **The overall magnitude of $\Sigma_{T,-k}$ is crucial for benign overfitting.** Under covariate shift within the last $d - k$ dimensions, when both $\frac{\text{tr}[\mathcal{U}]}{\text{tr}[\mathcal{V}]}$ and $nr_k^{-1} \frac{\|\Sigma_{T,-k}\|}{\|\Sigma_{S,-k}\|}$ are bounded by constants, we achieve the same non-asymptotic error rate as in Theorem 1. Note that $\frac{\text{tr}[\mathcal{U}]}{\text{tr}[\mathcal{V}]} \leq \frac{\|\Sigma_{T,-k}\|_{\text{F}}}{\|\Sigma_{S,-k}\|_{\text{F}}}$. In other words, matching our intuition, if the *overall magnitude* of the minor components of target covariance scales similarly to or smaller than those of the source, in terms of the covariance norms, "benign overfitting" also happens under covariate shift. Importantly, this condition does not impose constraints on the internal spectral structure of the minor components of the target covariance. For example, we do not force each eigenvalue of $\Sigma_{T,-k}$ to scale with its corresponding eigenvalue of $\Sigma_{S,-k}$ in decreasing order, as assumed in prior work (Mallinar et al., 2024). Surprisingly, for benign overfitting to happen, the source distribution must have a high effective rank in the minor directions. However, for the target distribution, only the overall magnitude of the minor components is relevant.

   Another observation is that the bias scales with $nr_k^{-1} \frac{\|\Sigma_{T,-k}\|}{\|\Sigma_{S,-k}\|}$, meaning that we only require $\frac{\|\Sigma_{T,-k}\|}{\|\Sigma_{S,-k}\|} = \mathcal{O}(r_k/n)$, which is a less restrictive condition for larger $r_k$. Thus, over-parameterization improves the robustness of the estimation bias against covariate shift in the minor direction.

**Remark 2** (Sample complexity). We have assumed $n > cN$ in Theorem 2. The explicit formula for $N$ is deferred to Theorem 25 and Remark 8. Here we summarize the sample complexity required for the bound to hold. The dependence on $k$ varies between $\Omega(k)$ and $\Omega(k^3)$, depending on the

degree of covariate shift. The optimal case, aligning with the sample complexity of classic linear regression, occurs when $\Sigma_{S,k} \approx \Sigma_{T,k}$. The worst case arises when there is a significant covariate shift in the first $k$ dimensions, such as when the test data lies predominantly in the subspace of the first dimension. This variation in sample complexity under covariate shift parallels the analysis of Ge et al. (2024) (see their Theorem 4.2) for the under-parameterized setting. Additionally, we require $n \gg \lambda + \sum_{j>k} \lambda_j$, ensuring that the regularization is not too strong to introduce a bias exceeding a constant (as reflected in the first term of $B_{\mathrm{ID}}$). On the other hand, we assume $n < r_k$ in the theorem, consistent with the over-parameterized regime and Assumption 1, where the last $d - k$ components are considered to be essentially high-dimensional.

**Remark 3** (Dependence on $L$). Theorem 2 does not explicitly show how the excess risk depends on the condition number $L$ of $A_k$. However, we demonstrate in Theorem 25 that the upper bounds scale at most as $L^2$. Notably, we maintain the same order of dependence on $L$ in each term of the upper bounds as in the analysis by Tsigler & Bartlett (2023) (see their Theorem 5).

Finally, Theorem 2 suggests an $\mathcal{O}(1/n)$ vanishing error under several conditions that naturally follow from the previous discussions, which we now state rigorously. First, the covariate space decomposes into subspaces spanned by low-dimensional major directions and high-dimensional minor directions, with $k = \mathcal{O}(1)$ and $R_k = \Omega(n^2)$. Second, the low-rank covariance structure is preserved after covariate shift, such that $\|\mathcal{T}\|, \frac{\mathrm{tr}[\mathcal{U}]}{\mathrm{tr}[\mathcal{V}]}, nr_k^{-1}\frac{\|\Sigma_{T,-k}\|}{\|\Sigma_{S,-k}\|} = \mathcal{O}(1)$. Third, the signal lies predominantly in the major directions, with $\|\beta_k^\star\|_{\Sigma_{S,k}^{-1}} = \mathcal{O}(1)$ and $\|\beta_{-k}^\star\|_{\Sigma_{S,-k}} = \mathcal{O}(1/\sqrt{n})$. Lastly, the regularization is not excessively strong to introduce a significant bias, with $\widetilde{\lambda} = \lambda + \sum_{j>k} \lambda_j = \mathcal{O}(\sqrt{n})$.

## 4 LARGE SHIFT IN MINOR DIRECTIONS

In the previous section, we established an upper bound for over-parameterized ridge regression under covariate shift. We showed that when the shift in the minor directions is controlled—specifically, when the overall magnitude of $\Sigma_{T,-k}$ is small—"benign overfitting" also occurs under covariate shift. However, when the shift in minor directions is significant, meaning the target covariance matrix has many large eigenvalues with corresponding eigenvectors outside the major directions, the excess risk for ridge regression deteriorates. In this section, we further illustrate the limitations of ridge regression in such cases by providing a lower bound for its performance for large distribution shifts in the minor directions. We show that, in certain instances, ridge regression can only achieve the slow statistical rate of $\mathcal{O}(1/\sqrt{n})$ for the excess risk. On the other hand, it is natural to consider alternative algorithms to ridge regression. We demonstrate that even with a large shift in the minor directions, Principal Component Regression (PCR) is guaranteed to achieve the fast statistical rate $\mathcal{O}(1/n)$ in the same instances, provided that the signal $\beta^\star$ lies primarily within the subspace spanned by the major directions. Moreover, PCR does not require the minor directions to have a high effective rank in the source distribution, highlighting its advantage over ridge regression in such cases. Throughout this section, we maintain the setup and source covariance structure described in Section 2. However, Assumption 1 is no longer required.

### 4.1 SLOW RATE FOR RIDGE REGRESSION

In this subsection, we demonstrate the limitations of ridge regression when the overall magnitude of $\Sigma_{T,-k}$ is large. Consider an instance where $\Sigma_S$ has its first $k$ components as $\Theta(1)$, while the minor directions have eigenvalues of $o(1)$. If we set $\Sigma_T = I_d$, in contrast to the "benign overfitting" regime described in Theorem 2, ridge regression will have a large excess risk for this instance. Although the signal from the major directions is effectively captured, the signal in the minor directions is nearly lost. Unlike the case in Section 3, here, the estimation error in the minor directions is crucial because the target distribution has significant components in these directions. We formalize this intuitive example through the following theorems:

**Corollary 3.** For some absolute constants $C_1, C_2$, consider the following instance of $\Sigma_S$:

$$\lambda_1 = \cdots = \lambda_k = 1, \quad \lambda_{k+1} = \cdots = \lambda_{k+\lfloor\frac{\sqrt{n}}{C_2}\rfloor} = \frac{C_1}{\sqrt{n}}, \quad \lambda_{k+\lfloor\frac{\sqrt{n}}{C_2}\rfloor+1} = \cdots = \lambda_d = 0.$$

Assume $\Sigma_{T,-k} = \mathbf{0}, \Sigma_{T,k} = I_k$, and $\beta^\star_{-k} = 0$. By choosing $\lambda = \sqrt{n}$, under the same conditions of Theorem 2, we can bound the excess risk of the ridge estimator with probability at least $1 - 3\delta$:

$$\mathbb{E}_\epsilon\big[\mathcal{R}\big(\widehat{\beta}(Y)\big)\big] \leq \mathcal{O}\Big(\frac{v^2 k + \|\beta^\star\|^2}{n}\Big).$$

**Remark 4.** Corollary 3 is a direct application of Theorem 2.

**Theorem 4.** Consider the same instance of $\Sigma_S$ as in Corollary 3. Assume $\Sigma_T = I_d$ and $\lambda = \sqrt{n}$. There exists an absolute constant $C > 0$, such that for some $0 < \delta < 1$, $N_2 > 0$ and for any $n > N_2$, with probability at least $1 - \delta$, we have $V \geq Cv^2$.

For any $\lambda > 0$, with probability at least $1 - \delta$, the excess risk of the ridge estimator satisfies:

$$\mathbb{E}_\epsilon\big[\mathcal{R}\big(\widehat{\beta}(Y)\big)\big] \geq C \frac{\|\beta^\star\|^2 \wedge v^2}{\sqrt{n}}.$$

From Theorem 4, we observe that when $\Sigma_T = I_d$, the performance of ridge regression deteriorates compared to the case where $\Sigma_{T,-k} = \mathbf{0}$. If we set $\lambda = \sqrt{n}$ as in Corollary 3, ridge regression incurs a constant excess risk under covariate shift while achieving an in-distribution error rate of $\mathcal{O}(1/n)$. Furthermore, Theorem 4 shows no matter how we choose the regularization parameter $\lambda$, the excess risk is always lower bounded by the slow statistical rate $\mathcal{O}(1/\sqrt{n})$, which is worse than the fast rate of $\mathcal{O}(1/n)$. However, as we will prove in the next subsection, Principal Component Regression (PCR) can achieve an excess risk of $\mathcal{O}(1/n)$ under this instance, even with $\Sigma_T = I_d$.

## 4.2 FAST RATE FOR PRINCIPAL COMPONENT REGRESSION

Ridge regression faces significant limitations when there is a large shift in the minor directions. In Section 3.1, it was shown that the signal in the minor directions, $\beta^\star_{-k}$, is nearly lost when projected from the high-dimensional ambient space onto the low-dimensional sample space. In other words, learning the true signal from the minor directions is essentially impossible. Therefore, in this subsection, we continue to focus on the scenario where the true signal $\beta^\star$ primarily resides in the major directions. In this case, Principal Component Regression (PCR) emerges as a natural algorithm that estimates the space spanned by the major directions and performs regression on that subspace.

**Principal Component Regression (PCR).**

- **Step 1: Obtain an estimator $\widehat{U}$ of the top-$k$ subspace of $\Sigma_S$.** For simplicity, we assume a sample size of $2n$ and use the first half of the data to compute $\widehat{U}$ by principal component analysis (PCA) on the sample covariance matrix $\widehat{\Sigma}_S := \frac{1}{n} X^T X$. Specifically, $\widehat{U} = (\widehat{u}_1, \cdots, \widehat{u}_k)$ where $\widehat{u}_i$ is the $i$-th eigenvector of $\widehat{\Sigma}_S$.

- **Step 2: Use the data projected on $\widehat{U}$ to conduct linear regression.** With a little abuse of notation, we use $X \in \mathbb{R}^{n \times d}$ to denote the data matrix $(x_{n+1}, \cdots, x_{2n})^T$, and $Y \in \mathbb{R}^n$ to denote $(y_{n+1}, \cdots, y_{2n})^T$. If we let $Z := X\widehat{U} \in \mathbb{R}^{n \times k}$ be the projected data matrix, the estimator $\widehat{\beta}$ we obtained is given by

$$\widehat{\beta} = \widehat{U}(Z^T Z)^{-1} Z^T Y = \widehat{U}(\widehat{U}^T X^T X \widehat{U})^{-1} \widehat{U}^T X^T Y.$$

Consider the scenario where the last $d - k$ components of the true signal $\beta^*$ is exactly zero, i.e., $\beta^\star_{-k} = 0$. In this case, if the subspace represented by $\widehat{U}$ perfectly matches the subspace represented by $U = \begin{pmatrix} I_k \\ 0 \end{pmatrix} \in \mathbb{R}^{d \times k}$, corresponding to the first $k$ components, then PCR performs linear regression using only the first $k$ components of the covariates. As a result, the excess risk would just be the usual variance of linear regression in the major directions. In this scenario, regardless of the norm $\|\Sigma_{T,-k}\|$, the PCR estimator assigns coefficients of zero to the last $d - k$ components, thus avoiding any large excess risk. Furthermore, if the distance between $\widehat{U}$ and $U$ is nonzero, an additional term in the excess risk will arise due to the estimation error of $\widehat{U}$. We formalize this intuition with the following upper bound for the excess risk of PCR. To facilitate the presentation, we introduce a measure of the estimation accuracy of $\widehat{U}$. We define $\Delta = \text{dist}(\widehat{U}, U) := \|UU^T - \widehat{U}\widehat{U}^T\|$, which represents the distance between the subspaces spanned by the columns of $\widehat{U}$ and $U$. Then we present the following theorem.

**Theorem 5.** Assume $\beta^\star_{-k} = 0$. If $\Delta \leq \Theta$, for any $0 < \delta < 1$ and any $n \geq N_1$, we can bound the excess risk of PCR estimator $\widehat{\beta}$ with probability $1 - \delta$:

$$\mathbb{E}_\epsilon\big[\mathcal{R}\big(\widehat{\beta}(Y)\big)\big] \leq \mathcal{O}\left(v^2 \frac{\text{tr}(\mathcal{T})}{n} + \|\beta^\star\|^2 \Big(\frac{\lambda_1}{\lambda_k}\Big)^2 \|\Sigma_T\| \Delta^2\right),$$

where $\Theta^{-1} = \text{Poly}(\lambda_1\lambda_k^{-1}, \|\Sigma_T\|\lambda_k^{-1}, k\,\text{tr}(\mathcal{T})^{-1})$ and
$N_1 = \text{Poly}(\sigma, \lambda_1\lambda_k^{-1}, \|\Sigma_T\|\lambda_k^{-1}, k\ln(1/\delta), k\,\text{tr}(\mathcal{T})^{-1})$.

**Remark 5.** Theorem 5 is a special case of Lemma 31. For explicit formulas of $\Theta$ and $N_1$, as well as an upper bound for cases where $\beta^\star_{-k} \neq 0$, refer to Lemma 31 for details.

The excess risk upper bound provided by Theorem 5 consists of two terms. The variance term $\text{tr}(\mathcal{T})/n$ is incurred by the nature of linear regression on the major directions and remains unavoidable even when the subspace estimation is exact (i.e., $\Delta = 0$). This term also appears as the first variance term in Theorem 2, and exactly matches the sharp rate $\text{tr}[\Sigma_S^{-1}\Sigma_T]/n$ for under-parameterized linear regression under covariate shift (Ge et al., 2024). The second term $\|\beta^\star\|^2(\frac{\lambda_1}{\lambda_k})^2\|\Sigma_T\|\Delta^2$ represents the bias induced by the subspace estimation error in Step 1, which exhibits a quadratic dependence on $\Delta$. By combining Theorem 5 with a bound on $\Delta$, we can derive an end-to-end excess risk upper bound of PCR. We present the following lemma to control $\Delta$.

**Lemma 6.** With probability at least $1 - \delta$, if $n \geq r + \ln(1/\delta)$, we have

$$\Delta \leq \mathcal{O}\left(\sigma^4 \frac{\lambda_1}{\lambda_k - \lambda_{k+1}} \sqrt{\frac{r + \ln\frac{1}{\delta}}{n}}\right),$$

where $r = \lambda_1^{-1}\sum_{i=1}^d \lambda_i$ is the effective rank of the entire covariance matrix $\Sigma_S$.

**Remark 6.** Lemma 6 shows that $\Delta$ depends on several quantities: the eigenvalue gap between the major and minor directions, i.e., $\lambda_k - \lambda_{k+1}$, and the effective rank $r$. We observe that $\Delta$ will be small if the major and minor directions are well separated, meaning $\lambda_k - \lambda_{k+1}$ is large, and the minor directions are relatively small compared to $\lambda_1$.

Combining Theorem 5 with Lemma 6, an end-to-end error bound for PCR directly follows (see Theorem 29 for a detailed statement), suggesting that PCR will achieve a small excess risk as long as the major and minor directions are well separated, and the effective rank of the entire source covariance matrix is small. In contrast to ridge regression, PCR does not rely on the minor components having a high effective rank. This highlights the superiority of PCR over ridge regression in certain scenarios.

As an example, consider the instance in Theorem 4, where $k, \|\Sigma_T\|, \lambda_1, \lambda_k$ are all $\Theta(1)$. In this case, the variance term scales as $1/n$, and the bias term scales as $\mathcal{O}(\Delta^2)$. Since $r = \Theta(1)$ in this instance, we have $\Delta \leq \mathcal{O}(1/\sqrt{n})$. Consequently, PCR achieves a $\mathcal{O}(1/n)$ rate in this instance, even when $\Sigma_T = I_d$. Compared with the excess risk for ridge regression, which is at least $1/\sqrt{n}$, PCR shows its superiority against ridge regression when there is a large shift in the minor directions.

## 5  CONCLUSION AND DISCUSSION

In conclusion, we provide an instance-dependent upper bound on the excess risk for ridge regression under general covariate shift. Our findings demonstrate that "benign overfitting" also occurs in OOD generalization when the shift in the minor directions is well controlled. We also investigate the regime with a large shift in the minor directions, where ridge regression may incur a large excess risk, whereas Principal Component Regression (PCR) exhibits superior performance.

Our work opens several directions for future research. First, while we have established a lower bound for ridge regression in certain instances, a key challenge remains in deriving a general lower bound that matches our upper bounds, offering a more precise characterization of the excess risk under covariate shift. Second, our analysis has focused on linear models as an initial step in understanding over-parameterized OOD problems. Extending this investigation to more complex, non-linear models would be a intriguing direction for future exploration.

### ACKNOWLEDGMENTS

This work is partially supported by the National Science Foundation CAREER grant NSF-IIS-2239297 and NSF grants DMS-2210833 and DMS-2412029.

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

# A  SIMULATION

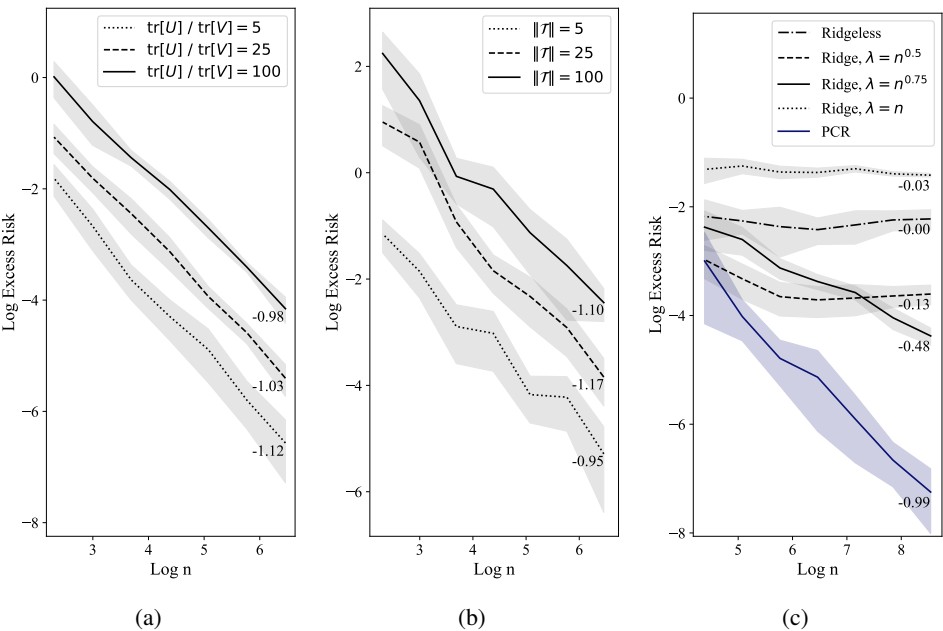

Figure 1: Simulation results for excess risks across varying training sample sizes. The shaded regions represent standard errors of 10 runs, using different samples of training and test sets. The slope of the fitted OLS model is marked along each curve. (a)(b) Minimum norm interpolation under distinct target covariance matrices with small shifts in minor directions. The source covariance matrix remains constant. (a) Various magnitudes of shifts in minor directions, with $\|\mathcal{T}\| = 1$. (b) Various magnitudes of shifts in major directions, with $\mathrm{tr}[\mathcal{U}]/\mathrm{tr}[\mathcal{V}] = 1$. (c) Ridge and PCR under large shifts in minor directions, following the setting of Theorem 4.

## A.1  BENIGN-OVERFITTING: SMALL SHIFT IN MINOR DIRECTIONS

We simulate the covariate shift discussed in section 3, where the overall magnitude of the target covariance matrix's minor directions is comparable to that of the source. Theorem 2 establishes an $\mathcal{O}(1/n)$ excess risk rate for ridge regression under certain benign-overfitting conditions. Specifically, for the training data, we assume $k = \mathcal{O}(1)$, $R_k = \Omega(n^2)$, $\|\beta_k^\star\|_{\Sigma_{S,k}^{-1}} = \mathcal{O}(1)$, $\|\beta_{-k}^\star\|_{\Sigma_{S,-k}} = \mathcal{O}(1/\sqrt{n})$, and $\widetilde{\lambda} = \mathcal{O}(\sqrt{n})$. For the test data, we assume $\|\mathcal{T}\|, \frac{\mathrm{tr}[\mathcal{U}]}{\mathrm{tr}[\mathcal{V}]}, nr_k^{-1}\frac{\|\Sigma_{T,-k}\|}{\|\Sigma_{S,-k}\|} = \mathcal{O}(1)$. In the experiment, data is generated according to these conditions.

$$y = x^T \beta^\star + \epsilon,$$

where $\beta^\star \in \mathbb{R}^{k+n^2}$, with $\beta_k^\star = (\frac{1}{\sqrt{k}}, ..., \frac{1}{\sqrt{k}})^T$, $\beta_{-k}^\star = \mathbf{0}$ and $k = 10$. The noise $\epsilon$ follows a centered gaussian distribution with variance $0.1$, and $x$ is drawn from a multivariate normal distribution with zero mean and a source covariance matrix $\Sigma_S = \mathrm{diag}(I_k, n^{-1.5}I_{n^2})$. The target covariance matrix is $\Sigma_T = \mathrm{diag}(\Sigma_{T,k}, \Sigma_{T,-k})$ where $\Sigma_{T,k}$ and $\Sigma_{T,-k}$ are randomly generated and scaled to achieve specific values for $\|\mathcal{T}\|$ and $\frac{\mathrm{tr}[\mathcal{U}]}{\mathrm{tr}[\mathcal{V}]}$, respectively. At the same time, we do not explicitly control $nr_k^{-1}\frac{\|\Sigma_{T,-k}\|}{\|\Sigma_{S,-k}\|}$, because it equals $\frac{1}{n}\frac{\|\Sigma_{T,-k}\|}{\|\Sigma_{S,-k}\|}$ in this setting and is typically bounded for a randomly generated $\Sigma_{T,-k}$. We run minimum norm interpolation (ridgeless regression) with $\lambda = 0$.

The source covariance matrix $\Sigma_S$ is fixed for all experiments while we vary the target covariance matrices. To study covariate shifts in major directions, we vary $\|\mathcal{T}\|$ among 5, 25, 100 and keep $\frac{\mathrm{tr}[\mathcal{U}]}{\mathrm{tr}[\mathcal{V}]} = 1$. To study covariate shifts in minor directions, we vary $\frac{\mathrm{tr}[\mathcal{U}]}{\mathrm{tr}[\mathcal{V}]} = 1$ among 5, 25, 100 and keep $\|\mathcal{T}\| = 1$. For each pair of $\|\mathcal{T}\|$ and $\frac{\mathrm{tr}[\mathcal{U}]}{\mathrm{tr}[\mathcal{V}]}$, we generate training samples of various sizes $n$ and

1000 test samples. For each $n$, a target covariance matrix $\Sigma_T$ is randomly generated to satisfy the specified $\|\mathcal{T}\|$ and $\frac{\mathrm{tr}[\mathcal{U}]}{\mathrm{tr}[\mathcal{V}]}$. We run 10 experiments for each set of $(\Sigma_S, \Sigma_T, n)$, using independently sampled training sets and test sets, and the mean and standard error of the excess risks are reported.

The results are shown in Figure 1a, 1b. The fast rate $\mathcal{O}(1)$ of minimum norm interpolation is confirmed, as the log-log plot of excess risk versus $n$ has a slope near -1 across all combinations of $\|\mathcal{T}\|$ and $\frac{\mathrm{tr}[\mathcal{U}]}{\mathrm{tr}[\mathcal{V}]}$. The excess risk increases with larger $\frac{\mathrm{tr}[\mathcal{U}]}{\mathrm{tr}[\mathcal{V}]}$, indicating a greater shift in minor directions. Similarly, the excess risk increases with larger $\|\mathcal{T}\|$, indicating a greater shift in major directions.

### A.2 RIDGE V.S. PCR: LARGE SHIFT IN MINOR DIRECTIONS

Theorem 4 identifies a setting where large covariate shifts occur in minor directions of the covariance matrix, leading to a lower bound of $\mathcal{O}(1/\sqrt{n})$ on the excess risk for ridge regression, while Principal Component Regression (PCR) achieves the fast rate of $\mathcal{O}(1)$. We design a simulation experiment under the same instance of the source and target covariance matrices. Specifically, data is generated as:

$$y = x^T \beta^\star + \epsilon,$$

where $\beta^\star \in \mathbb{R}^{k+\lfloor\sqrt{n}\rfloor}$, with $\beta_k^\star = (\frac{1}{\sqrt{k}}, ..., \frac{1}{\sqrt{k}})^T$, $\beta_{-k}^\star = \mathbf{0}$, and $k = 10$. The noise $\epsilon$ is drawn from a centered gaussian distribution with variance 0.1, and $x$ follows a multivariate normal distribution with zero mean. The source covariance matrix is $\Sigma_S = \mathrm{diag}(I_k, n^{-0.5} I_{\lfloor\sqrt{n}\rfloor})$, and the target covariance matrix is $\Sigma_T = I_{k+\lfloor\sqrt{n}\rfloor}$.

We evaluate ridge regression with various regularization strengths: $\lambda = 10^{-8}, n^{0.5}, n^{0.75}, n$. Here, we use $\lambda = 10^{-8}$ to approximate ridgeless regression, which has a singular solution under this setup. We compare PCR to ridge regression for different training sample sizes $n$. The test set contains 1000 samples. For each $n$, 10 experiments are conducted with independently sampled training and test sets. We report the mean and standard error of the excess risks.

Figure 1c present the results. As expected, PCR nearly achieves the fast rate of $\mathcal{O}(1/n)$, with the log-log slope of excess risk versus $n$ being -0.99. In contrast, the optimal rate of ridge regression is $\mathcal{O}(n^{-0.48})$, achieved with $\lambda = n^{0.75}$. This aligns with the lower bound of $\mathcal{O}(1/\sqrt{n})$ from Theorem 4, and its proof also suggests $\lambda = n^{0.75}$ as the optimal regularization strength. Additionally, ridge regression exhibits excess risks above a constant for certain choices of $\lambda$.

## B RIDGE REGRESSION

Let $X = (x_1, ..., x_n)^T \in \mathbb{R}^{n \times d}$, $Y = (y_1, ..., y_n)^T \in \mathbb{R}^n$ and $\epsilon = (\epsilon_1, ..., \epsilon_n)^T \in \mathbb{R}^n$. We denote the first $k$ columns of $X$ as $X_k$ and the remaining $d - k$ columns as $X_{-k}$. Similarly, $\beta_k^\star$ and $\beta_{-k}^\star$ represent the corresponding components of $\beta^\star$. $\Sigma_{S,k}, \Sigma_{S,-k}$ are the corresponding blocks on the diagonal of $\Sigma_S$. The $i$-th eigenvalue of a matrix is denoted by $\mu_i(\cdot)$. Define $Z = X\Sigma_S^{-1/2}$, where the rows of $Z$ are i.i.d. centered isotropic random vectors. Additionally, we assume the rows of $Z$ are $\sigma$-sub-gaussian, where the sub-gaussian norm is defined as follows.

For a random variable $s$, the sub-gaussian norm $\|s\|_{\psi_2}$ is given by:

$$\|s\|_{\psi_2} = \inf\left\{t > 0 : \mathbb{E}\left[\exp\frac{s^2}{t^2}\right] \leq 2\right\}.$$

For a random vector $S$, the sub-gaussian norm $\|S\|_{\psi_2}$ is given by:

$$\|S\|_{\psi_2} = \sup_{v \neq 0} \frac{\|\langle S, v\rangle\|_{\psi_2}}{\|v\|}.$$

For $\lambda \geq 0$, consider the ridge estimator:

$$\widehat{\beta}(Y) = X^T(XX^T + \lambda I_n)^{-1}Y$$

$$= X^T(XX^T + \lambda I_n)^{-1}X\beta^\star + X^T(XX^T + \lambda I_n)^{-1}\epsilon$$
$$= \widehat{\beta}(X\beta^\star) + \widehat{\beta}(\epsilon),$$

where we define $\widehat{\beta}(X\beta^\star) = X^T(XX^T + \lambda I_n)^{-1}X\beta^\star$ and $\widehat{\beta}(\epsilon) = X^T(XX^T + \lambda I_n)^{-1}\epsilon$. Additionally, we define $\widetilde{\Sigma}_S = \Sigma_S + \frac{\lambda}{n}I_d$. The effective rank of $\widetilde{\Sigma}_{S,k}$ is defined as $r_k = \lambda_{k+1}^{-1}(\lambda + \sum_{j>k}\lambda_j)$.

**Assumption 2** (CondNum$(k, \delta, L)$). Define a matrix $A_k = \lambda I_n + X_{-k}X_{-k}^T$. With probability at least $1 - \delta$, $A_k$ is positive definite and has a condition number no greater than $L$, i.e.,

$$\frac{\mu_1(A_k)}{\mu_n(A_k)} \le L.$$

### B.1 CONCENTRATION INEQUALITIES

Denote the element of a matrix $X$ in the $i$-th row and the $j$-th column as $X[i, j]$, and the $i$-th row of the matrix $X$ as $X[i, *]$.

**Lemma 7** (Lemma 20 of Tsigler & Bartlett (2023)). Let $z$ be a sub-gaussian vector in $\mathbb{R}^p$ with $\|z\|_{\psi_2} \le \sigma$, and consider $\Sigma = \text{diag}(\lambda_1, \dots, \lambda_p)$ where the sequence $\{\lambda_j\}_{j=1}^p$ is positive and non-increasing. Then there exists some absolute constant $c$, for any $t > 0$, with probability at least $1 - 2e^{-t/c}$:

$$\|\Sigma^{1/2}z\|^2 \le c\sigma^2 \left( t\lambda_1 + \sum_{j=1}^p \lambda_j \right).$$

**Lemma 8** (Lemma 23 of Tsigler & Bartlett (2023)). Let $\mathring{A}_k$ represent the matrix $X_{-k}X_{-k}^T$ with its diagonal elements set to zero:

$$\mathring{A}_k[i, j] = (1 - \delta_{i,j})(X_{-k}X_{-k}^T)[i, j].$$

Then there exists some absolute constant $c$, for any $t > 0$, with probability at least $1 - 4e^{-t/c}$:

$$\|\mathring{A}_k\| \le c\sigma^2 \sqrt{(t + n) \left( \lambda_{k+1}^2(t + n) + \sum_{j>k} \lambda_j^2 \right)}.$$

**Lemma 9** (Lemma 21 of Tsigler & Bartlett (2023)). Suppose $\{z_i\}_{i=1}^n$ is a sequence of independent isotropic sub-gaussian random vectors, where $\|z_i\|_{\psi_2} \le \sigma$. Let $\Sigma = \text{diag}(\lambda_1, \dots, \lambda_p)$ represent a diagonal matrix with a positive, non-increasing sequence $\{\lambda_i\}_{i=1}^p$. Then there exists some absolute constant $c$, for any $t \in (0, n)$, with probability at least $1 - 2e^{-ct}$:

$$(n - \sqrt{nt}\sigma^2) \sum_{j=1}^p \lambda_j \le \sum_{i=1}^n \|\Sigma^{1/2}z_i\|^2 \le (n + \sqrt{nt}\sigma^2) \sum_{j=1}^p \lambda_j.$$

**Lemma 10.** There exists a constant $c_x$, depending only on $\sigma$, such that for any $n$ satisfying $n\lambda_{k+1} \le \left( \lambda + \sum_{j>k}\lambda_j \right)$, under the assumption CondNum$(k, \delta, L)$ (Assumption 2), with probability at least $1 - \delta - c_x e^{-n/c_x}$:

$$\frac{1}{c_x L} \left( \lambda + \sum_{j>k}\lambda_j \right) \le \mu_n(A_k) \le \mu_1(A_k) \le c_x \left( \lambda + \sum_{j>k}\lambda_j \right).$$

$$\mu_1(X_{-k}X_{-k}^T) \le c_x \left( n\lambda_{k+1} + \sum_{j>k}\lambda_j \right).$$

*Proof.* This result follows from the proof of Lemma 3 in Tsigler & Bartlett (2023), which establishes both upper and lower bounds of $\mu_1(A_k)$. By combining the lower bound with the assumption CondNum, we derive a lower bound of $\mu_n(A_k)$. For completeness, we restate the entire proof here.

According to lemma 7 and lemma 8, there exists an absolute constant c, such that for any $t > 0$:

1. for all $1 \leq i \leq n$, with probability at least $1 - 2e^{-t/c}$:

$$\|X_{-k}[i, *]\|^2 \leq c\sigma^2 \left( t\lambda_{k+1} + \sum_{j>k} \lambda_j \right).$$

2. with probability at least $1 - 4e^{-t/c}$:

$$\|\mathring{A}_k\| \leq c\sigma^2 \sqrt{(t+n) \left( \lambda_{k+1}^2(t+n) + \sum_{j>k} \lambda_j^2 \right)}.$$

Since $\mu_1(A_k) \leq \lambda + \|\mathring{A}_k\| + \max_i \|X_{-k}[i, *]\|^2$, by setting $t = n$, we have with probability at least $1 - (2n + 4)e^{-n/c}$:

$$\mu_1(A_k) \leq \lambda + c\sigma^2 \left( n\lambda_{k+1} + \sum_{j>k} \lambda_j + \sqrt{(2n\lambda_{k+1})^2 + 2n \sum_{j>k} \lambda_j^2} \right)$$

$$\leq \lambda + c\sigma^2 \left( n\lambda_{k+1} + \sum_{j>k} \lambda_j + 2n\lambda_{k+1} + \sqrt{2n \sum_{j>k} \lambda_j^2} \right)$$

$$\leq \lambda + c\sigma^2 \left( n\lambda_{k+1} + \sum_{j>k} \lambda_j + 2n\lambda_{k+1} + \sqrt{2n\lambda_{k+1} \sum_{j>k} \lambda_j} \right)$$

$$\leq \lambda + c\sigma^2 \left( n\lambda_{k+1} + \sum_{j>k} \lambda_j + 2n\lambda_{k+1} + n\lambda_{k+1} + \frac{1}{2} \sum_{j>k} \lambda_j \right)$$

$$\leq \lambda + 4c\sigma^2 \left( n\lambda_{k+1} + \sum_{j>k} \lambda_j \right)$$

$$\leq \max\{1, 4c\sigma^2\} \left( \lambda + \sum_{j>k} \lambda_j + n\lambda_{k+1} \right)$$

$$\leq 2\max\{1, 4c\sigma^2\} \left( \lambda + \sum_{j>k} \lambda_j \right). \tag{2}$$

The last inequality follows from $n\lambda_{k+1} \leq \left( \lambda + \sum_{j>k} \lambda_j \right)$. Similarly,

$$\mu_1(X_{-k}X_{-k}^T) \leq 4c\sigma^2 \left( n\lambda_{k+1} + \sum_{j>k} \lambda_j \right). \tag{3}$$

On the other hand, by applying Lemma 9 with $t = \frac{n}{4\sigma^4}$, there exists an absolute constant $c'$, such that with probability at least $1 - 2\exp\left\{ -\frac{c'}{4\sigma^4} n \right\}$:

$$\sum_{i=1}^{n} \|X_{-k}[i, *]\|^2 \geq \frac{1}{2} n \sum_{j>k} \lambda_j.$$

On this event,

$$\mu_1(A_k) \geq \lambda + \frac{1}{n} \operatorname{tr}(X_{-k}X_{-k}^T)$$

$$= \lambda + \frac{1}{n} \sum_{i=1}^{n} \|X_{-k}[i, *]\|^2$$

$$\geq \lambda + \frac{1}{2} \sum_{j>k} \lambda_j$$

$$\geq \frac{1}{2} \left( \lambda + \sum_{j>k} \lambda_j \right).$$

By the assumption $\text{CondNum}(k, \delta, L)$, with probability at least $1 - \delta - 2\exp\left\{-\frac{c'}{4\sigma^4}n\right\}$:

$$\mu_n(A_k) \geq \frac{1}{L}\mu_1(A_k) \geq \frac{1}{2L} \left( \lambda + \sum_{j>k} \lambda_j \right). \tag{4}$$

Combining Equation 2, 3 and 4, there exists a constant $c_x$ depending only on $\sigma$, such that with probability at least $1 - \delta - c_x e^{-n/c_x}$:

$$\frac{1}{c_x L} \left( \lambda + \sum_{j>k} \lambda_j \right) \leq \mu_n(A_k) \leq \mu_1(A_k) \leq c_x \left( \lambda + \sum_{j>k} \lambda_j \right).$$

$$\mu_1(X_{-k}X_{-k}^T) \leq c_x \left( n\lambda_{k+1} + \sum_{j>k} \lambda_j \right).$$

$\square$

**Lemma 11.** There exists a constant $c_x$ depending only on $\sigma$, such that with probability at least $1 - \delta$, if $n > k + \ln(1/\delta)$,

$$\left\| \frac{1}{n}X_k^T X_k - \Sigma_{S,k} \right\| \leq c_x \lambda_1 \sqrt{\frac{k + \ln\frac{1}{\delta}}{n}}.$$

*Proof.* This follows directly from Theorem 5.39 and Remark 5.40 of Vershynin (2010), which shows there exists a constant $c'_x$ depending only on $\sigma$, such that for any $t \geq 0$, with probability at least $1 - 2\exp\{-t^2/c'_x\}$:

$$\left\| \frac{1}{n}X_k^T X_k - \Sigma_{S,k} \right\| \leq \lambda_1 \max\left\{ c'_x\sqrt{\frac{k}{n}} + \frac{t}{\sqrt{n}}, \left( c'_x\sqrt{\frac{k}{n}} + \frac{t}{\sqrt{n}} \right)^2 \right\}.$$

Taking $t = \sqrt{c'_x \ln(2/\delta)}$ completes the proof. $\square$

**Corollary 12.** Under the same conditions as in Lemma 11, and on the same event, the following holds:

$$\left\| \left(X_k^T X_k\right)^{\frac{1}{2}} - \sqrt{n}\Sigma_{S,k}^{\frac{1}{2}} \right\| \leq c_x \sqrt{k + \ln\frac{1}{\delta}} \lambda_1 \lambda_k^{-\frac{1}{2}}.$$

*Proof.* According to Proposition 3.2 of van Hemmen & Ando (1980), for any positive semi-definite matrix $A, B \in \mathbb{R}^k$, we have

$$\|A - B\| \geq \left( \mu_k\left(A^{\frac{1}{2}}\right) + \mu_k\left(B^{\frac{1}{2}}\right) \right) \left\| A^{\frac{1}{2}} - B^{\frac{1}{2}} \right\|.$$

Therefore,

$$\left\| \left(X_k^T X_k\right)^{\frac{1}{2}} - \sqrt{n}\Sigma_{S,k}^{\frac{1}{2}} \right\| \leq \frac{1}{\mu_k\left(\sqrt{n}\Sigma_{S,k}^{\frac{1}{2}}\right)} \left\| X_k^T X_k - n\Sigma_{S,k} \right\|$$

$$= \sqrt{n}\lambda_k^{-\frac{1}{2}} \left\| \frac{1}{n}X_k^T X_k - \Sigma_{S,k} \right\|.$$

By applying Lemma 11, the proof is complete. $\square$

**Lemma 13.** There exists a constant $c_x$ depending only on $\sigma$, such that for any $n > c_x k$, with probability at least $1 - 2e^{-n/c_x}$:

$$\frac{1}{c_x}n \leq \mu_k\left(\Sigma_{S,k}^{-\frac{1}{2}} X_k^T X_k \Sigma_{S,k}^{-\frac{1}{2}}\right) \leq \mu_1\left(\Sigma_{S,k}^{-\frac{1}{2}} X_k^T X_k \Sigma_{S,k}^{-\frac{1}{2}}\right) \leq c_x n.$$

*Proof.* According to Theorem 5.39 of Vershynin (2010), there exists a constant $c_x'$ depending only on $\sigma$, such that for any $t \geq 0$, with probability at least $1 - 2\exp\{-t^2/c_x'\}$:

$$\mu_k\left(\Sigma_{S,k}^{-\frac{1}{2}} X_k^T X_k \Sigma_{S,k}^{-\frac{1}{2}}\right) \geq \left(\sqrt{n} - c_x'\sqrt{k} - t\right)^2.$$

$$\mu_1\left(\Sigma_{S,k}^{-\frac{1}{2}} X_k^T X_k \Sigma_{S,k}^{-\frac{1}{2}}\right) \leq \left(\sqrt{n} + c_x'\sqrt{k} + t\right)^2.$$

Let $t = \frac{1}{2}\sqrt{n}$. For $n > 16(c_x')^2 k$, with probability at least $1 - 2\exp\{-n/(4c_x')\}$:

$$\mu_k\left(\Sigma_{S,k}^{-\frac{1}{2}} X_k^T X_k \Sigma_{S,k}^{-\frac{1}{2}}\right) \geq \left(\sqrt{n} - \frac{1}{4}\sqrt{n} - \frac{1}{2}\sqrt{n}\right)^2 = \frac{1}{16}n.$$

$$\mu_1\left(\Sigma_{S,k}^{-\frac{1}{2}} X_k^T X_k \Sigma_{S,k}^{-\frac{1}{2}}\right) \leq \left(\sqrt{n} + \frac{1}{4}\sqrt{n} + \frac{1}{2}\sqrt{n}\right)^2 = \frac{49}{16}n.$$

By taking $c_x = \max\left\{16(c_x')^2, 4c_x', 16\right\}$, the proof is complete. $\square$

**Remark 7.** On the same event, the following inequalities also hold:

$$\mu_1(X_k^T X_k) \leq \|\Sigma_{S,k}\| \left\|\Sigma_{S,k}^{-\frac{1}{2}} X_k^T X_k \Sigma_{S,k}^{-\frac{1}{2}}\right\| \leq c_x \lambda_1 n.$$

$$\mu_k(X_k^T X_k) \geq \mu_k(\Sigma_{S,k}) \mu_k\left(\Sigma_{S,k}^{-\frac{1}{2}} X_k^T X_k \Sigma_{S,k}^{-\frac{1}{2}}\right) \geq \frac{1}{c_x}\lambda_k n.$$

**Lemma 14.** There exists a constant $c_x$ depending only on $\sigma$, with probability at least $1 - 2e^{-n/c_x}$:

$$\text{tr}\left(X_{-k}\Sigma_{T,-k}X_{-k}^T\right) \leq c_x n \, \text{tr}\left(\Sigma_{S,-k}^{\frac{1}{2}}\Sigma_{T,-k}\Sigma_{S,-k}^{\frac{1}{2}}\right).$$

*Proof.* According to Hanson-Wright Inequality (Vershynin, 2018), there exists an absolute constant $c$, such that for any $1 \leq i \leq n$,

$$\left\|Z_{-k}[i,*]\Sigma_{S,-k}^{\frac{1}{2}}\Sigma_{T,-k}\Sigma_{S,-k}^{\frac{1}{2}}Z_{-k}[i,*]^T\right\|_{\psi_1} \leq c\sigma^2 \left\|\Sigma_{S,-k}^{\frac{1}{2}}\Sigma_{T,-k}\Sigma_{S,-k}^{\frac{1}{2}}\right\|_F$$

$$\leq c\sigma^2 \, \text{tr}\left(\Sigma_{S,-k}^{\frac{1}{2}}\Sigma_{T,-k}\Sigma_{S,-k}^{\frac{1}{2}}\right).$$

By Bernstein Inequality (Proposition 5.16 of Vershynin (2010)), there exists an absolute constant $c'$, for any $t \geq 0$,

$$\mathbb{P}\left\{\frac{1}{n}\left|\sum_{i=1}^n \left[Z_{-k}[i,*]\Sigma_{S,-k}^{\frac{1}{2}}\Sigma_{T,-k}\Sigma_{S,-k}^{\frac{1}{2}}Z_{-k}[i,*]^T - \text{tr}\left(\Sigma_{S,-k}^{\frac{1}{2}}\Sigma_{T,-k}\Sigma_{S,-k}^{\frac{1}{2}}\right)\right]\right| \geq t\right\}$$

$$\leq 2\exp\left\{-c'n\min\left\{\frac{t^2}{K^2}, \frac{t}{K}\right\}\right\},$$

where $K = \max_i \left\|Z_{-k}[i,*]\Sigma_{S,-k}^{\frac{1}{2}}\Sigma_{T,-k}\Sigma_{S,-k}^{\frac{1}{2}}Z_{-k}[i,*]^T\right\|_{\psi_1}$.

Let $t = c\sigma^2 \, \text{tr}\left(\Sigma_{S,-k}^{\frac{1}{2}}\Sigma_{T,-k}\Sigma_{S,-k}^{\frac{1}{2}}\right)$. Then, with probability at least $1 - 2e^{-c'n}$:

$$\text{tr}\left(X_{-k}\Sigma_{T,-k}X_{-k}^T\right) = \sum_{i=1}^n Z_{-k}[i,*]\Sigma_{S,-k}^{\frac{1}{2}}\Sigma_{T,-k}\Sigma_{S,-k}^{\frac{1}{2}}Z_{-k}[i,*]^T$$

$$\leq (1 + c\sigma^2)n \, \text{tr}\left(\Sigma_{S,-k}^{\frac{1}{2}}\Sigma_{T,-k}\Sigma_{S,-k}^{\frac{1}{2}}\right).$$

By taking $c_x = \max\left\{1 + c\sigma^2, \frac{1}{c'}\right\}$, the proof is complete. $\square$

**Lemma 15.** There exists a constant $c_x$ depending only on $\sigma$, with probablity at least $1 - 2e^{-n/c_x}$:

$$(\beta^\star_{-k})^T X^T_{-k} X_{-k} \beta^\star_{-k} \leq c_x n (\beta^\star_{-k})^T \Sigma_{S,-k} \beta^\star_{-k}.$$

*Proof.* The result follows from the proof of Lemma 3 in Tsigler & Bartlett (2023), which we restate here for completeness. Consider the isotropic vector $\left[(\beta^\star_{-k})^T \Sigma_{S,-k} \beta^\star_{-k}\right]^{-1/2} X_{-k} \beta^\star_{-k}$. For the $i$-th component,

$$\left\| \left[(\beta^\star_{-k})^T \Sigma_{S,-k} \beta^\star_{-k}\right]^{-\frac{1}{2}} X_{-k}[i,*] \beta^\star_{-k} \right\|_{\psi_2} = \left[(\beta^\star_{-k})^T \Sigma_{S,-k} \beta^\star_{-k}\right]^{-\frac{1}{2}} \left\| Z_{-k}[i,*] \Sigma^{\frac{1}{2}}_{S,-k} \beta^\star_{-k} \right\|_{\psi_2}$$

$$\leq \left[(\beta^\star_{-k})^T \Sigma_{S,-k} \beta^\star_{-k}\right]^{-\frac{1}{2}} \sigma \left\| \Sigma^{\frac{1}{2}}_{S,-k} \beta^\star_{-k} \right\|$$

$$= \sigma.$$

By applying Lemma 9 for the sequence $\left\{ \left[(\beta^\star_{-k})^T \Sigma_{S,-k} \beta^\star_{-k}\right]^{-1/2} X_{-k}[i,*] \beta^\star_{-k} \right\}^n_{i=1}$, there exists an absolute constant $c$, for any $t \in (0, n)$, with probability at least $1 - 2e^{-ct}$:

$$\frac{(\beta^\star_{-k})^T X^T_{-k} X_{-k} \beta^\star_{-k}}{(\beta^\star_{-k})^T \Sigma_{S,-k} \beta^\star_{-k}} \leq n + \sqrt{nt}\sigma^2.$$

Let $t = n/4$, with probability at least $1 - 2e^{-cn/4}$:

$$(\beta^\star_{-k})^T X^T_{-k} X_{-k} \beta^\star_{-k} \leq (1 + \frac{1}{2}\sigma^2) n \cdot (\beta^\star_{-k})^T \Sigma_{S,-k} \beta^\star_{-k}.$$

By taking $c_x = \max\left\{1 + \frac{1}{2}\sigma^2, \frac{4}{c}\right\}$, the proof is complete. $\square$

## B.2 BLOCK DECOMPOSITION OF $X_{-k} X^T_{-k}$

Let $X_k = U \widetilde{M^{\frac{1}{2}}} V$, where $U \in \mathbb{R}^{n \times n}$ and $V \in \mathbb{R}^{d \times d}$ are orthogonal matrices representing the left and right singular vectors, respectively. The matrix $\widetilde{M^{\frac{1}{2}}}$ is defined as:

$$\widetilde{M^{\frac{1}{2}}} = \begin{pmatrix} m^{\frac{1}{2}}_1 & & \\ & \ddots & \\ & & m^{\frac{1}{2}}_k \\ \mathbf{0}^{(n-k) \times k} & & \end{pmatrix} \in \mathbb{R}^{n \times k}.$$

Therefore, we have $X_k X^T_k = UMU^T$, where $M = \text{diag}(m_1, ..., m_k, 0, ..., 0) \in \mathbb{R}^{n \times n}$. Similarly, $X^T_k X_k = V^T M_k V$, where $M_k = \text{diag}(m_1, ..., m_k) \in \mathbb{R}^{k \times k}$.

Let $\Delta = U^T X_{-k} X^T_{-k} U$, and write $\Delta$ in block matrix form as:

$$\Delta = \begin{pmatrix} \Delta_{11} & \Delta_{12} \\ \Delta^T_{12} & \Delta_{22} \end{pmatrix},$$

where $\Delta_{11} \in \mathbb{R}^{k \times k}$, $\Delta_{12} \in \mathbb{R}^{k \times (n-k)}$, and $\Delta_{22} \in \mathbb{R}^{(n-k) \times (n-k)}$.

We will repeatedly use the first $k$ rows of $(M + \lambda I_n + \Delta)^{-1}$, which we compute here. Because $M + \lambda I_n + \Delta$ and $\lambda I_{n-k} + \Delta_{22}$ are invertible when $A_k$ is positive definite, by block matrix inverse,

$$(M + \lambda I_n + \Delta)^{-1}[k, *]$$
$$= \left(M_k + \lambda I_k + \Delta_{11} - \Delta_{12}(\lambda I_{n-k} + \Delta_{22})^{-1} \Delta^T_{12}\right)^{-1} \left(I_k, -\Delta_{12}(\lambda I_{n-k} + \Delta_{22})^{-1}\right). \tag{5}$$

**Corollary 16** (Corollary of Lemma 10). There exists a constant depending only on $\sigma$, such that for any $n < \lambda^{-1}_{k+1}\left(\lambda + \sum_{j>k} \lambda_j\right)$, if the assumption condNum$(k, \delta, L)$ is satisfied, the following inequalities hold with probability at least $1 - \delta - c_x e^{-n/c_x}$, on the same event as in Lemma 10.

$$\|\Delta_{11}\|, \|\Delta_{12}\| \leq \|\Delta\| \leq c_x \left(\lambda + \sum_{j>k} \lambda_j\right).$$

$$\left\|(\lambda I_{n-k} + \Delta_{22})^{-1}\right\| \le \|\Delta^{-1}\| \le c_x L \left(\lambda + \sum_{j>k} \lambda_j\right)^{-1}.$$

$$\left\|\Delta_{12}(\lambda I_{n-k} + \Delta_{22})^{-2}\Delta_{12}^T\right\| \le c_x^4 L^2.$$

$$\left\|\Delta_{12}(\lambda I_{n-k} + \Delta_{22})^{-1}\Delta_{12}^T\right\| \le c_x^3 L \left(\lambda + \sum_{j>k} \lambda_j\right).$$

$$\left\|\Delta_{11} - \Delta_{12}(\lambda I_{n-k} + \Delta_{22})^{-1}\Delta_{12}^T\right\| \le c_x \left(\lambda + \sum_{j>k} \lambda_j\right).$$

*Proof.*    1. The first inequality.

$$\|\Delta_{11}\|, \|\Delta_{12}\| \le \|\Delta\| = \|X_{-k}X_{-k}^T\| \le \|A_k\| \le c_x \left(\lambda + \sum_{j>k} \lambda_j\right).$$

2. The second inequality.

$$\left\|(\lambda I_{n-k} + \Delta_{22})^{-1}\right\| \le \|(\lambda I_n + \Delta)^{-1}\| = \|A_k^{-1}\| \le c_x L \left(\lambda + \sum_{j>k} \lambda_j\right)^{-1},$$

where the first inequality holds because $\lambda I_n + \Delta$ is positive definite.

3. The third inequality.

$$\left\|\Delta_{12}(\lambda I_{n-k} + \Delta_{22})^{-2}\Delta_{12}^T\right\| \le \|\Delta_{12}\|^2\|(\lambda I_{n-k} + \Delta_{22})^{-1}\|^2 \le c_x^4 L^2.$$

4. The fourth inequality.

$$\left\|\Delta_{12}(\lambda I_{n-k} + \Delta_{22})^{-1}\Delta_{12}^T\right\| \le \|\Delta_{12}\|^2\|(\lambda I_{n-k} + \Delta_{22})^{-1}\| \le c_x^3 L \left(\lambda + \sum_{j>k} \lambda_j\right).$$

5. The last inequality.

$$\begin{aligned}
&\left\|\Delta_{11} - \Delta_{12}(\lambda I_{n-k} + \Delta_{22})^{-1}\Delta_{12}^T\right\| \\
&= \left\|\Delta_{11} + \lambda I_k - \Delta_{12}(\lambda I_{n-k} + \Delta_{22})^{-1}\Delta_{12}^T\right\| - \lambda \\
&\le \|\Delta_{11} + \lambda I_k\| - \lambda \\
&= \|\Delta_{11}\| \\
&\le c_x \left(\lambda + \sum_{j>k} \lambda_j\right).
\end{aligned}$$

The first inequality holds because $\Delta_{11} + \lambda I_k - \Delta_{12}(\lambda I_{n-k} + \Delta_{22})^{-1}\Delta_{12}^T$ is the Schur complement of the block $\Delta_{11} + \lambda I_k$ of the matrix $\Delta + \lambda I_n$, which is positive definite. Therefore, we have

$$\Delta_{11} + \lambda I_k \succcurlyeq \Delta_{11} + \lambda I_k - \Delta_{12}(\lambda I_{n-k} + \Delta_{22})^{-1}\Delta_{12}^T.$$

$\square$

**Lemma 17.** There exists a constant $c_x > 2$ depending only on $\sigma$, such that for any $N_1 < n < N_2$, if the assumption condNum$(k, \delta, L)$ is satisfied, the following holds with probability at least $1 - 2\delta - c_x e^{-n/c_x}$, on both events from Lemma 10 and Lemma 11,

$$\left\|\left[X_k^T X_k + \lambda I_k + V^T \left(\Delta_{11} - \Delta_{12}(\lambda I_{n-k} + \Delta_{22})^{-1}\Delta_{12}^T\right) V\right]^{-1} - \left(n\widetilde{\Sigma}_{S,k}\right)^{-1}\right\|$$

$$\leq \frac{c_x^2 \left( \sqrt{n(k + \ln \frac{1}{\delta})}\lambda_1 + c_x^2 L \left(\lambda + \sum_{j>k} \lambda_j\right) \right)}{(\lambda + n\lambda_k)^2}.$$

where

$$N_1 = \max \left\{ 4c_x^4 (k + \ln(1/\delta))\frac{\lambda_1^2}{\lambda_k^2}, 2c_x^4 L\lambda_k^{-1} \left( \lambda + \sum_{j>k} \lambda_j \right) \right\}.$$

$$N_2 = \frac{1}{\lambda_{k+1}} \left( \lambda + \sum_{j>k} \lambda_j \right).$$

*Proof.*

$$\left\| \left[ X_k^T X_k + \lambda I_k + V^T \left( \Delta_{11} - \Delta_{12}(\lambda I_{n-k} + \Delta_{22})^{-1}\Delta_{12}^T \right) V \right]^{-1} - \left( n\widetilde{\Sigma}_{S,k} \right)^{-1} \right\|$$

$$\leq \left\| \left[ X_k^T X_k + \lambda I_k + V^T \left( \Delta_{11} - \Delta_{12}(\lambda I_{n-k} + \Delta_{22})^{-1}\Delta_{12}^T \right) V \right]^{-1} \right\|$$

$$\cdot \left\| \left[ X_k^T X_k + \lambda I_k + V^T \left( \Delta_{11} - \Delta_{12}(\lambda I_{n-k} + \Delta_{22})^{-1}\Delta_{12}^T \right) V \right] - \left( n\widetilde{\Sigma}_{S,k} \right) \right\|$$

$$\cdot \left\| \left( n\widetilde{\Sigma}_{S,k} \right)^{-1} \right\|$$

$$= \frac{1}{\lambda + n\lambda_k} \left\| \left[ X_k^T X_k + \lambda I_k + V^T \left( \Delta_{11} - \Delta_{12}(\lambda I_{n-k} + \Delta_{22})^{-1}\Delta_{12}^T \right) V \right]^{-1} \right\|$$

$$\cdot \left\| X_k^T X_k - n\Sigma_{S,k} + V^T \left( \Delta_{11} - \Delta_{12}(\lambda I_{n-k} + \Delta_{22})^{-1}\Delta_{12}^T \right) V \right\|.$$

According to Lemma 11, Corollary 16, there exists a constant $c_x > 2$ depending only on $\sigma$, such that for any $k + \ln(1/\delta) < N_1 < n < N_2 = \lambda_{k+1}^{-1}(\lambda + \sum_{j>k} \lambda_j)$, with probability at least $1 - 2\delta - c_x e^{-n/c_x}$, on both events in Lemma 10 and Lemma 11,

$$\left\| \frac{1}{n} X_k^T X_k - \Sigma_{S,k} \right\| \leq c_x \lambda_1 \sqrt{\frac{k + \ln \frac{1}{\delta}}{n}}.$$

$$\left\| \Delta_{12}(\lambda I_{n-k} + \Delta_{22})^{-1}\Delta_{12}^T \right\| \leq c_x^3 L \left( \lambda + \sum_{j>k} \lambda_j \right).$$

1. $\left\| X_k^T X_k - n\Sigma_{S,k} + V^T \left( \Delta_{11} - \Delta_{12}(\lambda I_{n-k} + \Delta_{22})^{-1}\Delta_{12}^T \right) V \right\|.$

$$\left\| X_k^T X_k - n\Sigma_{S,k} + V^T \left( \Delta_{11} - \Delta_{12}(\lambda I_{n-k} + \Delta_{22})^{-1}\Delta_{12}^T \right) V \right\|$$

$$\leq \left\| X_k^T X_k - n\Sigma_{S,k} \right\| + \left\| \left( \Delta_{11} - \Delta_{12}(\lambda I_{n-k} + \Delta_{22})^{-1}\Delta_{12}^T \right) \right\|$$

$$\leq c_x \sqrt{n(k + \ln \frac{1}{\delta})}\lambda_1 + c_x^3 L \left( \lambda + \sum_{j>k} \lambda_j \right).$$

2. $\left\| \left[ X_k^T X_k + \lambda I_k + V^T \left( \Delta_{11} - \Delta_{12}(\lambda I_{n-k} + \Delta_{22})^{-1}\Delta_{12}^T \right) V \right]^{-1} \right\|$

$$\frac{1}{\lambda + n\lambda_k} \left\| X_k^T X_k - n\Sigma_{S,k} + V^T \left( \Delta_{11} - \Delta_{12}(\lambda I_{n-k} + \Delta_{22})^{-1}\Delta_{12}^T \right) V \right\|$$

$$\leq \frac{1}{\lambda + n\lambda_k} \left( c_x \sqrt{n(k + \ln \frac{1}{\delta})}\lambda_1 + c_x^3 L \left( \lambda + \sum_{j>k} \lambda_j \right) \right).$$

Since $n > 4c_x^4 (k + \ln(1/\delta))\frac{\lambda_1^2}{\lambda_k^2}$,

$$\frac{1}{\lambda + n\lambda_k} c_x \sqrt{n(k + \ln \frac{1}{\delta})}\lambda_1 \leq \frac{c_x \sqrt{n(k + \ln \frac{1}{\delta})}\lambda_1}{n\lambda_k}$$

$$= \frac{c_x \sqrt{(k + \ln \frac{1}{\delta})\lambda_1}}{\sqrt{n}\lambda_k}$$

$$< \frac{1}{2c_x}.$$

Since $n > 2c_x^4 L \lambda_k^{-1} \left( \lambda + \sum_{j>k} \lambda_j \right)$,

$$\frac{1}{\lambda + n\lambda_k} c_x^3 L \left( \lambda + \sum_{j>k} \lambda_j \right) \leq \frac{c_x^3 L \left( \lambda + \sum_{j>k} \lambda_j \right)}{n\lambda_k}$$

$$< \frac{1}{2c_x}.$$

Therefore, we have

$$\frac{1}{\lambda + n\lambda_k} \left\| X_k^T X_k - n\Sigma_{S,k} + V^T \left( \Delta_{11} - \Delta_{12}(\lambda I_{n-k} + \Delta_{22})^{-1}\Delta_{12}^T \right) V \right\| < \frac{1}{c_x}.$$

Now we derive the upper bound for our target.

$$\left\| \left[ X_k^T X_k + \lambda I_k + V^T \left( \Delta_{11} - \Delta_{12}(\lambda I_{n-k} + \Delta_{22})^{-1}\Delta_{12}^T \right) V \right]^{-1} \right\|$$

$$= \left\| \left[ n\widetilde{\Sigma}_{S,k} + X_k^T X_k - n\Sigma_{S,k} + V^T \left( \Delta_{11} - \Delta_{12}(\lambda I_{n-k} + \Delta_{22})^{-1}\Delta_{12}^T \right) V \right]^{-1} \right\|$$

$$\leq \left\| \left( n\widetilde{\Sigma}_{S,k} \right)^{-1} \right\| \left[ 1 - \left\| \left( n\widetilde{\Sigma}_{S,k} \right)^{-1} \right\| \right.$$

$$\left. \cdot \left\| \left[ X_k^T X_k + \lambda I_k + V^T \left( \Delta_{11} - \Delta_{12}(\lambda I_{n-k} + \Delta_{22})^{-1}\Delta_{12}^T \right) V \right]^{-1} \right\| \right]^{-1}$$

$$\leq \frac{1}{\lambda + n\lambda_k} \left( 1 - \frac{1}{c_x} \right)^{-1}$$

$$\leq \frac{c_x}{\lambda + n\lambda_k}.$$

The first inequality follows from the result $\|(A + T)^{-1}\| \leq \|A^{-1}\| \left( 1 - \|A^{-1}\|\|T\| \right)^{-1}$, provided that both $A$ and $A + T$ are invertible and $\|A^{-1}\|\|T\| < 1$ (see Lemma 3.1 in Wedin (1973)).

Combining the above two inequalities,

$$\left\| \left[ X_k^T X_k + \lambda I_k + V^T \left( \Delta_{11} - \Delta_{12}(\lambda I_{n-k} + \Delta_{22})^{-1}\Delta_{12}^T \right) V \right]^{-1} - \left( n\widetilde{\Sigma}_{S,k} \right)^{-1} \right\|$$

$$\leq \frac{1}{\lambda + n\lambda_k} \frac{c_x}{\lambda + n\lambda_k} \left( c_x \sqrt{n(k + \ln \frac{1}{\delta})\lambda_1} + c_x^3 L \left( \lambda + \sum_{j>k} \lambda_j \right) \right)$$

$$= \frac{c_x^2 \left( \sqrt{n(k + \ln \frac{1}{\delta})\lambda_1} + c_x^2 L \left( \lambda + \sum_{j>k} \lambda_j \right) \right)}{(\lambda + n\lambda_k)^2}.$$

$\square$

### B.3 BIAS VARIANCE DECOMPOSITION

We consider the expection of the excess risk $\mathcal{R}\left( \widehat{\beta}(Y) \right) = \mathcal{R}\left( \widehat{\beta}(X\beta^\star) + \widehat{\beta}(\epsilon) \right)$ with respect to the distribution of the noise $\epsilon$.

$$\mathbb{E}_\epsilon \left[ \mathcal{R}\left( \widehat{\beta}(Y) \right) \right] = \mathbb{E}_\epsilon \left[ \left( \widehat{\beta}(Y) - \beta^\star \right)^T \Sigma_T \left( \widehat{\beta}(Y) - \beta^\star \right) \right]$$

$$= \mathbb{E}_{\epsilon} \left[ \widehat{\beta}(\epsilon)^T \Sigma_T \widehat{\beta}(\epsilon) \right] + \left( \widehat{\beta}(X\beta^{\star}) - \beta^{\star} \right)^T \Sigma_T \left( \widehat{\beta}(X\beta^{\star}) - \beta^{\star} \right).$$

We decompose the expected excess risk into variance and bias terms.

$$V = \mathbb{E}_{\epsilon} \left[ \widehat{\beta}(\epsilon)^T \Sigma_T \widehat{\beta}(\epsilon) \right]$$

$$\leq 2\mathbb{E}_{\epsilon} \left[ \widehat{\beta}(\epsilon)_k^T \Sigma_{T,k} \widehat{\beta}(\epsilon)_k \right] + 2\mathbb{E}_{\epsilon} \left[ \widehat{\beta}(\epsilon)_{-k}^T \Sigma_{T,-k} \widehat{\beta}(\epsilon)_{-k} \right].$$

$$B = \left( \widehat{\beta}(X\beta^{\star}) - \beta^{\star} \right)^T \Sigma_T \left( \widehat{\beta}(X\beta^{\star}) - \beta^{\star} \right)$$

$$\leq 2 \left( \widehat{\beta}(X\beta^{\star})_k - \beta_k^{\star} \right)^T \Sigma_{T,k} \left( \widehat{\beta}(X\beta^{\star})_k - \beta_k^{\star} \right)$$

$$+ 2 \left( \widehat{\beta}(X\beta^{\star})_{-k} - \beta_{-k}^{\star} \right)^T \Sigma_{T,-k} \left( \widehat{\beta}(X\beta^{\star})_{-k} - \beta_{-k}^{\star} \right).$$

The inequalities follow from the result for a positive definite block quadratic form:

$$(x_1^T, x_2^T) \begin{pmatrix} A & B \\ B^T & D \end{pmatrix} \begin{pmatrix} x_1 \\ x_2 \end{pmatrix} = x_1^T A x_1 + 2x_1^T B x_2 + x_1^T D x_1,$$

where the positive definiteness implies $x_1^T A x_1 + x_1^T D x_1 \geq 2x_1^T B x_2$.

**Lemma 18.** There exists a constant $c_x > 2$ depending only on $\sigma$, such that for any $N_1 < n < N_2$, if the assumption condNum$(k, \delta, L)$ (Assumption 2) is satisfied, then with probability at least $1 - 2\delta - c_x e^{-n/c_x}$, the following inequalities hold simultaneously:

$$\mu_n(A_k) \geq \frac{1}{c_x L} \left( \lambda + \sum_{j > k} \lambda_j \right).$$

$$\mu_1(A_k) \leq c_x \left( \lambda + \sum_{j > k} \lambda_j \right).$$

$$\mu_1(X_{-k} X_{-k}^T) \leq c_x \left( n\lambda_{k+1} + \sum_{j > k} \lambda_j \right).$$

$$\left\| \frac{1}{n} X_k^T X_k - \Sigma_{S,k} \right\| \leq c_x \lambda_1 \sqrt{\frac{k + \ln \frac{1}{\delta}}{n}}.$$

$$\left\| (X_k^T X_k)^{\frac{1}{2}} - \sqrt{n} \Sigma_{S,k}^{\frac{1}{2}} \right\| \leq c_x \sqrt{k + \ln \frac{1}{\delta}} \lambda_1 \lambda_k^{-\frac{1}{2}}.$$

$$\mu_k \left( \Sigma_{S,k}^{-\frac{1}{2}} X_k^T X_k \Sigma_{S,k}^{-\frac{1}{2}} \right) \geq \frac{1}{c_x} n.$$

$$\mu_1 \left( \Sigma_{S,k}^{-\frac{1}{2}} X_k^T X_k \Sigma_{S,k}^{-\frac{1}{2}} \right) \leq c_x n.$$

$$\mu_1(X_k^T X_k) \leq c_x \lambda_1 n.$$

$$\mu_k(X_k^T X_k) \geq \frac{1}{c_x} \lambda_k n.$$

$$\text{tr} \left( X_{-k} \Sigma_{T,-k} X_{-k}^T \right) \leq c_x n \, \text{tr} \left( \Sigma_{S,-k}^{\frac{1}{2}} \Sigma_{T,-k} \Sigma_{S,-k}^{\frac{1}{2}} \right).$$

$$(\beta_{-k}^{\star})^T X_{-k}^T X_{-k} \beta_{-k}^{\star} \leq c_x n (\beta_{-k}^{\star})^T \Sigma_{S,-k} \beta_{-k}^{\star}.$$

$$\|\Delta_{11}\|, \|\Delta_{12}, \|\Delta\|\| \leq c_x \left( \lambda + \sum_{j > k} \lambda_j \right).$$

$$\|(\lambda I_{n-k} + \Delta_{22})^{-1}\|, \|\Delta^{-1}\| \leq c_x L \left( \lambda + \sum_{j > k} \lambda_j \right)^{-1}.$$

$$\left\|\Delta_{12}(\lambda I_{n-k} + \Delta_{22})^{-2}\Delta_{12}^T\right\| \le c_x^4 L^2.$$

$$\left\|\Delta_{12}(\lambda I_{n-k} + \Delta_{22})^{-1}\Delta_{12}^T\right\| \le c_x^3 L \left(\lambda + \sum_{j>k}\lambda_j\right).$$

$$\left\|\Delta_{11} - \Delta_{12}(\lambda I_{n-k} + \Delta_{22})^{-1}\Delta_{12}^T\right\| \le c_x \left(\lambda + \sum_{j>k}\lambda_j\right).$$

And,

$$\left\|\left[X_k^T X_k + \lambda I_k + V^T\left(\Delta_{11} - \Delta_{12}(\lambda I_{n-k} + \Delta_{22})^{-1}\Delta_{12}^T\right)V\right]^{-1} - \left(n\widetilde{\Sigma}_{S,k}\right)^{-1}\right\|$$

$$\le \frac{c_x^2\left(\sqrt{n(k + \ln\frac{1}{\delta})}\lambda_1 + c_x^2 L\left(\lambda + \sum_{j>k}\lambda_j\right)\right)}{(\lambda + n\lambda_k)^2}.$$

$N_1$ and $N_2$ are defined as follows:

$$N_1 = \max\left\{4c_x^4(k + \ln(1/\delta))\frac{\lambda_1^2}{\lambda_k^2}, 2c_x^4 L\lambda_k^{-1}\left(\lambda + \sum_{j>k}\lambda_j\right)\right\}.$$

$$N_2 = \frac{1}{\lambda_{k+1}}\left(\lambda + \sum_{j>k}\lambda_j\right).$$

*Proof.* The lemma is a direct corollary from Lemma 10, Lemma 11, Corollary 12, Lemma 13, Lemma 14, Lemma 15, Corollary 16, Lemma 17. □

### B.3.1 VARIANCE IN THE FIRST $k$ DIMENSIONS

**Lemma 19.** Under the same conditions as in Lemma 18, and on the same event, for any $N_1 < n < N_2$,

$$\mathbb{E}_{\boldsymbol{\epsilon}}\left[\widehat{\beta}(\boldsymbol{\epsilon})_k^T \Sigma_{T,k}\widehat{\beta}(\boldsymbol{\epsilon})_k\right] \le 16v^2(1 + c_x^4 L^2)\frac{1}{n}\operatorname{tr}\left[\Sigma_{S,k}^{-\frac{1}{2}}\Sigma_{T,k}\Sigma_{S,k}^{-\frac{1}{2}}\right],$$

where

$$N_1 = \max\left\{4c_x^4\left(k + \ln\frac{1}{\delta}\right)\lambda_1^4\lambda_k^{-4},\right.$$

$$2c_x^4 L\lambda_1\lambda_k^{-2}\left(\lambda + \sum_{j>k}\lambda_j\right),$$

$$4c_x^4\left(k + \ln\frac{1}{\delta}\right)\lambda_1^6\lambda_k^{-8}\|\Sigma_{T,k}\|^2 k^2\left(\operatorname{tr}\left[\Sigma_{S,k}^{-\frac{1}{2}}\Sigma_{T,k}\Sigma_{S,k}^{-\frac{1}{2}}\right]\right)^{-2},$$

$$\left.2c_x^4 L\lambda_1^2\lambda_k^{-4}\left(\lambda + \sum_{j>k}\lambda_j\right)\|\Sigma_{T,k}\|k\left(\operatorname{tr}\left[\Sigma_{S,k}^{-\frac{1}{2}}\Sigma_{T,k}\Sigma_{S,k}^{-\frac{1}{2}}\right]\right)^{-1}\right\},$$

$$N_2 = \frac{1}{\lambda_{k+1}}\left(\lambda + \sum_{j>k}\lambda_j\right).$$

*Proof.*

$$\mathbb{E}_{\boldsymbol{\epsilon}}\left[\widehat{\beta}(\boldsymbol{\epsilon})_k^T \Sigma_{T,k}\widehat{\beta}(\boldsymbol{\epsilon})_k\right]$$

$$= \mathbb{E}_{\boldsymbol{\epsilon}}\operatorname{tr}\left[\boldsymbol{\epsilon}\boldsymbol{\epsilon}^T(XX^T + \lambda I_n)^{-1}X_k\Sigma_{T,k}X_k^T(XX^T + \lambda I_n)^{-1}\right]$$

$$= v^2 \operatorname{tr}\left[(XX^T + \lambda I_n)^{-1} X_k \Sigma_{T,k} X_k^T (XX^T + \lambda I_n)^{-1}\right]$$

$$= v^2 \operatorname{tr}\left[(UMU^T + U\Delta U^T + \lambda I_n)^{-1} U\widetilde{M}^{\frac{1}{2}} V \Sigma_{T,k}\right.$$

$$\left. \cdot V^T \left(\widetilde{M}^{\frac{1}{2}}\right)^T U^T (UMU^T + U\Delta U^T + \lambda I_n)^{-1}\right]$$

$$= v^2 \operatorname{tr}\left[U(M + \Delta + \lambda I_n)^{-1} \widetilde{M}^{\frac{1}{2}} V \Sigma_{T,k} V^T \left(\widetilde{M}^{\frac{1}{2}}\right)^T (M + \Delta + \lambda I_n)^{-1} U^T\right]$$

$$= v^2 \operatorname{tr}\left[\left(\widetilde{M}^{\frac{1}{2}}\right)^T (M + \Delta + \lambda I_n)^{-1} (M + \Delta + \lambda I_n)^{-1} \widetilde{M}^{\frac{1}{2}} V \Sigma_{T,k} V^T\right]$$

$$= v^2 \operatorname{tr}\left[M_k^{\frac{1}{2}} \left(M_k + \lambda I_k + \Delta_{11} - \Delta_{12}(\lambda I_{n-k} + \Delta_{22})^{-1}\Delta_{12}^T\right)^{-1} \left(I_k, -\Delta_{12}(\lambda I_{n-k} + \Delta_{22})^{-1}\right)\right.$$

$$\left. \cdot \left(I_k, -\Delta_{12}(\lambda I_{n-k} + \Delta_{22})^{-1}\right)^T \left(M_k + \lambda I_k + \Delta_{11} - \Delta_{12}(\lambda I_{n-k} + \Delta_{22})^{-1}\Delta_{12}^T\right)^{-1} M_k^{\frac{1}{2}} \right.$$

$$\left. \cdot V\Sigma_{T,k}V^T\right]$$

$$= v^2 \operatorname{tr}\left[M_k^{\frac{1}{2}} \left(M_k + \lambda I_k + \Delta_{11} - \Delta_{12}(\lambda I_{n-k} + \Delta_{22})^{-1}\Delta_{12}^T\right)^{-1}\right.$$

$$\left. \cdot \left(I_k + \Delta_{12}(\lambda I_{n-k} + \Delta_{22})^{-2}\Delta_{12}^T\right) \left(M_k + \lambda I_k + \Delta_{11} - \Delta_{12}(\lambda I_{n-k} + \Delta_{22})^{-1}\Delta_{12}^T\right)^{-1} M_k^{\frac{1}{2}}\right.$$

$$\left. \cdot V\Sigma_{T,k}V^T\right]$$

$$= v^2 \operatorname{tr}\left[\left(I_k + \Delta_{12}(\lambda I_{n-k} + \Delta_{22})^{-2}\Delta_{12}^T\right) \left(M_k + \lambda I_k + \Delta_{11} - \Delta_{12}(\lambda I_{n-k} + \Delta_{22})^{-1}\Delta_{12}^T\right)^{-1}\right.$$

$$\left. \cdot M_k^{\frac{1}{2}} V\Sigma_{T,k}V^T M_k^{\frac{1}{2}} \left(M_k + \lambda I_k + \Delta_{11} - \Delta_{12}(\lambda I_{n-k} + \Delta_{22})^{-1}\Delta_{12}^T\right)^{-1}\right]$$

$$\leq v^2 \left\|I_k + \Delta_{12}(\lambda I_{n-k} + \Delta_{22})^{-2}\Delta_{12}^T\right\| \operatorname{tr}\left[\left(M_k + \lambda I_k + \Delta_{11} - \Delta_{12}(\lambda I_{n-k} + \Delta_{22})^{-1}\Delta_{12}^T\right)^{-1}\right.$$

$$\left. \cdot M_k^{\frac{1}{2}} V\Sigma_{T,k}V^T M_k^{\frac{1}{2}} \left(M_k + \lambda I_k + \Delta_{11} - \Delta_{12}(\lambda I_{n-k} + \Delta_{22})^{-1}\Delta_{12}^T\right)^{-1}\right]$$

$$\leq v^2 (1 + c_x^4 L^2) \operatorname{tr}\left[\left(M_k + \lambda I_k + \Delta_{11} - \Delta_{12}(\lambda I_{n-k} + \Delta_{22})^{-1}\Delta_{12}^T\right)^{-1}\right.$$

$$\left. \cdot M_k^{\frac{1}{2}} V\Sigma_{T,k}V^T M_k^{\frac{1}{2}} \left(M_k + \lambda I_k + \Delta_{11} - \Delta_{12}(\lambda I_{n-k} + \Delta_{22})^{-1}\Delta_{12}^T\right)^{-1}\right]$$

$$= v^2 (1 + c_x^4 L^2) \operatorname{tr}\left[\left(V^T \left(M_k + \lambda I_k + \Delta_{11} - \Delta_{12}(\lambda I_{n-k} + \Delta_{22})^{-1}\Delta_{12}^T\right) V\right)^{-1}\right.$$

$$\left. \cdot V^T M_k^{\frac{1}{2}} V \cdot \Sigma_{T,k} \cdot V^T M_k^{\frac{1}{2}} V \cdot \left(V^T \left(M_k + \lambda I_k + \Delta_{11} - \Delta_{12}(\lambda I_{n-k} + \Delta_{22})^{-1}\Delta_{12}^T\right) V\right)^{-1}\right]$$

$$= v^2 (1 + c_x^4 L^2) \operatorname{tr}\left[\left(X_k^T X_k + \lambda I_k + V^T \left(\Delta_{11} - \Delta_{12}(\lambda I_{n-k} + \Delta_{22})^{-1}\Delta_{12}^T\right) V\right)^{-1}\right.$$

$$\left. \cdot \left(X_k^T X_k\right)^{\frac{1}{2}} \Sigma_{T,k} \left(X_k^T X_k\right)^{\frac{1}{2}}\right.$$

$$\left. \cdot \left(X_k^T X_k + \lambda I_k + V^T \left(\Delta_{11} - \Delta_{12}(\lambda I_{n-k} + \Delta_{22})^{-1}\Delta_{12}^T\right) V\right)^{-1}\right].$$

The sixth equation follows from Equation 5. The first inequality follows from the result $\operatorname{tr}[AB] \leq \|A\| \operatorname{tr}[B]$ where the matrix $B$ is positive semi-definite.

We define two quantities that represent concentration error terms:

$$E_1 = \left\|\left[X_k^T X_k + \lambda I_k + V^T \left(\Delta_{11} - \Delta_{12}(\lambda I_{n-k} + \Delta_{22})^{-1}\Delta_{12}^T\right) V\right]^{-1} - \left(n\widetilde{\Sigma}_{S,k}\right)^{-1}\right\|.$$

$$E_2 = \left(X_k^T X_k\right)^{\frac{1}{2}} - (n\Sigma_{S,k})^{\frac{1}{2}}.$$

Since $n > 4c_x^4 \left(k + \ln \frac{1}{\delta}\right) \lambda_1^6 \lambda_k^{-8} \|\Sigma_{T,k}\|^2 k^2 \left(\operatorname{tr}\left[\Sigma_{S,k}^{-\frac{1}{2}} \Sigma_{T,k} \Sigma_{S,k}^{-\frac{1}{2}}\right]\right)^{-2}$,

and $n > 2c_x^4 L \left(\lambda + \sum_{j>k} \lambda_j\right) \lambda_1^2 \lambda_k^{-4} \|\Sigma_{T,k}\| k \left(\operatorname{tr}\left[\Sigma_{S,k}^{-\frac{1}{2}} \Sigma_{T,k} \Sigma_{S,k}^{-\frac{1}{2}}\right]\right)^{-1}$,

$$\|E_1\| \left\|n\widetilde{\Sigma}_{S,k}\right\| \left\|\left(n\widetilde{\Sigma}_{S,k}\right)^{-1}\right\| \left\|(n\Sigma_{S,k})^{\frac{1}{2}}\right\| \|\Sigma_{T,k}\| \left\|(n\Sigma_{S,k})^{\frac{1}{2}}\right\| \left\|\left(n\widetilde{\Sigma}_{S,k}\right)^{-1}\right\|$$

$$\leq \frac{c_x^2\left(\sqrt{n(k+\ln\frac{1}{\delta})}\lambda_1 + c_x^2 L\left(\lambda + \sum_{j>k}\lambda_j\right)\right)}{(\lambda + n\lambda_k)^2}(\lambda + n\lambda_1)\frac{n\lambda_1}{(\lambda + n\lambda_k)^2}\|\Sigma_{T,k}\|$$

$$\leq \frac{c_x^2\left(\sqrt{n(k+\ln\frac{1}{\delta})}\lambda_1 + c_x^2 L\left(\lambda + \sum_{j>k}\lambda_j\right)\right)}{n^2}\frac{\lambda_1^2}{\lambda_k^4}\|\Sigma_{T,k}\|$$

$$= \frac{c_x^2\sqrt{(k+\ln\frac{1}{\delta})}}{n\sqrt{n}}\frac{\lambda_1^3}{\lambda_k^4}\|\Sigma_{T,k}\| + \frac{c_x^4 L\left(\lambda + \sum_{j>k}\lambda_j\right)}{n^2}\frac{\lambda_1^2}{\lambda_k^4}\|\Sigma_{T,k}\|$$

$$< \frac{1}{2nk}\operatorname{tr}\left[\Sigma_{S,k}^{-\frac{1}{2}}\Sigma_{T,k}\Sigma_{S,k}^{-\frac{1}{2}}\right] + \frac{1}{2nk}\operatorname{tr}\left[\Sigma_{S,k}^{-\frac{1}{2}}\Sigma_{T,k}\Sigma_{S,k}^{-\frac{1}{2}}\right]$$

$$= \frac{1}{nk}\operatorname{tr}\left[\Sigma_{S,k}^{-\frac{1}{2}}\Sigma_{T,k}\Sigma_{S,k}^{-\frac{1}{2}}\right].$$

Since $n > 4c_x^4\left(k+\ln\frac{1}{\delta}\right)\lambda_1^4\lambda_k^{-4}$ and $n > 2c_x^4 L\left(\lambda + \sum_{j>k}\lambda_j\right)\lambda_1\lambda_k^{-2}$,

$$\|E_1\|\left\|n\widetilde{\Sigma}_{S,k}\right\|$$

$$\leq \frac{c_x^2\left(\sqrt{n(k+\ln\frac{1}{\delta})}\lambda_1 + c_x^2 L\left(\lambda + \sum_{j>k}\lambda_j\right)\right)}{(\lambda + n\lambda_k)^2}(\lambda + n\lambda_1)$$

$$\leq \frac{c_x^2\left(\sqrt{n(k+\ln\frac{1}{\delta})}\lambda_1 + c_x^2 L\left(\lambda + \sum_{j>k}\lambda_j\right)\right)}{n}\frac{\lambda_1}{\lambda_k^2} \tag{6}$$

$$= \frac{c_x^2\sqrt{(k+\ln\frac{1}{\delta})}}{\sqrt{n}}\frac{\lambda_1^2}{\lambda_k^2} + \frac{c_x^4 L\left(\lambda + \sum_{j>k}\lambda_j\right)}{n}\frac{\lambda_1}{\lambda_k^2}$$

$$< \frac{1}{2} + \frac{1}{2}$$

$$= 1.$$

Since $n > c_x^2\left(k+\ln\frac{1}{\delta}\right)\lambda_1^4\lambda_k^{-6}\|\Sigma_{T,k}\|^2 k^2\left(\operatorname{tr}\left[\Sigma_{S,k}^{-\frac{1}{2}}\Sigma_{T,k}\Sigma_{S,k}^{-\frac{1}{2}}\right]\right)^{-2}$,

$$\|E_2\|\left\|\left(n\widetilde{\Sigma}_{S,k}\right)^{-\frac{1}{2}}\right\|\left\|\left(n\widetilde{\Sigma}_{S,k}\right)^{-1}\right\|\left\|(n\Sigma_{S,k})^{\frac{1}{2}}\right\|\|\Sigma_{T,k}\|\left\|(n\Sigma_{S,k})^{\frac{1}{2}}\right\|\left\|\left(n\widetilde{\Sigma}_{S,k}\right)^{-1}\right\|$$

$$\leq c_x\sqrt{k+\ln\frac{1}{\delta}}\lambda_1\lambda_k^{-\frac{1}{2}}(n\lambda_k)^{-\frac{1}{2}}\frac{n\lambda_1}{(\lambda + n\lambda_k)^2}\|\Sigma_{T,k}\|$$

$$\leq \frac{c_x\sqrt{k+\ln\frac{1}{\delta}}}{n\sqrt{n}}\frac{\lambda_1^2}{\lambda_k^3}\|\Sigma_{T,k}\|$$

$$\leq \frac{1}{nk}\operatorname{tr}\left[\Sigma_{S,k}^{-\frac{1}{2}}\Sigma_{T,k}\Sigma_{S,k}^{-\frac{1}{2}}\right].$$

Since $n > c_x^2\left(k+\ln\frac{1}{\delta}\right)\lambda_1^2\lambda_k^{-2}$,

$$\|E_2\|\left\|\left(n\widetilde{\Sigma}_{S,k}\right)^{-\frac{1}{2}}\right\|$$

$$\leq c_x\sqrt{k+\ln\frac{1}{\delta}}\lambda_1\lambda_k^{-\frac{1}{2}}(n\lambda_k)^{-\frac{1}{2}} \tag{7}$$

$$= \frac{c_x\sqrt{k+\ln\frac{1}{\delta}}}{\sqrt{n}}\frac{\lambda_1}{\lambda_k}$$

$$< 1.$$

Combing the above four inequalities, we have

$$\operatorname{tr}\left[\left(X_k^T X_k + \lambda I_k + V^T\left(\Delta_{11} - \Delta_{12}(\lambda I_{n-k} + \Delta_{22})^{-1}\Delta_{12}^T\right)V\right)^{-1}\right.$$

$$
\cdot \left(X_k^T X_k\right)^{\frac{1}{2}} \Sigma_{T,k} \left(X_k^T X_k\right)^{\frac{1}{2}}
$$

$$
\cdot \left(X_k^T X_k + \lambda I_k + V^T \left(\Delta_{11} - \Delta_{12}(\lambda I_{n-k} + \Delta_{22})^{-1}\Delta_{12}^T\right) V\right)^{-1}\Big]
$$

$$
= \mathrm{tr}\left[\left(n\widetilde{\Sigma}_{S,k}\right)^{-1} (n\Sigma_{S,k})^{\frac{1}{2}} \Sigma_{T,k} (n\Sigma_{S,k})^{\frac{1}{2}} \left(n\widetilde{\Sigma}_{S,k}\right)^{-1}\right]
$$

$$
+ 2\,\mathrm{tr}\left[E_1 (n\Sigma_{S,k})^{\frac{1}{2}} \Sigma_{T,k} (n\Sigma_{S,k})^{\frac{1}{2}} \left(n\widetilde{\Sigma}_{S,k}\right)^{-1}\right]
$$

$$
+ 2\,\mathrm{tr}\left[\left(n\widetilde{\Sigma}_{S,k}\right)^{-1} E_2\Sigma_{T,k} (n\Sigma_{S,k})^{\frac{1}{2}} \left(n\widetilde{\Sigma}_{S,k}\right)^{-1}\right]
$$

$$
+ \mathrm{tr}\left[E_1 (n\Sigma_{S,k})^{\frac{1}{2}} \Sigma_{T,k} (n\Sigma_{S,k})^{\frac{1}{2}} E_1\right]
$$

$$
+ \mathrm{tr}\left[\left(n\widetilde{\Sigma}_{S,k}\right)^{-1} E_2\Sigma_{T,k}E_2 \left(n\widetilde{\Sigma}_{S,k}\right)^{-1}\right]
$$

$$
+ 2\,\mathrm{tr}\left[E_1 (n\Sigma_{S,k})^{\frac{1}{2}} \Sigma_{T,k}E_2 \left(n\widetilde{\Sigma}_{S,k}\right)^{-1}\right]
$$

$$
+ 2\,\mathrm{tr}\left[E_1 E_2\Sigma_{T,k} (n\Sigma_{S,k})^{\frac{1}{2}} \left(n\widetilde{\Sigma}_{S,k}\right)^{-1}\right]
$$

$$
+ 2\,\mathrm{tr}\left[E_1 E_2\Sigma_{T,k} (n\Sigma_{S,k})^{\frac{1}{2}} E_1\right]
$$

$$
+ 2\,\mathrm{tr}\left[E_1 E_2\Sigma_{T,k}E_2 \left(n\widetilde{\Sigma}_{S,k}\right)^{-1}\right]
$$

$$
+ \mathrm{tr}\left[E_1 E_2\Sigma_{T,k}E_2 E_1\right].
$$

In particular,

$$
\mathrm{tr}\left[\left(n\widetilde{\Sigma}_{S,k}\right)^{-1} (n\Sigma_{S,k})^{\frac{1}{2}} \Sigma_{T,k} (n\Sigma_{S,k})^{\frac{1}{2}} \left(n\widetilde{\Sigma}_{S,k}\right)^{-1}\right]
$$

$$
= \frac{1}{n}\,\mathrm{tr}\left[\widetilde{\Sigma}_{S,k}^{-1}\Sigma_{S,k}^{\frac{1}{2}}\Sigma_{T,k}\Sigma_{S,k}^{\frac{1}{2}}\widetilde{\Sigma}_{S,k}^{-1}\right]
$$

$$
\leq \frac{1}{n}\,\mathrm{tr}\left[\Sigma_{S,k}^{-1}\Sigma_{S,k}^{\frac{1}{2}}\Sigma_{T,k}\Sigma_{S,k}^{\frac{1}{2}}\Sigma_{S,k}^{-1}\right]
$$

$$
= \frac{1}{n}\,\mathrm{tr}\left[\Sigma_{S,k}^{-\frac{1}{2}}\Sigma_{T,k}\Sigma_{S,k}^{-\frac{1}{2}}\right].
$$

The inequality follows from the fact that $\mathrm{tr}[BAB] = \mathrm{tr}[A^{\frac{1}{2}}BA^{\frac{1}{2}}] \leq \mathrm{tr}[A^{\frac{1}{2}}CA^{\frac{1}{2}}] = \mathrm{tr}[CAC]$, where $A, B, C$ are positive semi-definite matrices, and $C \succcurlyeq B$, which implies that $A^{\frac{1}{2}}CA^{\frac{1}{2}} \succcurlyeq A^{\frac{1}{2}}BA^{\frac{1}{2}}$.

$$
\mathrm{tr}\left[E_1 E_2\Sigma_{T,k}E_2 E_1\right]
$$

$$
= \mathrm{tr}\left[E_1 n\widetilde{\Sigma}_{S,k}\left(n\widetilde{\Sigma}_{S,k}\right)^{-1} E_2 \left(n\widetilde{\Sigma}_{S,k}\right)^{-\frac{1}{2}} \left(n\widetilde{\Sigma}_{S,k}\right)^{\frac{1}{2}}\right.
$$

$$
\left. \cdot \Sigma_{T,k}E_2 \left(n\widetilde{\Sigma}_{S,k}\right)^{-\frac{1}{2}} \left(n\widetilde{\Sigma}_{S,k}\right)^{\frac{1}{2}} E_1 n\widetilde{\Sigma}_{S,k}\left(n\widetilde{\Sigma}_{S,k}\right)^{-1}\right]
$$

$$
\leq k \left(\|E_1\| \left\|n\widetilde{\Sigma}_{S,k}\right\|\right)^2 \left(\|E_2\| \left\|\left(n\widetilde{\Sigma}_{S,k}\right)^{-\frac{1}{2}}\right\|\right)
$$

$$
\cdot \|E_2\| \left\|\left(n\widetilde{\Sigma}_{S,k}\right)^{-\frac{1}{2}}\right\| \left\|\left(n\widetilde{\Sigma}_{S,k}\right)^{-1}\right\| \left\|(n\Sigma_{S,k})^{\frac{1}{2}}\right\| \|\Sigma_{T,k}\| \left\|(n\Sigma_{S,k})^{\frac{1}{2}}\right\| \left\|\left(n\widetilde{\Sigma}_{S,k}\right)^{-1}\right\|
$$

$$
\leq \frac{1}{n}\,\mathrm{tr}\left[\Sigma_{S,k}^{-\frac{1}{2}}\Sigma_{T,k}\Sigma_{S,k}^{-\frac{1}{2}}\right].
$$

The other terms can be similarly bounded. Therefore,

$$
\mathrm{tr}\left[\left(X_k^T X_k + \lambda I_k + V^T \left(\Delta_{11} - \Delta_{12}(\lambda I_{n-k} + \Delta_{22})^{-1}\Delta_{12}^T\right) V\right)^{-1}\right.
$$

$$\cdot \left( X_k^T X_k \right)^{\frac{1}{2}} \Sigma_{T,k} \left( X_k^T X_k \right)^{\frac{1}{2}}$$
$$\cdot \left( X_k^T X_k + \lambda I_k + V^T \left( \Delta_{11} - \Delta_{12}(\lambda I_{n-k} + \Delta_{22})^{-1}\Delta_{12}^T \right) V \right)^{-1} \Big]$$
$$\leq \frac{16}{n} \operatorname{tr} \left[ \Sigma_{S,k}^{-\frac{1}{2}} \Sigma_{T,k} \Sigma_{S,k}^{-\frac{1}{2}} \right].$$

The proof is complete by combing all the inequalities above. $\qquad\square$

### B.3.2 VARIANCE IN THE LAST $d - k$ DIMENSIONS

**Lemma 20.** Under the same conditions as in Lemma 18, and on the same event, for any $N_1 < n < N_2$,

$$\mathbb{E}_\epsilon \left[ \widehat{\beta}(\epsilon)_{-k}^T \Sigma_{T,-k} \widehat{\beta}(\epsilon)_{-k} \right] \leq v^2 c_x^3 L^2 n \left( \lambda + \sum_{j>k} \lambda_j \right)^{-2} \operatorname{tr} \left[ \Sigma_{S,-k}^{\frac{1}{2}} \Sigma_{T,-k} \Sigma_{S,-k}^{\frac{1}{2}} \right].$$

where $N_1, N_2$ are defined as in Lemma 18.

*Proof.*

$$\mathbb{E}_\epsilon \left[ \widehat{\beta}(\epsilon)_{-k}^T \Sigma_{T,-k} \widehat{\beta}(\epsilon)_{-k} \right]$$
$$= \mathbb{E}_\epsilon \operatorname{tr} \left[ \epsilon \epsilon^T (XX^T + \lambda I_n)^{-1} X_{-k} \Sigma_{T,-k} X_{-k}^T (XX^T + \lambda I_n)^{-1} \right]$$
$$= v^2 \operatorname{tr} \left[ (XX^T + \lambda I_n)^{-1} X_{-k} \Sigma_{T,-k} X_{-k}^T (XX^T + \lambda I_n)^{-1} \right]$$
$$\leq v^2 \left\| (XX^T + \lambda I_n)^{-2} \right\| \operatorname{tr} \left[ X_{-k} \Sigma_{T,-k} X_{-k}^T \right]$$
$$\leq v^2 \left\| (X_{-k} X_{-k}^T + \lambda I_n)^{-2} \right\| \operatorname{tr} \left[ X_{-k} \Sigma_{T,-k} X_{-k}^T \right]$$
$$\leq v^2 \left( \frac{1}{c_x L} \left( \lambda + \sum_{j>k} \lambda_j \right) \right)^{-2} c_x n \operatorname{tr} \left[ \Sigma_{S,-k}^{\frac{1}{2}} \Sigma_{T,-k} \Sigma_{S,-k}^{\frac{1}{2}} \right]$$
$$= v^2 c_x^3 L^2 n \left( \lambda + \sum_{j>k} \lambda_j \right)^{-2} \operatorname{tr} \left[ \Sigma_{S,-k}^{\frac{1}{2}} \Sigma_{T,-k} \Sigma_{S,-k}^{\frac{1}{2}} \right].$$

The first inequality follows from the result $\operatorname{tr}[ABA] = \operatorname{tr}[A^2 B] \leq \|A^2\| \operatorname{tr}[B]$ where the matrix $B$ is positive semi-definite. The second inequality follows from $XX^T + \lambda I_n \succcurlyeq X_{-k} X_{-k}^T + \lambda I_n$. $\quad\square$

### B.3.3 BIAS IN THE FIRST $k$ DIMENSIONS

The bias in the first $k$ dimensions can be decomposed into two terms.

$$\left( \widehat{\beta}(X\beta^\star)_k - \beta_k^\star \right)^T \Sigma_{T,k} \left( \widehat{\beta}(X\beta^\star)_k - \beta_k^\star \right)$$
$$= \left( X_k^T (XX^T + \lambda I_n)^{-1} X\beta^\star - \beta_k^\star \right)^T \Sigma_{T,k} \left( X_k^T (XX^T + \lambda I_n)^{-1} X\beta^\star - \beta_k^\star \right)$$
$$\leq 2 \left( X_k^T (XX^T + \lambda I_n)^{-1} X_k \beta_k^\star - \beta_k^\star \right)^T \Sigma_{T,k} \left( X_k^T (XX^T + \lambda I_n)^{-1} X_k \beta_k^\star - \beta_k^\star \right)$$
$$+ 2 \left( X_k^T (XX^T + \lambda I_n)^{-1} X_{-k} \beta_{-k}^\star \right)^T \Sigma_{T,k} \left( X_k^T (XX^T + \lambda I_n)^{-1} X_{-k} \beta_{-k}^\star \right).$$

The inequality follows from the result $x_1^T A x_1 + x_2^T A x_2 \geq 2 x_1^T A x_2$ where $A$ is positive semi-definite.

**Lemma 21.** Under the same conditions as in Lemma 18, and on the same event, for any $N_1 < n < N_2$,

$$\left( X_k^T (XX^T + \lambda I_n)^{-1} X_k \beta_k^\star - \beta_k^\star \right)^T \Sigma_{T,k} \left( X_k^T (XX^T + \lambda I_n)^{-1} X_k \beta_k^\star - \beta_k^\star \right)$$
$$\leq \frac{16 c_x^4}{n^2} \left( \lambda + \sum_{j>k} \lambda_j \right)^2 (\beta_k^\star)^T \Sigma_{S,k}^{-1} \beta_k^\star \left\| \Sigma_{S,k}^{-\frac{1}{2}} \Sigma_{T,k} \Sigma_{S,k}^{-\frac{1}{2}} \right\|.$$

where

$$N_1 = \max \left\{ 2c_x^3(\lambda + \sum_{j>k} \lambda_j)\lambda_1\lambda_k^{-2}, \right.$$
$$4c_x^4(k + \ln(1/\delta))\lambda_1^2\lambda_k^{-2},$$
$$\left. 2c_x^4 L\lambda_k^{-1}\left(\lambda + \sum_{j>k} \lambda_j\right) \right\},$$
$$N_2 = \frac{1}{\lambda_{k+1}}\left(\lambda + \sum_{j>k} \lambda_j\right).$$

*Proof.*

$$\left(X_k^T(XX^T + \lambda I_n)^{-1}X_k\beta_k^\star - \beta_k^\star\right)^T \Sigma_{T,k}\left(X_k^T(XX^T + \lambda I_n)^{-1}X_k\beta_k^\star - \beta_k^\star\right)$$
$$= (\beta_k^\star)^T\left(X_k^T(XX^T + \lambda I_n)^{-1}X_k - I_k\right)^T\Sigma_{T,k}\left(X_k^T(XX^T + \lambda I_n)^{-1}X_k - I_k\right)\beta_k^\star$$
$$= (\beta_k^\star)^T\Sigma_{S,k}^{-\frac{1}{2}}\left(\Sigma_{S,k}^{\frac{1}{2}}X_k^T(XX^T + \lambda I_n)^{-1}X_k\Sigma_{S,k}^{\frac{1}{2}} - \Sigma_{S,k}\right)^T\Sigma_{S,k}^{-\frac{1}{2}}$$
$$\cdot \Sigma_{T,k}\Sigma_{S,k}^{-\frac{1}{2}}\left(\Sigma_{S,k}^{\frac{1}{2}}X_k^T(XX^T + \lambda I_n)^{-1}X_k\Sigma_{S,k}^{\frac{1}{2}} - \Sigma_{S,k}\right)^T\Sigma_{S,k}^{-\frac{1}{2}}\beta_k^\star$$
$$\leq (\beta_k^\star)^T\Sigma_{S,k}^{-1}\beta_k^\star \cdot \left\|\Sigma_{S,k}^{-\frac{1}{2}}\Sigma_{T,k}\Sigma_{S,k}^{-\frac{1}{2}}\right\|$$
$$\cdot \left\|\Sigma_{S,k}^{\frac{1}{2}}X_k^T(XX^T + \lambda I_n)^{-1}X_k\Sigma_{S,k}^{\frac{1}{2}} - \Sigma_{S,k}\right\|^2.$$

Subsequently,

$$\left\|\Sigma_{S,k}^{\frac{1}{2}}X_k^T(XX^T + \lambda I_n)^{-1}X_k\Sigma_{S,k}^{\frac{1}{2}} - \Sigma_{S,k}\right\|$$
$$= \left\|\Sigma_{S,k}^{\frac{1}{2}}V^T\left(\widetilde{M}^{\frac{1}{2}}\right)^T U^T U(M + \lambda I_n + \Delta)^{-1}U^T U\widetilde{M}^{\frac{1}{2}}V\Sigma_{S,k}^{\frac{1}{2}} - \Sigma_{S,k}\right\|$$
$$= \left\|\Sigma_{S,k}^{\frac{1}{2}}V^T M_k^{\frac{1}{2}}\left(M_k + \lambda I_k + \Delta_{11} - \Delta_{12}(\lambda I_{n-k} + \Delta_{22})^{-1}\Delta_{12}^T\right)^{-1}M_k^{\frac{1}{2}}V\Sigma_{S,k}^{\frac{1}{2}} - \Sigma_{S,k}\right\|$$
$$= \left\|\Sigma_{S,k}^{\frac{1}{2}}\left(V^T M_k^{-\frac{1}{2}}\left(M_k + \lambda I_k + \Delta_{11} - \Delta_{12}(\lambda I_{n-k} + \Delta_{22})^{-1}\Delta_{12}^T\right)M_k^{-\frac{1}{2}}V\right)^{-1}\Sigma_{S,k}^{\frac{1}{2}}\right.$$
$$\left.-\Sigma_{S,k}\right\|$$
$$= \left\|\Sigma_{S,k}^{\frac{1}{2}}\left(I_k + V^T M_k^{-\frac{1}{2}}\left(\lambda I_k + \Delta_{11} - \Delta_{12}(\lambda I_{n-k} + \Delta_{22})^{-1}\Delta_{12}^T\right)M_k^{-\frac{1}{2}}V\right)^{-1}\Sigma_{S,k}^{\frac{1}{2}}\right.$$
$$\left.-\Sigma_{S,k}\right\|$$
$$= \left\|\Sigma_{S,k}^{\frac{1}{2}}\left(\left(I_k + V^T M_k^{-\frac{1}{2}}\left(\lambda I_k + \Delta_{11} - \Delta_{12}(\lambda I_{n-k} + \Delta_{22})^{-1}\Delta_{12}^T\right)M_k^{-\frac{1}{2}}V\right)^{-1} - I_k\right)\right.$$
$$\left.\cdot\Sigma_{S,k}^{\frac{1}{2}}\right\|$$
$$= \left\|\Sigma_{S,k}^{\frac{1}{2}}V^T M_k^{-\frac{1}{2}}\left(\lambda I_k + \Delta_{11} - \Delta_{12}(\lambda I_{n-k} + \Delta_{22})^{-1}\Delta_{12}^T\right)M_k^{-\frac{1}{2}}V\right.$$
$$\left.\cdot\left(I_k + V^T M_k^{-\frac{1}{2}}\left(\lambda I_k + \Delta_{11} - \Delta_{12}(\lambda I_{n-k} + \Delta_{22})^{-1}\Delta_{12}^T\right)M_k^{-\frac{1}{2}}V\right)^{-1}\Sigma_{S,k}^{\frac{1}{2}}\right\|$$
$$\leq \left\|\Sigma_{S,k}^{\frac{1}{2}}V^T M_k^{-\frac{1}{2}}\left(\lambda I_k + \Delta_{11} - \Delta_{12}(\lambda I_{n-k} + \Delta_{22})^{-1}\Delta_{12}^T\right)M_k^{-\frac{1}{2}}V\Sigma_{S,k}^{\frac{1}{2}}\right\|$$
$$+ \left\|\Sigma_{S,k}^{\frac{1}{2}}V^T M_k^{-\frac{1}{2}}\left(\lambda I_k + \Delta_{11} - \Delta_{12}(\lambda I_{n-k} + \Delta_{22})^{-1}\Delta_{12}^T\right)M_k^{-\frac{1}{2}}V\right.$$
$$\left.\cdot\left[\left(I_k + V^T M_k^{-\frac{1}{2}}\left(\lambda I_k + \Delta_{11} - \Delta_{12}(\lambda I_{n-k} + \Delta_{22})^{-1}\Delta_{12}^T\right)M_k^{-\frac{1}{2}}V\right)^{-1} - I_k\right]\Sigma_{S,k}^{\frac{1}{2}}\right\|.$$

The second equation follows from Equation 5.

We will derive upper bounds for both terms in the last equation above.

1. The first term.

$$\left\| \Sigma_{S,k}^{\frac{1}{2}} V^T M_k^{-\frac{1}{2}} \left( \lambda I_k + \Delta_{11} - \Delta_{12}(\lambda I_{n-k} + \Delta_{22})^{-1} \Delta_{12}^T \right) M_k^{-\frac{1}{2}} V \Sigma_{S,k}^{\frac{1}{2}} \right\|$$

$$\leq \left\| \lambda I_k + \Delta_{11} - \Delta_{12}(\lambda I_{n-k} + \Delta_{22})^{-1} \Delta_{12}^T \right\| \left\| \Sigma_{S,k}^{\frac{1}{2}} V^T M_k^{-1} V \Sigma_{S,k}^{\frac{1}{2}} \right\|$$

$$= \left\| \lambda I_k + \Delta_{11} - \Delta_{12}(\lambda I_{n-k} + \Delta_{22})^{-1} \Delta_{12}^T \right\| \left\| \left( \Sigma_{S,k}^{-\frac{1}{2}} \left( X_k^T X_k \right)^{-1} \Sigma_{S,k}^{-\frac{1}{2}} \right)^{-1} \right\|$$

$$\leq \left( \lambda + c_x \left( \lambda + \sum_{j>k} \lambda_j \right) \right) \frac{c_x}{n}$$

$$\leq \frac{2c_x^2}{n} \left( \lambda + \sum_{j>k} \lambda_j \right).$$

The inequality follows from $c_x > 2$.

2. The second term.

Since $n > 2c_x^3(\lambda + \sum_{j>k} \lambda_j)\lambda_k^{-1}$,

$$\left\| M_k^{-\frac{1}{2}} \left( \lambda I_k + \Delta_{11} - \Delta_{12}(\lambda I_{n-k} + \Delta_{22})^{-1} \Delta_{12}^T \right) M_k^{-\frac{1}{2}} \right\|$$

$$\leq \left\| M_k^{-1} \right\| \left\| \lambda I_k + \Delta_{11} - \Delta_{12}(\lambda I_{n-k} + \Delta_{22})^{-1} \Delta_{12}^T \right\|$$

$$\leq \frac{c_x}{n\lambda_k} \cdot 2c_x \left( \lambda + \sum_{j>k} \lambda_j \right)$$

$$< \frac{1}{c_x}.$$

Therefore,

$$\left\| \left( I_k + V^T M_k^{-\frac{1}{2}} \left( \lambda I_k + \Delta_{11} - \Delta_{12}(\lambda I_{n-k} + \Delta_{22})^{-1} \Delta_{12}^T \right) M_k^{-\frac{1}{2}} V \right)^{-1} - I_k \right\|$$

$$\leq \left\| \left( I_k + V^T M_k^{-\frac{1}{2}} \left( \lambda I_k + \Delta_{11} - \Delta_{12}(\lambda I_{n-k} + \Delta_{22})^{-1} \Delta_{12}^T \right) M_k^{-\frac{1}{2}} V \right)^{-1} \right\|$$

$$\cdot \left\| V^T M_k^{-\frac{1}{2}} \left( \lambda I_k + \Delta_{11} - \Delta_{12}(\lambda I_{n-k} + \Delta_{22})^{-1} \Delta_{12}^T \right) M_k^{-\frac{1}{2}} V \right\|$$

$$\leq \left( 1 - \frac{1}{c_x} \right)^{-1} \left\| V^T M_k^{-\frac{1}{2}} \left( \lambda I_k + \Delta_{11} - \Delta_{12}(\lambda I_{n-k} + \Delta_{22})^{-1} \Delta_{12}^T \right) M_k^{-\frac{1}{2}} V \right\|$$

$$\leq c_x \cdot \frac{c_x}{n\lambda_k} \cdot 2c_x \left( \lambda + \sum_{j>k} \lambda_j \right)$$

$$= \frac{2c_x^3}{n} \frac{\lambda + \sum_{j>k} \lambda_j}{\lambda_k}.$$

The second inequality follows from $\|(A + T)^{-1}\| \leq \|A^{-1}\| \left( 1 - \|A^{-1}\|\|T\| \right)^{-1}$, where both $A$ and $A + T$ are invertible and $\|A^{-1}\|\|T\| < 1$. Note that $c_x > 2$.

Since $n > 2c_x^3(\lambda + \sum_{j>k} \lambda_j)\lambda_1\lambda_k^{-2}$,

$$\left\| \Sigma_{S,k}^{\frac{1}{2}} V^T M_k^{-\frac{1}{2}} \left( \lambda I_k + \Delta_{11} - \Delta_{12}(\lambda I_{n-k} + \Delta_{22})^{-1} \Delta_{12}^T \right) M_k^{-\frac{1}{2}} V \right.$$

$$
\cdot \left[ \left( I_k + V^T M_k^{-\frac{1}{2}} \left( \lambda I_k + \Delta_{11} - \Delta_{12}(\lambda I_{n-k} + \Delta_{22})^{-1} \Delta_{12}^T \right) M_k^{-\frac{1}{2}} V \right)^{-1} - I_k \right] \Sigma_{S,k}^{\frac{1}{2}} \bigg\|
$$

$$
\leq \| \Sigma_{S,k} \| \, \| M_k^{-1} \| \, \| \lambda I_k + \Delta_{11} - \Delta_{12}(\lambda I_{n-k} + \Delta_{22})^{-1} \Delta_{12}^T \|
$$

$$
\cdot \left\| \left( I_k + V^T M_k^{-\frac{1}{2}} \left( \lambda I_k + \Delta_{11} - \Delta_{12}(\lambda I_{n-k} + \Delta_{22})^{-1} \Delta_{12}^T \right) M_k^{-\frac{1}{2}} V \right)^{-1} - I_k \right\|
$$

$$
\leq \lambda_1 \cdot \frac{c_x}{n\lambda_k} \cdot 2c_x \left( \lambda + \sum_{j>k} \lambda_j \right) \cdot \frac{2c_x^3}{n} \frac{\lambda + \sum_{j>k} \lambda_j}{\lambda_k}
$$

$$
= \frac{1}{n} \cdot \frac{4c_x^5}{n} \lambda_1 \lambda_k^{-2} \left( \lambda + \sum_{j>k} \lambda_j \right)^2
$$

$$
< \frac{2c_x^2}{n} \left( \lambda + \sum_{j>k} \lambda_j \right).
$$

Combining both terms above, we have

$$
\left\| \Sigma_{S,k}^{\frac{1}{2}} X_k^T (XX^T + \lambda I_n)^{-1} X_k \Sigma_{S,k}^{\frac{1}{2}} - \Sigma_{S,k} \right\| \leq \frac{4c_x^2}{n} \left( \lambda + \sum_{j>k} \lambda_j \right).
$$

Therefore,

$$
\left( X_k^T (XX^T + \lambda I_n)^{-1} X_k \beta_k^\star - \beta_k^\star \right)^T \Sigma_{T,k} \left( X_k^T (XX^T + \lambda I_n)^{-1} X_k \beta_k^\star - \beta_k^\star \right)
$$

$$
\leq (\beta_k^\star)^T \Sigma_{S,k}^{-1} \beta_k^\star \cdot \left\| \Sigma_{S,k}^{-\frac{1}{2}} \Sigma_{T,k} \Sigma_{S,k}^{-\frac{1}{2}} \right\|
$$

$$
\cdot \left\| \Sigma_{S,k}^{\frac{1}{2}} X_k^T (XX^T + \lambda I_n)^{-1} X_k \Sigma_{S,k}^{\frac{1}{2}} - \Sigma_{S,k} \right\|^2
$$

$$
\leq \frac{16c_x^4}{n^2} \left( \lambda + \sum_{j>k} \lambda_j \right)^2 (\beta_k^\star)^T \Sigma_{S,k}^{-1} \beta_k^\star \left\| \Sigma_{S,k}^{-\frac{1}{2}} \Sigma_{T,k} \Sigma_{S,k}^{-\frac{1}{2}} \right\|.
$$

$\square$

**Lemma 22.** Under the same conditions as in Lemma 18, and on the same event, for any $N_1 < n < N_2$,

$$
\left( X_k^T (XX^T + \lambda I_n)^{-1} X_{-k} \beta_{-k}^\star \right)^T \Sigma_{T,k} \left( X_k^T (XX^T + \lambda I_n)^{-1} X_{-k} \beta_{-k}^\star \right)
$$

$$
\leq 16c_x(1 + c_x^4 L^2) \left\| \Sigma_{S,k}^{-\frac{1}{2}} \Sigma_{T,k} \Sigma_{S,k}^{-\frac{1}{2}} \right\| (\beta_{-k}^\star)^T \Sigma_{S,-k} \beta_{-k}^\star.
$$

where

$$
N_1 = \max \left\{ 4c_x^4 \left( k + \ln \frac{1}{\delta} \right) \lambda_1^4 \lambda_k^{-4}, \right.
$$

$$
2c_x^4 L \lambda_1 \lambda_k^{-2} \left( \lambda + \sum_{j>k} \lambda_j \right),
$$

$$
4c_x^4 \left( k + \ln \frac{1}{\delta} \right) \lambda_1^6 \lambda_k^{-8} \| \Sigma_{T,k} \|^2 \left\| \Sigma_{S,k}^{-\frac{1}{2}} \Sigma_{T,k} \Sigma_{S,k}^{-\frac{1}{2}} \right\|^{-2},
$$

$$
\left. 2c_x^4 L \left( \lambda + \sum_{j>k} \lambda_j \right) \lambda_1^2 \lambda_k^{-4} \| \Sigma_{T,k} \| \left\| \Sigma_{S,k}^{-\frac{1}{2}} \Sigma_{T,k} \Sigma_{S,k}^{-\frac{1}{2}} \right\|^{-1} \right\},
$$

$$
N_2 = \frac{1}{\lambda_{k+1}} \left( \lambda + \sum_{j>k} \lambda_j \right).
$$

*Proof.*

$$\left(X_k^T(XX^T+\lambda I_n)^{-1}X_{-k}\beta_{-k}^\star\right)^T \Sigma_{T,k}\left(X_k^T(XX^T+\lambda I_n)^{-1}X_{-k}\beta_{-k}^\star\right)$$
$$\leq \left\|(XX^T+\lambda I_n)^{-1}X_k\Sigma_{T,k}X_k^T(XX^T+\lambda I_n)^{-1}\right\| \cdot (\beta_{-k}^\star)^T X_{-k}^T X_{-k}\beta_{-k}^\star.$$

From Lemma 18,

$$(\beta_{-k}^\star)^T X_{-k}^T X_{-k}\beta_{-k}^\star \leq c_x n(\beta_{-k}^\star)^T \Sigma_{S,-k}\beta_{-k}^\star.$$

In the following, we derive an upper bound for the other term.

$$\left\|(XX^T+\lambda I_n)^{-1}X_k\Sigma_{T,k}X_k^T(XX^T+\lambda I_n)^{-1}\right\|$$
$$= \left\|(M+\lambda I_n+\Delta)^{-1}\widetilde{M}^{\frac{1}{2}}V\Sigma_{T,k}V^T\left(\widetilde{M}^{\frac{1}{2}}\right)^T(M+\lambda I_n+\Delta)^{-1}\right\|$$
$$= \left\|\Sigma_{T,k}^{\frac{1}{2}}V^T\left(\widetilde{M}^{\frac{1}{2}}\right)^T(M+\lambda I_n+\Delta)^{-2}\widetilde{M}^{\frac{1}{2}}V\Sigma_{T,k}^{\frac{1}{2}}\right\|$$
$$= \left\|\Sigma_{T,k}^{\frac{1}{2}}V^T M_k^{\frac{1}{2}}\left(M_k+\lambda I_k+\Delta_{11}-\Delta_{12}(\lambda I_{n-k}+\Delta_{22})^{-1}\Delta_{12}^T\right)^{-1}\right.$$
$$\cdot\left(I_k+\Delta_{12}(\lambda I_{n-k}+\Delta_{22})^{-2}\Delta_{12}^T\right)\left(M_k+\lambda I_k+\Delta_{11}-\Delta_{12}(\lambda I_{n-k}+\Delta_{22})^{-1}\Delta_{12}^T\right)^{-1}$$
$$\left.\cdot M_k^{\frac{1}{2}}V\Sigma_{T,k}^{\frac{1}{2}}\right\|$$
$$\leq \left\|I_k+\Delta_{12}(\lambda I_{n-k}+\Delta_{22})^{-2}\Delta_{12}^T\right\|$$
$$\cdot\left\|\left(M_k+\lambda I_k+\Delta_{11}-\Delta_{12}(\lambda I_{n-k}+\Delta_{22})^{-1}\Delta_{12}^T\right)^{-1}M_k^{\frac{1}{2}}V\Sigma_{T,k}\right.$$
$$\left.\cdot V^T M_k^{\frac{1}{2}}\left(M_k+\lambda I_k+\Delta_{11}-\Delta_{12}(\lambda I_{n-k}+\Delta_{22})^{-1}\Delta_{12}^T\right)^{-1}\right\|$$
$$\leq (1+c_x^4L^2)\left\|\left(M_k+\lambda I_k+\Delta_{11}-\Delta_{12}(\lambda I_{n-k}+\Delta_{22})^{-1}\Delta_{12}^T\right)^{-1}M_k^{\frac{1}{2}}V\Sigma_{T,k}\right.$$
$$\left.\cdot V^T M_k^{\frac{1}{2}}\left(M_k+\lambda I_k+\Delta_{11}-\Delta_{12}(\lambda I_{n-k}+\Delta_{22})^{-1}\Delta_{12}^T\right)^{-1}\right\|$$
$$= (1+c_x^4L^2)\left\|\left(V^T\left(M_k+\lambda I_k+\Delta_{11}-\Delta_{12}(\lambda I_{n-k}+\Delta_{22})^{-1}\Delta_{12}^T\right)V\right)^{-1}\right.$$
$$\cdot V^T M_k^{\frac{1}{2}}V\Sigma_{T,k}V^T M_k^{\frac{1}{2}}V$$
$$\left.\cdot\left(V^T\left(M_k+\lambda I_k+\Delta_{11}-\Delta_{12}(\lambda I_{n-k}+\Delta_{22})^{-1}\Delta_{12}^T\right)V\right)^{-1}\right\|$$
$$= (1+c_x^4L^2)\left\|\left(X_k^T X_k+\lambda I_k+V^T(\Delta_{11}-\Delta_{12}(\lambda I_{n-k}+\Delta_{22})^{-1}\Delta_{12}^T)V\right)^{-1}\right.$$
$$\cdot\left(X_k^T X_k\right)^{\frac{1}{2}}\Sigma_{T,k}\left(X_k^T X_k\right)^{\frac{1}{2}}$$
$$\left.\cdot\left(X_k^T X_k+\lambda I_k+V^T(\Delta_{11}-\Delta_{12}(\lambda I_{n-k}+\Delta_{22})^{-1}\Delta_{12}^T)V\right)^{-1}\right\|.$$

The third equation follows from Equation 5.

We define two quantities that represent concentration error terms:

$$E_1 = \left\|\left[X_k^T X_k+\lambda I_k+V^T\left(\Delta_{11}-\Delta_{12}(\lambda I_{n-k}+\Delta_{22})^{-1}\Delta_{12}^T\right)V\right]^{-1}-\left(n\widetilde{\Sigma}_{S,k}\right)^{-1}\right\|.$$

$$E_2 = \left(X_k^T X_k\right)^{\frac{1}{2}}-(n\Sigma_{S,k})^{\frac{1}{2}}.$$

Since $n > 4c_x^4\left(k+\ln\frac{1}{\delta}\right)\lambda_1^6\lambda_k^{-8}\|\Sigma_{T,k}\|^2\left\|\Sigma_{S,k}^{-\frac{1}{2}}\Sigma_{T,k}\Sigma_{S,k}^{-\frac{1}{2}}\right\|^{-2}$,

and $n > 2c_x^4L\left(\lambda+\sum_{j>k}\lambda_j\right)\lambda_1^2\lambda_k^{-4}\|\Sigma_{T,k}\|\left\|\Sigma_{S,k}^{-\frac{1}{2}}\Sigma_{T,k}\Sigma_{S,k}^{-\frac{1}{2}}\right\|^{-1}$,

$$\|E_1\|\left\|n\widetilde{\Sigma}_{S,k}\right\|\left\|\left(n\widetilde{\Sigma}_{S,k}\right)^{-1}\right\|\left\|(n\Sigma_{S,k})^{\frac{1}{2}}\right\|\|\Sigma_{T,k}\|\left\|(n\Sigma_{S,k})^{\frac{1}{2}}\right\|\left\|\left(n\widetilde{\Sigma}_{S,k}\right)^{-1}\right\|$$

$$\leq \frac{c_x^2\left(\sqrt{n(k+\ln\frac{1}{\delta})}\lambda_1+c_x^2L\left(\lambda+\sum_{j>k}\lambda_j\right)\right)}{(\lambda+n\lambda_k)^2}(\lambda+n\lambda_1)\frac{n\lambda_1}{(\lambda+n\lambda_k)^2}\|\Sigma_{T,k}\|$$

$$\leq \frac{c_x^2 \left( \sqrt{n(k + \ln \frac{1}{\delta})} \lambda_1 + c_x^2 L \left( \lambda + \sum_{j>k} \lambda_j \right) \right) \frac{\lambda_1^2}{\lambda_k^4} \|\Sigma_{T,k}\|}{n^2}$$

$$= \frac{c_x^2 \sqrt{(k + \ln \frac{1}{\delta})}}{n\sqrt{n}} \frac{\lambda_1^3}{\lambda_k^4} \|\Sigma_{T,k}\| + \frac{c_x^4 L \left( \lambda + \sum_{j>k} \lambda_j \right)}{n^2} \frac{\lambda_1^2}{\lambda_k^4} \|\Sigma_{T,k}\|$$

$$< \frac{1}{2n} \left\| \Sigma_{S,k}^{-\frac{1}{2}} \Sigma_{T,k} \Sigma_{S,k}^{-\frac{1}{2}} \right\| + \frac{1}{2n} \left\| \Sigma_{S,k}^{-\frac{1}{2}} \Sigma_{T,k} \Sigma_{S,k}^{-\frac{1}{2}} \right\|$$

$$= \frac{1}{n} \left\| \Sigma_{S,k}^{-\frac{1}{2}} \Sigma_{T,k} \Sigma_{S,k}^{-\frac{1}{2}} \right\|.$$

Similar to Equation 6, since $n > 4c_x^4 \left( k + \ln \frac{1}{\delta} \right) \lambda_1^4 \lambda_k^{-4}$ and $n > 2c_x^4 L \left( \lambda + \sum_{j>k} \lambda_j \right) \lambda_1 \lambda_k^{-2}$,

$$\|E_1\| \left\| n\widetilde{\Sigma}_{S,k} \right\| < 1.$$

Since $n > c_x^2 \left( k + \ln \frac{1}{\delta} \right) \lambda_1^4 \lambda_k^{-6} \|\Sigma_{T,k}\|^2 \left\| \Sigma_{S,k}^{-\frac{1}{2}} \Sigma_{T,k} \Sigma_{S,k}^{-\frac{1}{2}} \right\|^{-2}$,

$$\|E_2\| \left\| \left( n\widetilde{\Sigma}_{S,k} \right)^{-\frac{1}{2}} \right\| \left\| \left( n\widetilde{\Sigma}_{S,k} \right)^{-1} \right\| \left\| (n\Sigma_{S,k})^{\frac{1}{2}} \right\| \|\Sigma_{T,k}\| \left\| (n\Sigma_{S,k})^{\frac{1}{2}} \right\| \left\| \left( n\widetilde{\Sigma}_{S,k} \right)^{-1} \right\|$$

$$\leq c_x \sqrt{k + \ln \frac{1}{\delta}} \lambda_1 \lambda_k^{-\frac{1}{2}} (n\lambda_k)^{-\frac{1}{2}} \frac{n\lambda_1}{(\lambda + n\lambda_k)^2} \|\Sigma_{T,k}\|$$

$$\leq \frac{c_x \sqrt{k + \ln \frac{1}{\delta}}}{n\sqrt{n}} \frac{\lambda_1^2}{\lambda_k^3} \|\Sigma_{T,k}\|$$

$$\leq \frac{1}{n} \left\| \Sigma_{S,k}^{-\frac{1}{2}} \Sigma_{T,k} \Sigma_{S,k}^{-\frac{1}{2}} \right\|.$$

Similar to Equation 7, since $n > c_x^2 \left( k + \ln \frac{1}{\delta} \right) \lambda_1^2 \lambda_k^{-2}$,

$$\|E_2\| \left\| \left( n\widetilde{\Sigma}_{S,k} \right)^{-\frac{1}{2}} \right\| < 1.$$

Combining the four inequalities above,

$$\left\| \left( X_k^T X_k + \lambda I_k + V^T (\Delta_{11} - \Delta_{12}(\lambda I_{n-k} + \Delta_{22})^{-1} \Delta_{12}^T) V \right)^{-1} \right.$$

$$\cdot \left( X_k^T X_k \right)^{\frac{1}{2}} \Sigma_{T,k} \left( X_k^T X_k \right)^{\frac{1}{2}}$$

$$\left. \cdot \left( X_k^T X_k + \lambda I_k + V^T (\Delta_{11} - \Delta_{12}(\lambda I_{n-k} + \Delta_{22})^{-1} \Delta_{12}^T) V \right)^{-1} \right\|$$

$$\leq \left\| \left( n\widetilde{\Sigma}_{S,k} \right)^{-1} (n\Sigma_{S,k})^{\frac{1}{2}} \Sigma_{T,k} (n\Sigma_{S,k})^{\frac{1}{2}} \left( n\widetilde{\Sigma}_{S,k} \right)^{-1} \right\|$$

$$+ 2 \left\| E_1 (n\Sigma_{S,k})^{\frac{1}{2}} \Sigma_{T,k} (n\Sigma_{S,k})^{\frac{1}{2}} \left( n\widetilde{\Sigma}_{S,k} \right)^{-1} \right\|$$

$$+ 2 \left\| \left( n\widetilde{\Sigma}_{S,k} \right)^{-1} E_2 \Sigma_{T,k} (n\Sigma_{S,k})^{\frac{1}{2}} \left( n\widetilde{\Sigma}_{S,k} \right)^{-1} \right\|$$

$$+ \left\| E_1 (n\Sigma_{S,k})^{\frac{1}{2}} \Sigma_{T,k} (n\Sigma_{S,k})^{\frac{1}{2}} E_1 \right\|$$

$$+ \left\| \left( n\widetilde{\Sigma}_{S,k} \right)^{-1} E_2 \Sigma_{T,k} E_2 \left( n\widetilde{\Sigma}_{S,k} \right)^{-1} \right\|$$

$$+ 2 \left\| E_1 (n\Sigma_{S,k})^{\frac{1}{2}} \Sigma_{T,k} E_2 \left( n\widetilde{\Sigma}_{S,k} \right)^{-1} \right\|$$

$$+ 2 \left\| E_1 E_2 \Sigma_{T,k} (n\Sigma_{S,k})^{\frac{1}{2}} \left( n\widetilde{\Sigma}_{S,k} \right)^{-1} \right\|$$

$$+ 2 \left\| E_1 E_2 \Sigma_{T,k} (n\Sigma_{S,k})^{\frac{1}{2}} E_1 \right\|$$

$$+ 2 \left\| E_1 E_2 \Sigma_{T,k} E_2 \left( n\widetilde{\Sigma}_{S,k} \right)^{-1} \right\|$$
$$+ \| E_1 E_2 \Sigma_{T,k} E_2 E_1 \|.$$

In particular,

$$\left\| \left( n\widetilde{\Sigma}_{S,k} \right)^{-1} (n\Sigma_{S,k})^{\frac{1}{2}} \Sigma_{T,k} (n\Sigma_{S,k})^{\frac{1}{2}} \left( n\widetilde{\Sigma}_{S,k} \right)^{-1} \right\|$$
$$= \frac{1}{n} \left\| \widetilde{\Sigma}_{S,k}^{-1} \Sigma_{S,k}^{\frac{1}{2}} \Sigma_{T,k} \Sigma_{S,k}^{\frac{1}{2}} \widetilde{\Sigma}_{S,k}^{-1} \right\|$$
$$\leq \frac{1}{n} \left\| \Sigma_{S,k}^{-1} \Sigma_{S,k}^{\frac{1}{2}} \Sigma_{T,k} \Sigma_{S,k}^{\frac{1}{2}} \Sigma_{S,k}^{-1} \right\|$$
$$= \frac{1}{n} \left\| \Sigma_{S,k}^{-\frac{1}{2}} \Sigma_{T,k} \Sigma_{S,k}^{-\frac{1}{2}} \right\|.$$

The inequality follows from the fact that $\|BAB\| = \|A^{\frac{1}{2}} B A^{\frac{1}{2}}\| \leq \|A^{\frac{1}{2}} C A^{\frac{1}{2}}\| = \|CAC\|$, where $A, B, C$ are positive semi-definite matrices, and $C \succcurlyeq B$, which implies that $A^{\frac{1}{2}} C A^{\frac{1}{2}} \succcurlyeq A^{\frac{1}{2}} B A^{\frac{1}{2}}$.

$$\|E_1 E_2 \Sigma_{T,k} E_2 E_1 \|$$
$$= \left\| E_1 n\widetilde{\Sigma}_{S,k} \left( n\widetilde{\Sigma}_{S,k} \right)^{-1} E_2 \left( n\widetilde{\Sigma}_{S,k} \right)^{-\frac{1}{2}} \left( n\widetilde{\Sigma}_{S,k} \right)^{\frac{1}{2}} \right.$$
$$\left. \cdot \Sigma_{T,k} E_2 \left( n\widetilde{\Sigma}_{S,k} \right)^{-\frac{1}{2}} \left( n\widetilde{\Sigma}_{S,k} \right)^{\frac{1}{2}} E_1 n\widetilde{\Sigma}_{S,k} \left( n\widetilde{\Sigma}_{S,k} \right)^{-1} \right\|$$
$$\leq \left( \|E_1\| \left\| n\widetilde{\Sigma}_{S,k} \right\| \right)^2 \left( \|E_2\| \left\| \left( n\widetilde{\Sigma}_{S,k} \right)^{-\frac{1}{2}} \right\| \right)$$
$$\cdot \|E_2\| \left\| \left( n\widetilde{\Sigma}_{S,k} \right)^{-\frac{1}{2}} \right\| \left\| \left( n\widetilde{\Sigma}_{S,k} \right)^{-1} \right\| \left\| (n\Sigma_{S,k})^{\frac{1}{2}} \right\| \|\Sigma_{T,k}\| \left\| (n\Sigma_{S,k})^{\frac{1}{2}} \right\| \left\| \left( n\widetilde{\Sigma}_{S,k} \right)^{-1} \right\|$$
$$\leq \frac{1}{n} \left\| \Sigma_{S,k}^{-\frac{1}{2}} \Sigma_{T,k} \Sigma_{S,k}^{-\frac{1}{2}} \right\|.$$

The other terms can be similarly bounded. Therefore,

$$\left\| \left( X_k^T X_k + \lambda I_k + V^T (\Delta_{11} - \Delta_{12}(\lambda I_{n-k} + \Delta_{22})^{-1} \Delta_{12}^T) V \right)^{-1} \right.$$
$$\cdot \left( X_k^T X_k \right)^{\frac{1}{2}} \Sigma_{T,k} \left( X_k^T X_k \right)^{\frac{1}{2}}$$
$$\left. \cdot \left( X_k^T X_k + \lambda I_k + V^T (\Delta_{11} - \Delta_{12}(\lambda I_{n-k} + \Delta_{22})^{-1} \Delta_{12}^T) V \right)^{-1} \right\|$$
$$\leq \frac{16}{n} \left\| \Sigma_{S,k}^{-\frac{1}{2}} \Sigma_{T,k} \Sigma_{S,k}^{-\frac{1}{2}} \right\|.$$

$\square$

### B.3.4 BIAS IN THE LAST $d - k$ DIMENSIONS

The upper bound for the bias in the last $d-k$ dimensions is extended from Tsigler & Bartlett (2023)'s Lemma 28. The bias can be decomposed into three terms.

$$\left( \widehat{\beta}(X\beta^\star)_{-k} - \beta_{-k}^\star \right)^T \Sigma_{T,-k} \left( \widehat{\beta}(X\beta^\star)_{-k} - \beta_{-k}^\star \right)$$
$$\leq 3(\beta_{-k}^\star)^T \Sigma_{T,-k} \beta_{-k}^\star$$
$$+ 3(\beta_{-k}^\star)^T X_{-k}^T (XX^T + \lambda I_n)^{-1} X_{-k} \Sigma_{T,-k} X_{-k}^T (XX^T + \lambda I_n)^{-1} X_{-k} \beta_{-k}^\star$$
$$+ 3(\beta_k^\star)^T X_k^T (XX^T + \lambda I_n)^{-1} X_{-k} \Sigma_{T,-k} X_{-k}^T (XX^T + \lambda I_n)^{-1} X_k \beta_k^\star.$$

**Lemma 23.** Under the same conditions as in Lemma 18, and on the same event, for any $N_1 < n < N_2$,

$$(\beta_{-k}^\star)^T X_{-k}^T (XX^T + \lambda I_n)^{-1} X_{-k} \Sigma_{T,-k} X_{-k}^T (XX^T + \lambda I_n)^{-1} X_{-k} \beta_{-k}^\star$$

$$\leq c_x^2 L \left(\lambda + \sum_j \lambda_j\right)^{-1} n\|\Sigma_{T,-k}\|(\beta_{-k}^\star)^T \Sigma_{S,-k}\beta_{-k}^\star.$$

where $N_1, N_2$ are defined as in Lemma 18.

*Proof.*

$$(\beta_{-k}^\star)^T X_{-k}^T (XX^T + \lambda I_n)^{-1} X_{-k} \Sigma_{T,-k} X_{-k}^T (XX^T + \lambda I_n)^{-1} X_{-k} \beta_{-k}^\star$$

$$\leq \|\Sigma_{T,-k}\|(\beta_{-k}^\star)^T X_{-k}^T (XX^T + \lambda I_n)^{-1} X_{-k} X_{-k}^T (XX^T + \lambda I_n)^{-1} X_{-k}\beta_{-k}^\star$$

$$\leq \|\Sigma_{T,-k}\|(\beta_{-k}^\star)^T X_{-k}^T (XX^T + \lambda I_n)^{-1} (XX^T + \lambda I_n)(XX^T + \lambda I_n)^{-1} X_{-k}\beta_{-k}^\star$$

$$\leq \|\Sigma_{T,-k}\| \left\|(XX^T + \lambda I_n)^{-1}\right\| (\beta_{-k}^\star)^T X_{-k}^T X_{-k}\beta_{-k}^\star$$

$$\leq \|\Sigma_{T,-k}\| \left\|(X_{-k} X_{-k}^T + \lambda I_n)^{-1}\right\| (\beta_{-k}^\star)^T X_{-k}^T X_{-k}\beta_{-k}^\star$$

$$\leq \|\Sigma_{T,-k}\| \left(\frac{1}{c_x L}\left(\lambda + \sum_j \lambda_j\right)\right)^{-1} c_x n(\beta_{-k}^\star)^T \Sigma_{S,-k}\beta_{-k}^\star$$

$$= c_x^2 L \left(\lambda + \sum_j \lambda_j\right)^{-1} n\|\Sigma_{T,-k}\|(\beta_{-k}^\star)^T \Sigma_{S,-k}\beta_{-k}^\star.$$

The fourth inequality follows from $XX^T + \lambda I_n \succcurlyeq X_{-k}X_{-k}^T + \lambda I_n$. $\square$

**Lemma 24.** Under the same conditions as in Lemma 18, and on the same event, for any $N_1 < n < N_2$,

$$(\beta_k^\star)^T X_k^T (XX^T + \lambda I_n)^{-1} X_{-k} \Sigma_{T,-k} X_{-k}^T (XX^T + \lambda I_n)^{-1} X_k \beta_k^\star$$

$$\leq \frac{c_x^6}{n} L\left(\lambda + \sum_{j>k} \lambda_j\right) \|\Sigma_{T,-k}\|(\beta_k^\star)^T \Sigma_{S,k}^{-1}\beta_k^\star.$$

where $N_1, N_2$ are defined as in Lemma 18.

*Proof.* It can be verified by Woodbury matrix identity that:

$$(XX^T + \lambda I_n)^{-1} X_k = (X_{-k}X_{-k}^T + \lambda I_n)^{-1} X_k \left(I_k + X_k^T (X_{-k}X_{-k}^T + \lambda I_n)^{-1} X_k\right)^{-1}.$$

Therefore,

$$(\beta_k^\star)^T X_k^T (XX^T + \lambda I_n)^{-1} X_{-k} \Sigma_{T,-k} X_{-k}^T (XX^T + \lambda I_n)^{-1} X_k \beta_k^\star$$

$$= \left\|\Sigma_{T,-k}^{\frac{1}{2}} X_{-k}^T (X_{-k}X_{-k}^T + \lambda I_n)^{-1} X_k \left(I_k + X_k^T (X_{-k}X_{-k}^T + \lambda I_n)^{-1} X_k\right)^{-1} \beta_k^\star\right\|^2$$

$$\leq \|\Sigma_{T,-k}\| \left\|(X_{-k}X_{-k}^T + \lambda I_n)^{-1} X_{-k} X_{-k}^T (X_{-k}X_{-k}^T + \lambda I_n)^{-1}\right\|$$

$$\cdot \left\|X_k \left(I_k + X_k^T (X_{-k}X_{-k}^T + \lambda I_n)^{-1} X_k\right)^{-1} \beta_k^\star\right\|^2$$

$$= \|\Sigma_{T,-k}\| \left\|(X_{-k}X_{-k}^T + \lambda I_n)^{-1} X_{-k} X_{-k}^T (X_{-k}X_{-k}^T + \lambda I_n)^{-1}\right\|$$

$$\cdot \left\|X_k \Sigma_{S,k}^{-\frac{1}{2}} \left(\Sigma_{S,k}^{-1} + \Sigma_{S,k}^{-\frac{1}{2}} X_k^T (X_{-k}X_{-k}^T + \lambda I_n)^{-1} X_k \Sigma_{S,k}^{-\frac{1}{2}}\right)^{-1} \Sigma_{S,k}^{\frac{1}{2}}\beta_k^\star\right\|^2$$

$$\leq \|\Sigma_{T,-k}\| \left\|(X_{-k}X_{-k}^T + \lambda I_n)^{-1}\right\| \left\|\Sigma_{S,k}^{-\frac{1}{2}} X_k^T X_k \Sigma_{S,k}^{-\frac{1}{2}}\right\|$$

$$\cdot \left\|\left(\Sigma_{S,k}^{-1} + \Sigma_{S,k}^{-\frac{1}{2}} X_k^T (X_{-k}X_{-k}^T + \lambda I_n)^{-1} X_k \Sigma_{S,k}^{-\frac{1}{2}}\right)^{-2}\right\| (\beta_k^\star)^T \Sigma_{S,k}^{-1}\beta_k^\star.$$

In particular,

$$\left\|\left(\Sigma_{S,k}^{-1} + \Sigma_{S,k}^{-\frac{1}{2}} X_k^T (X_{-k}X_{-k}^T + \lambda I_n)^{-1} X_k \Sigma_{S,k}^{-\frac{1}{2}}\right)^{-1}\right\|$$

$$\leq \left\| \left( \Sigma_{S,k}^{-\frac{1}{2}} X_k^T (X_{-k} X_{-k}^T + \lambda I_n)^{-1} X_k \Sigma_{S,k}^{-\frac{1}{2}} \right)^{-1} \right\|$$

$$\leq \left\| X_{-k} X_{-k}^T + \lambda I_n \right\| \left\| \left( \Sigma_{S,k}^{-\frac{1}{2}} X_k^T X_k \Sigma_{S,k}^{-\frac{1}{2}} \right)^{-1} \right\|$$

$$\leq c_x \left( \lambda + \sum_{j>k} \lambda_j \right) \frac{c_x}{n}$$

$$= \frac{c_x^2}{n} \left( \lambda + \sum_{j>k} \lambda_j \right).$$

The second inequality follows from $\mu_{\min}(ABA^T) \geq \mu_{\min}(B)\mu_{\min}(AA^T)$ where the matrix $B$ is positive definite.

Therefore,

$$(\beta_k^\star)^T X_k^T (XX^T + \lambda I_n)^{-1} X_{-k} \Sigma_{T,-k} X_{-k}^T (XX^T + \lambda I_n)^{-1} X_k \beta_k^\star$$

$$\leq \left\| \Sigma_{T,-k} \right\| \left\| (X_{-k} X_{-k}^T + \lambda I_n)^{-1} \right\| \left\| \Sigma_{S,k}^{-\frac{1}{2}} X_k^T X_k \Sigma_{S,k}^{-\frac{1}{2}} \right\|$$

$$\cdot \left\| \left( \Sigma_{S,k}^{-1} + \Sigma_{S,k}^{-\frac{1}{2}} X_k^T (X_{-k} X_{-k}^T + \lambda I_n)^{-1} X_k \Sigma_{S,k}^{-\frac{1}{2}} \right)^{-2} \right\| (\beta_k^\star)^T \Sigma_{S,k}^{-1} \beta_k^\star$$

$$\leq \left\| \Sigma_{T,-k} \right\| \cdot \left( \frac{1}{c_x L} \left( \lambda + \sum_{j>k} \lambda_j \right) \right)^{-1} \cdot c_x n \cdot \frac{c_x^4}{n^2} \left( \lambda + \sum_{j>k} \lambda_j \right)^2 \cdot (\beta_k^\star)^T \Sigma_{S,k}^{-1} \beta_k^\star$$

$$= \frac{c_x^6}{n} L \left( \lambda + \sum_{j>k} \lambda_j \right) \left\| \Sigma_{T,-k} \right\| (\beta_k^\star)^T \Sigma_{S,k}^{-1} \beta_k^\star.$$

$\square$

## B.4 MAIN RESULTS

**Theorem 25.** Let $\mathcal{T} = \Sigma_{S,k}^{-\frac{1}{2}} \Sigma_{T,k} \Sigma_{S,k}^{-\frac{1}{2}}$ and $\mathcal{U} = \Sigma_{S,-k}^{\frac{1}{2}} \Sigma_{T,-k} \Sigma_{S,-k}^{\frac{1}{2}}$. There exists a constant $c > 2$ depending only on $\sigma$, such that for any $cN < n < r_k$, if the assumption condNum$(k, \delta, L)$ (Assumption 2) is satisfied, then with probability at least $1 - 2\delta - ce^{-n/c}$,

$$\frac{V}{cv^2} \leq L^2 \frac{\operatorname{tr}[\mathcal{T}]}{n} + L^2 \frac{n \operatorname{tr}[\mathcal{U}]}{\left( \lambda + \sum_{j>k} \lambda_j \right)^2}.$$

$$\frac{B}{c} \leq \|\beta_k^\star\|_{\Sigma_{S,k}^{-1}}^2 \left( \frac{\lambda + \sum_{j>k} \lambda_j}{n} \right)^2 \left[ \|\mathcal{T}\| + L \frac{n\|\Sigma_{T,-k}\|}{\lambda + \sum_{j>k} \lambda_j} \right]$$

$$+ \|\beta_{-k}^\star\|_{\Sigma_{S,-k}}^2 \left[ L^2 \|\mathcal{T}\| + L \frac{n\|\Sigma_{T,-k}\|}{\lambda + \sum_{j>k} \lambda_j} \right].$$

$N$ is defined as follows:

$$N = \max \left\{ \left( k + \ln \frac{1}{\delta} \right) \lambda_1^6 \lambda_k^{-8} \|\Sigma_{T,k}\|^2 k^2 \left( \operatorname{tr}[\mathcal{T}] \right)^{-2}, \right.$$

$$\left. L \lambda_1^2 \lambda_k^{-4} \left( \lambda + \sum_{j>k} \lambda_j \right) \|\Sigma_{T,k}\| k \left( \operatorname{tr}[\mathcal{T}] \right)^{-1} \right\}.$$

**Remark 8** (Sample complexity). We have assumed $n > cN$ in the theorem. The first condition on $N$ indicates $n \gg k$. From the inequality $\lambda_k^2 \leq \|\Sigma_{T,k}\|^2 k^2 \left( \operatorname{tr}[\mathcal{T}] \right)^{-2} \leq k^2 \lambda_1^2$, it follows that $n = \Omega(k)$ in the best case, consistent with the sample complexity of classic linear regression.

This optimal case occurs when $\Sigma_{S,k} \approx \Sigma_{T,k}$. In the worst case, $n = \Omega(k^3)$ where covariate shift is significant in the first $k$ dimensions–e.g., when the test data lies predominantly in the subspace of the first dimension. This shift in sample complexity under varying degrees of covariate shift parallels the analysis of Ge et al. (2024) (see theire Theorem 4.2) for the under-parameterized setting. The second condition implies $n \gg \lambda + \sum_{j>k} \lambda_j$, such that the regularization is not too strong to introduce a bias greater than a constant (as shown in the first bias term). On the other hand, we assume $n < r_k$ in the theorem, which is consistent with the over-parameterized regime and Assumption 1, where the last $d - k$ components are considered to be essentially high-dimensional.

*Proof.* The theorem follows from Lemma 18, Lemma 19, Lemma 20, Lemma 21, Lemma 22, Lemma 23 and Lemma 24. For a constant $c'_x > 2$ depending only on $\sigma$, these lemmas hold for values of $n$ that satisfy the following inequalities:

$$n > 4c'^4_x (k + \ln(1/\delta))\lambda_1^2 \lambda_k^{-2},$$

$$n > 2c'^4_x L \lambda_k^{-1} \left( \lambda + \sum_{j>k} \lambda_j \right),$$

$$n > 4c'^4_x \left( k + \ln \frac{1}{\delta} \right) \lambda_1^4 \lambda_k^{-4},$$

$$n > 2c'^4_x L \lambda_1 \lambda_k^{-2} \left( \lambda + \sum_{j>k} \lambda_j \right),$$

$$n > 4c'^4_x \left( k + \ln \frac{1}{\delta} \right) \lambda_1^6 \lambda_k^{-8} \|\Sigma_{T,k}\|^2 k^2 \left( \text{tr} \left[ \Sigma_{S,k}^{-\frac{1}{2}} \Sigma_{T,k} \Sigma_{S,k}^{-\frac{1}{2}} \right] \right)^{-2},$$

$$n > 2c'^4_x L \lambda_1^2 \lambda_k^{-4} \left( \lambda + \sum_{j>k} \lambda_j \right) \|\Sigma_{T,k}\| k \left( \text{tr} \left[ \Sigma_{S,k}^{-\frac{1}{2}} \Sigma_{T,k} \Sigma_{S,k}^{-\frac{1}{2}} \right] \right)^{-1},$$

$$n > 2c'^3_x (\lambda + \sum_{j>k} \lambda_j)\lambda_1 \lambda_k^{-2},$$

$$n > 4c'^4_x \left( k + \ln \frac{1}{\delta} \right) \lambda_1^6 \lambda_k^{-8} \|\Sigma_{T,k}\|^2 \left\| \Sigma_{S,k}^{-\frac{1}{2}} \Sigma_{T,k} \Sigma_{S,k}^{-\frac{1}{2}} \right\|^{-2},$$

$$n > 2c'^4_x L \left( \lambda + \sum_{j>k} \lambda_j \right) \lambda_1^2 \lambda_k^{-4} \|\Sigma_{T,k}\| \left\| \Sigma_{S,k}^{-\frac{1}{2}} \Sigma_{T,k} \Sigma_{S,k}^{-\frac{1}{2}} \right\|^{-1},$$

$$n < \lambda_{k+1}^{-1} \left( \lambda + \sum_{j>k} \lambda_j \right).$$

A sufficient condition for all the inequalities above is given by $4c'^4_x N_1 < n < r_k$. This follows from the following facts:

$$\lambda_1 \lambda_k^{-1} \geq 1,$$
$$c'_x > 2,$$
$$L \geq 1,$$
$$k \left( \text{tr} \left[ \Sigma_{S,k}^{-\frac{1}{2}} \Sigma_{T,k} \Sigma_{S,k}^{-\frac{1}{2}} \right] \right)^{-1} \geq \left\| \Sigma_{S,k}^{-\frac{1}{2}} \Sigma_{T,k} \Sigma_{S,k}^{-\frac{1}{2}} \right\|^{-1},$$
$$k\|\Sigma_{T,k}\| \left( \text{tr} \left[ \Sigma_{S,k}^{-\frac{1}{2}} \Sigma_{T,k} \Sigma_{S,k}^{-\frac{1}{2}} \right] \right)^{-1} \geq \lambda_k.$$

Then, with probability at least $1 - 2\delta - c'_x e^{-n/c'_x}$:

$$V/2 \leq 16\upsilon^2 (1 + c'^4_x L^2) \frac{1}{n} \text{tr} \left[ \Sigma_{S,k}^{-\frac{1}{2}} \Sigma_{T,k} \Sigma_{S,k}^{-\frac{1}{2}} \right]$$

$$+ v^2 c_x'^3 L^2 n \left(\lambda + \sum_{j>k} \lambda_j\right)^{-2} \text{tr}\left[\Sigma_{S,-k}^{\frac{1}{2}} \Sigma_{T,-k} \Sigma_{S,-k}^{\frac{1}{2}}\right]$$

$$\leq 32 v^2 c_x'^4 L^2 \frac{1}{n} \text{tr}\left[\Sigma_{S,k}^{-\frac{1}{2}} \Sigma_{T,k} \Sigma_{S,k}^{-\frac{1}{2}}\right]$$

$$+ v^2 c_x'^3 L^2 n \left(\lambda + \sum_{j>k} \lambda_j\right)^{-2} \text{tr}\left[\Sigma_{S,-k}^{\frac{1}{2}} \Sigma_{T,-k} \Sigma_{S,-k}^{\frac{1}{2}}\right],$$

$$B/2 \leq \frac{16 c_x'^4}{n^2} \left(\lambda + \sum_{j>k} \lambda_j\right)^2 (\beta_k^\star)^T \Sigma_{S,k}^{-1} \beta_k^\star \left\|\Sigma_{S,k}^{-\frac{1}{2}} \Sigma_{T,k} \Sigma_{S,k}^{-\frac{1}{2}}\right\|$$

$$+ 32 c_x'(1 + c_x'^4 L^2) \left\|\Sigma_{S,k}^{-\frac{1}{2}} \Sigma_{T,k} \Sigma_{S,k}^{-\frac{1}{2}}\right\| (\beta_{-k}^\star)^T \Sigma_{S,-k} \beta_{-k}^\star$$

$$+ 3 c_x'^2 L \left(\lambda + \sum_j \lambda_j\right)^{-1} n \|\Sigma_{T,-k}\| (\beta_{-k}^\star)^T \Sigma_{S,-k} \beta_{-k}^\star$$

$$+ 3 \frac{c_x'^6}{n} L \left(\lambda + \sum_{j>k} \lambda_j\right) \|\Sigma_{T,-k}\| (\beta_k^\star)^T \Sigma_{S,k}^{-1} \beta_k^\star$$

$$+ 3 (\beta_{-k}^\star)^T \Sigma_{T,-k} \beta_{-k}^\star$$

$$\leq 16 c_x'^4 \frac{1}{n^2} \left(\lambda + \sum_{j>k} \lambda_j\right)^2 \left\|\Sigma_{S,k}^{-\frac{1}{2}} \Sigma_{T,k} \Sigma_{S,k}^{-\frac{1}{2}}\right\| (\beta_k^\star)^T \Sigma_{S,k}^{-1} \beta_k^\star$$

$$+ 64 c_x'^5 L^2 \left\|\Sigma_{S,k}^{-\frac{1}{2}} \Sigma_{T,k} \Sigma_{S,k}^{-\frac{1}{2}}\right\| (\beta_{-k}^\star)^T \Sigma_{S,-k} \beta_{-k}^\star$$

$$+ 3 c_x'^2 L n \left(\lambda + \sum_j \lambda_j\right)^{-1} \|\Sigma_{T,-k}\| (\beta_{-k}^\star)^T \Sigma_{S,-k} \beta_{-k}^\star$$

$$+ 3 c_x'^6 L \frac{1}{n} \left(\lambda + \sum_{j>k} \lambda_j\right) \|\Sigma_{T,-k}\| (\beta_k^\star)^T \Sigma_{S,k}^{-1} \beta_k^\star$$

$$+ 3 (\beta_{-k}^\star)^T \Sigma_{T,-k} \beta_{-k}^\star$$

$$\leq 16 c_x'^4 \frac{1}{n^2} \left(\lambda + \sum_{j>k} \lambda_j\right)^2 \left\|\Sigma_{S,k}^{-\frac{1}{2}} \Sigma_{T,k} \Sigma_{S,k}^{-\frac{1}{2}}\right\| (\beta_k^\star)^T \Sigma_{S,k}^{-1} \beta_k^\star$$

$$+ 64 c_x'^5 L^2 \left\|\Sigma_{S,k}^{-\frac{1}{2}} \Sigma_{T,k} \Sigma_{S,k}^{-\frac{1}{2}}\right\| (\beta_{-k}^\star)^T \Sigma_{S,-k} \beta_{-k}^\star$$

$$+ 3 c_x'^2 L n \left(\lambda + \sum_{j>k} \lambda_j\right)^{-1} \|\Sigma_{T,-k}\| (\beta_{-k}^\star)^T \Sigma_{S,-k} \beta_{-k}^\star$$

$$+ 3 c_x'^6 L \frac{1}{n} \left(\lambda + \sum_{j>k} \lambda_j\right) \|\Sigma_{T,-k}\| (\beta_k^\star)^T \Sigma_{S,k}^{-1} \beta_k^\star$$

$$+ 3 c_x'^5 L^2 \left\|\Sigma_{S,k}^{-\frac{1}{2}} \Sigma_{T,k} \Sigma_{S,k}^{-\frac{1}{2}}\right\| (\beta_{-k}^\star)^T \Sigma_{S,-k} \beta_{-k}^\star.$$

The last inequality follows from:

$$(\beta_{-k}^\star)^T \Sigma_{T,-k} \beta_{-k}^\star = (\beta_{-k}^\star)^T \Sigma_{S,-k}^{\frac{1}{2}} \Sigma_{S,-k}^{-\frac{1}{2}} \Sigma_{T,-k} \Sigma_{S,-k}^{-\frac{1}{2}} \Sigma_{S,-k}^{\frac{1}{2}} \beta_{-k}^\star$$

$$\leq \left\|\Sigma_{S,-k}^{-\frac{1}{2}} \Sigma_{T,-k} \Sigma_{S,-k}^{-\frac{1}{2}}\right\| (\beta_{-k}^\star)^T \Sigma_{S,-k} \beta_{-k}^\star.$$

By taking $c = 134c_x'^6$, the proof is complete. $\qquad\qquad\qquad\qquad\qquad\qquad\qquad\qquad\qquad$ $\square$

**Corollary 26** (Restatement of Theorem 2). Let $\mathcal{T} = \Sigma_{S,k}^{-\frac{1}{2}} \Sigma_{T,k} \Sigma_{S,k}^{-\frac{1}{2}}$, $\mathcal{U} = \Sigma_{S,-k} \Sigma_{T,-k}$ and $\mathcal{V} = \Sigma_{S,-k}^2$. There exists a constant $c > 2$ depending only on $\sigma, L$, such that for any $cN < n < r_k$, if the assumption condNum$(k, \delta, L)$ (Assumption 2) is satisfied, then with probability at least $1 - 3\delta$,

$$\frac{V}{cv^2} \leq \frac{k}{n} \frac{\operatorname{tr}[\mathcal{T}]}{k} + \frac{n}{R_k} \frac{\operatorname{tr}[\mathcal{U}]}{\operatorname{tr}[\mathcal{V}]}.$$

$$\frac{B}{c} \leq \left( \|\beta_k^\star\|_{\Sigma_{S,k}^{-1}}^2 \left( \frac{\lambda + \sum_{j>k} \lambda_j}{n} \right)^2 + \|\beta_{-k}^\star\|_{\Sigma_{S,-k}}^2 \right) \left[ \|\mathcal{T}\| + \frac{n}{r_k} \frac{\|\Sigma_{T,-k}\|}{\|\Sigma_{S,-k}\|} \right].$$

$N$ is a polynomial function of $k + \ln(1/\delta)$, $\lambda_1 \lambda_k^{-1}$, $1 + (\lambda + \sum_{j>k} \lambda_j) \lambda_k^{-1}$.

*Proof.* The first variance term follows directly from Theorem 25.

For the second variance term, by plugging in the definition of $R_k$,

$$L^2 \frac{n \operatorname{tr}[\mathcal{U}]}{(\lambda + \sum_{j>k} \lambda_j)^2} = L^2 \frac{n}{R_k} \frac{\operatorname{tr}[\Sigma_{S,-k} \Sigma_{T,-k}]}{\sum_{j>k} \lambda_j^2}$$

$$= L^2 \frac{n}{R_k} \frac{\operatorname{tr}[\mathcal{U}]}{\operatorname{tr}[\mathcal{V}]}.$$

For the first bias term, by plugging in the definition of $r_k$,

$$\|\beta_k^\star\|_{\Sigma_{S,k}^{-1}}^2 \left( \frac{\lambda + \sum_{j>k} \lambda_j}{n} \right)^2 \left[ \|\mathcal{T}\| + L \frac{n \|\Sigma_{T,-k}\|}{\lambda + \sum_{j>k} \lambda_j} \right]$$

$$= \|\beta_k^\star\|_{\Sigma_{S,k}^{-1}}^2 \left( \frac{\lambda + \sum_{j>k} \lambda_j}{n} \right)^2 \left[ \|\mathcal{T}\| + L \frac{n}{r_k} \frac{\|\Sigma_{T,-k}\|}{\lambda_{k+1}} \right].$$

Similarly, the second bias term can be transformed into:

$$\|\beta_{-k}^\star\|_{\Sigma_{S,-k}}^2 \left[ L^2 \|\mathcal{T}\| + L \frac{n \|\Sigma_{T,-k}\|}{\lambda + \sum_{j>k} \lambda_j} \right] = \|\beta_{-k}^\star\|_{\Sigma_{S,-k}}^2 \left[ L^2 \|\mathcal{T}\| + L \frac{n}{r_k} \frac{\|\Sigma_{T,-k}\|}{\lambda_{k+1}} \right].$$

Since the statement of Theorem 25 holds with probability at least $1 - 2\delta - ce^{-n/c}$, we only require $ce^{-n/c} < \delta$, which is equivalent as $n > c \ln c + c \ln(1/\delta)$. Combining the lower bounds of $n$ in Theorem 25, we should have:

$$n > \max \left\{ c \ln c + c \ln \frac{1}{\delta}, \right.$$

$$c\left(k + \ln \frac{1}{\delta}\right) \lambda_1^6 \lambda_k^{-8} \|\Sigma_{T,k}\|^2 k^2 \left(\operatorname{tr}[\mathcal{T}]\right)^{-2},$$

$$\left. cL\lambda_1^2 \lambda_k^{-4} \left(\lambda + \sum_{j>k} \lambda_j\right) \|\Sigma_{T,k}\| k \left(\operatorname{tr}[\mathcal{T}]\right)^{-1} \right\}.$$

For the first term in the maximum argument,

$$c \ln c + c \ln \frac{1}{\delta} \leq c^2 + c \ln \frac{1}{\delta}$$

$$\leq c^2 \left( k + \ln \frac{1}{\delta} \right).$$

The second term:

$$c\left(k + \ln \frac{1}{\delta}\right) \lambda_1^6 \lambda_k^{-8} \|\Sigma_{T,k}\|^2 k^2 \left(\operatorname{tr}[\mathcal{T}]\right)^{-2}$$

$$\leq c\left(k + \ln \frac{1}{\delta}\right) \lambda_1^6 \lambda_k^{-8} \|\Sigma_{T,k}\|^2 k^2 \left( \mu_k(\Sigma_{S,k}^{-1}) \operatorname{tr}[\Sigma_{T,k}] \right)^{-2}$$

$$\leq c\left(k + \ln \frac{1}{\delta}\right)\lambda_1^8 \lambda_k^{-8} \|\Sigma_{T,k}\|^2 k^2 \|\Sigma_{T,k}\|^{-2}$$

$$= c\left(k + \ln \frac{1}{\delta}\right)^3 \lambda_1^8 \lambda_k^{-8}.$$

The first inequality follows from $\mathrm{tr}[MN] \geq \mu_{\min}(M)\,\mathrm{tr}[N]$ for positive semi-definite matrices $M, N$.

Similar, for the third term:

$$cL\lambda_1^2 \lambda_k^{-4}\left(\lambda + \sum_{j>k} \lambda_j\right)\|\Sigma_{T,k}\| k \left(\mathrm{tr}\,[\mathcal{T}]\right)^{-1}$$

$$\leq cL\lambda_1^2 \lambda_k^{-4}\left(\lambda + \sum_{j>k} \lambda_j\right)\|\Sigma_{T,k}\| k \lambda_1 \|\Sigma_{T,k}\|^{-1}$$

$$\leq cL\left(k + \ln \frac{1}{\delta}\right)\lambda_1^3 \lambda_k^{-4}\left(\lambda + \sum_{j>k} \lambda_j\right).$$

The proof is complete by taking $c$ as $c^2 L^2$ and $N = \left(k + \ln \frac{1}{\delta}\right)^3 \left(\lambda_1 \lambda_k^{-1}\right)^8 \left[1 + \left(\lambda + \sum_{j>k} \lambda_j\right)\lambda_k^{-1}\right]$.

□

## C  LARGE SHIFT IN MINOR DIRECTIONS

In this section, we consider the scenario where the signal $\beta^\star$ mainly concentrate on the first $k$ components (here we choose the basis to be the eigenvectors of $\Sigma_S$), but the target covariance $\Sigma_T$ may not be small on the last $d - k$ components.

### C.1  LOWER BOUND FOR RIDGE REGRESSION

In this subsection, we will show that the original ridge regression algorithm will not work under this scenario.

Recall our model:

$$y = \beta^{\star T} x + \epsilon, \tag{8}$$

We can write our data as

$$Y = X\beta^\star + \boldsymbol{\epsilon}, \tag{9}$$

where $Y = (y_1, \cdots, y_n)^T \in \mathbb{R}^{n \times 1}$, $X = (x_1, \cdots, x_n)^T \in \mathbb{R}^{n \times d}$, $\boldsymbol{\epsilon} = (\epsilon_1, \cdots, \epsilon_n)^T \in \mathbb{R}^{n \times 1}$. We denote by $\widehat{\Sigma}_S := \frac{1}{n} X^T X$ the sample covariance matrix.

Assume the same assumptions as in our previous section still holds. We let $\Sigma_S = \mathbb{E}[x_i x_i^T]$ be the following: its eigenvalues $\lambda_1, \cdots, \lambda_d$ satisfies $\lambda_1 = \cdots = \lambda_k = 1$, $\lambda_{k+1} = \cdots = \lambda_{k+\lfloor\sqrt{n}/C_2\rfloor} = C_1/\sqrt{n}$ for sufficiently large constants $C_1, C_2$, and the remaining eigenvalues are all set to zero. We let $\Sigma_T = I_d$. Then the excess risk is $\mathbb{E}_\epsilon[(\widehat{\beta} - \beta^\star)^T \Sigma_T (\widehat{\beta} - \beta^\star)] = \mathbb{E}_\epsilon\|\widehat{\beta} - \beta^\star\|^2$. We will show that under this scenario, ridge regression can not obtain an error rate of $\mathcal{O}(\frac{1}{n})$. To see this, we explicitly write out the ridge solution:

$$\widehat{\beta} = (X^T X + \lambda I_d)^{-1} X^T Y$$

$$= \left(\widehat{\Sigma}_S + \frac{\lambda}{n} I_d\right)^{-1}\left(\frac{1}{n} X^T Y\right)$$

$$= \left(\widehat{\Sigma}_S + \frac{\lambda}{n} I_d\right)^{-1}\left(\frac{1}{n} X^T (X\beta^\star + \boldsymbol{\epsilon})\right)$$

$$= \left(\widehat{\Sigma}_S + \frac{\lambda}{n} I_d\right)^{-1}\left(\frac{1}{n} X^T X\beta^\star + \frac{1}{n} X^T \boldsymbol{\epsilon}\right)$$

$$= \left(\widehat{\Sigma}_S + \frac{\lambda}{n} I_d\right)^{-1}\left(\widehat{\Sigma}_S \beta^\star + \frac{1}{n} X^T \boldsymbol{\epsilon}\right)$$

$$= (\widehat{\Sigma}_S + \frac{\lambda}{n}I_d)^{-1}\widehat{\Sigma}_S\beta^\star + (\widehat{\Sigma}_S + \frac{\lambda}{n}I_d)^{-1}\frac{1}{n}X^T\boldsymbol{\epsilon}. \tag{10}$$

Therefore

$$\begin{aligned}
\widehat{\beta} - \beta^\star &= (\widehat{\Sigma}_S + \frac{\lambda}{n}I_d)^{-1}\widehat{\Sigma}_S\beta^\star - \beta^\star + (\widehat{\Sigma}_S + \frac{\lambda}{n}I_d)^{-1}\frac{1}{n}X^T\boldsymbol{\epsilon} \\
&= (\widehat{\Sigma}_S + \frac{\lambda}{n}I_d)^{-1}\widehat{\Sigma}_S\beta^\star - (\widehat{\Sigma}_S + \frac{\lambda}{n}I_d)^{-1}(\widehat{\Sigma}_S + \frac{\lambda}{n}I_d)\beta^\star + (\widehat{\Sigma}_S + \frac{\lambda}{n}I_d)^{-1}\frac{1}{n}X^T\boldsymbol{\epsilon} \\
&= -\frac{\lambda}{n}(\widehat{\Sigma}_S + \frac{\lambda}{n}I_d)^{-1}\beta^\star + (\widehat{\Sigma}_S + \frac{\lambda}{n}I_d)^{-1}\frac{1}{n}X^T\boldsymbol{\epsilon}
\end{aligned}$$

Taking expectation with respect to $\epsilon$,

$$\begin{aligned}
\mathbb{E}_\epsilon\|\widehat{\beta} - \beta^\star\|^2 &= \frac{\lambda^2}{n^2}\|(\widehat{\Sigma}_S + \frac{\lambda}{n}I_d)^{-1}\beta^\star\|^2 + \frac{1}{n^2}\operatorname{tr}(\boldsymbol{\epsilon}^T X(\widehat{\Sigma}_S + \frac{\lambda}{n}I_d)^{-2}X^T\boldsymbol{\epsilon}) \\
&= \frac{\lambda^2}{n^2}\|(\widehat{\Sigma}_S + \frac{\lambda}{n}I_d)^{-1}\beta^\star\|^2 + v^2\frac{1}{n}\operatorname{tr}((\widehat{\Sigma}_S + \frac{\lambda}{n}I_d)^{-2}\widehat{\Sigma}_S) \\
&:= B + V \tag{11}
\end{aligned}$$

where $B = \frac{\lambda^2}{n^2}\|(\widehat{\Sigma}_S + \frac{\lambda}{n}I_d)^{-1}\beta^\star\|^2$ is the bias, $V = \frac{v^2}{n}\operatorname{tr}((\widehat{\Sigma}_S + \frac{\lambda}{n}I_d)^{-2}\widehat{\Sigma}_S)$ is the variance. We state the formal version of Theorem 4 in the following:

**Theorem 27.** Under the instance we consider, namely $\lambda_1, \cdots, \lambda_d$ satisfies $\lambda_1 = \cdots = \lambda_k = 1$, $\lambda_{k+1} = \cdots = \lambda_{k+\lfloor\sqrt{n}/C_2\rfloor} = C_1/\sqrt{n}$, $\lambda_{k+\lfloor\sqrt{n}/C_2\rfloor+1} = \cdots = \lambda_d = 0$. WLOG assume $\sigma = 1$, $C_2 \geq C_1((\frac{C_1}{4C})^2 - k - \log\frac{1}{\delta})^{-1}$ for some absolute constant $C$, and $n \geq (\frac{3C_1}{2})^4$. With probability $1 - \delta$, when $\lambda = c\sqrt{n}$, we have $\frac{V}{v^2} \geq C'$, where $C' > 0$ is some absolute constant. When $\lambda \leq n^{3/4}$, we have $\frac{V}{v^2} \geq C'\frac{1}{\sqrt{n}}$. When $\lambda \geq n^{3/4}$, $B \geq \frac{\|\beta^\star\|^2}{9\sqrt{n}}$.

*Proof.* We will use the following concentration lemma modified from (Vershynin, 2018, Exercise 9.2.5):

**Lemma 28.** Let $\{x_i\}_{i=1}^n$ be i.i.d. $d-$dimensional random vectors, satisfying: $x_i$ is mean zero, $\mathbb{E}[xx^T] = \Sigma$ and is $\sigma^2\Sigma$-sub-gaussian, in the sense that

$$\mathbb{E}[\exp(v^T x_i)] \leq \exp\left(\frac{\|\sigma\Sigma^{1/2}v\|^2}{2}\right).$$

$X = (x_1, \cdots, x_n)^T \in \mathbb{R}^{n \times d}$. Then with probability $1 - \delta$,

$$\|\widehat{\Sigma} - \Sigma\| \leq C\sigma^4\left(\sqrt{\frac{r + \log\frac{1}{\delta}}{n}} + \frac{r + \log\frac{1}{\delta}}{n}\right)\|\Sigma\|$$

where $r := \operatorname{tr}(\Sigma)/\|\Sigma\|$ is the stable rank of $\Sigma$, $C$ is an absolute constant.

Applying Lemma 28, we have

$$\|\widehat{\Sigma}_S - \Sigma_S\| \leq C\left(\sqrt{\frac{r + \log\frac{1}{\delta}}{n}} + \frac{r + \log\frac{1}{\delta}}{n}\right)$$

where $r = \sum_{i=1}^d \lambda_i = k + \lfloor\sqrt{n}/C_2\rfloor\frac{C_1}{\sqrt{n}} \leq k + C_1/C_2$. When $n \geq C_1/C_2 + k + \log\frac{1}{\delta}$, we have

$$\|\widehat{\Sigma}_S - \Sigma_S\| \leq 2C\sqrt{\frac{C_1/C_2 + k + \log\frac{1}{\delta}}{n}}.$$

We denote by $\widehat{\lambda}_1 \geq \cdots \geq \widehat{\lambda}_d$ the eigenvalues of $\widehat{\Sigma}_S$. Then by Weyl's inequality (Chen et al., 2021, Lemma 2.2), $\|\widehat{\lambda}_i - \lambda_i\| \leq \|\widehat{\Sigma}_S - \Sigma_S\|$. Combining with previous inequalities, we have $1 -$

$2C\sqrt{\frac{C_1/C_2+k+\log\frac{1}{\delta}}{n}} \leq \widehat{\lambda}_i \leq 1+2C\sqrt{\frac{C_1/C_2+k+\log\frac{1}{\delta}}{n}}$ for $1 \leq i \leq k$, $\frac{C_1}{\sqrt{n}} - 2C\sqrt{\frac{C_1/C_2+k+\log\frac{1}{\delta}}{n}} \leq$ $\widehat{\lambda}_i \leq \frac{C_1}{\sqrt{n}} + 2C\sqrt{\frac{C_1/C_2+k+\log\frac{1}{\delta}}{n}}$ for $k+1 \leq i \leq k+\lfloor\sqrt{n}/C_2\rfloor$. If we take $C_2 \geq C_1((\frac{C_1}{4C})^2 - k - \log\frac{1}{\delta})^{-1}$ then $2C\sqrt{\frac{C_1/C_2+k+\log\frac{1}{\delta}}{n}} \leq \frac{C_1}{2\sqrt{n}}$. Therefore we have $\frac{C_1}{2\sqrt{n}} \leq \widehat{\lambda}_i \leq \frac{3C_1}{2\sqrt{n}}$ for $k+1 \leq i \leq k+\lfloor\sqrt{n}/C_2\rfloor$. When $\lambda = c\sqrt{n}$, we have

$$
\begin{aligned}
\frac{V}{v^2} &= \frac{1}{n}\operatorname{tr}((\widehat{\Sigma}_S + \frac{\lambda}{n}I_d)^{-2}\widehat{\Sigma}_S) \\
&= \frac{1}{n}\sum_{i=1}^{d}(\widehat{\lambda}_i + \frac{\lambda}{n})^{-2}\widehat{\lambda}_i \\
&\geq \frac{1}{n}\sum_{i=k+1}^{k+\lfloor\sqrt{n}/C_2\rfloor}(\widehat{\lambda}_i + \frac{\lambda}{n})^{-2}\widehat{\lambda}_i \\
&= \frac{1}{n}\sum_{i=k+1}^{k+\lfloor\sqrt{n}/C_2\rfloor}(\widehat{\lambda}_i + \frac{c}{\sqrt{n}})^{-2}\widehat{\lambda}_i \\
&\geq \frac{1}{n}\sum_{i=k+1}^{k+\lfloor\sqrt{n}/C_2\rfloor}(\frac{3C_1}{2\sqrt{n}} + \frac{c}{\sqrt{n}})^{-2}\frac{C_1}{2\sqrt{n}} \\
&= \frac{1}{n}\lfloor\sqrt{n}/C_2\rfloor\frac{C_1}{2}(\frac{3C_1}{2} + c)^{-2}\sqrt{n} \\
&\geq \frac{C_1}{4C_2}(\frac{3C_1}{2} + c)^{-2}.
\end{aligned}
\tag{12}
$$

Similarly, if $\lambda \leq n^{3/4}$,

$$
\begin{aligned}
\frac{V}{v^2} &\geq \frac{1}{n}\sum_{i=k+1}^{k+\lfloor\sqrt{n}/C_2\rfloor}(\widehat{\lambda}_i + \frac{\lambda}{n})^{-2}\widehat{\lambda}_i \\
&\geq \frac{1}{n}\sum_{i=k+1}^{k+\lfloor\sqrt{n}/C_2\rfloor}(\widehat{\lambda}_i + n^{-1/4})^{-2}\widehat{\lambda}_i \\
&\geq \frac{1}{n}\sum_{i=k+1}^{k+\lfloor\sqrt{n}/C_2\rfloor}(\frac{3C_1}{2\sqrt{n}} + n^{-1/4})^{-2}\frac{C_1}{2\sqrt{n}} \\
&= \frac{1}{n}\lfloor\sqrt{n}/C_2\rfloor\frac{C_1}{2}(\frac{3C_1}{2} + n^{1/4})^{-2}\sqrt{n} \\
&\geq \frac{C_1}{16C_2}n^{-1/2},
\end{aligned}
\tag{13}
$$

when $n \geq (\frac{3C_1}{2})^4$.

As for the bias term, assume $\lambda \geq n^{3/4}$. Using the same concentration argument, we have $2 > \widehat{\lambda}_i > 1/2$, for $1 \leq i \leq k$. When $\lambda \leq n$, $\lambda_{\max}(\widehat{\Sigma}_S + \frac{\lambda}{n}I_d) \leq 2 + \lambda/n \leq 3$, therefore $\lambda_{\min}((\widehat{\Sigma}_S + \frac{\lambda}{n}I_d)^{-1}) \geq \frac{1}{3}$. This implies

$$
\begin{aligned}
B &= \frac{\lambda^2}{n^2}\|(\widehat{\Sigma}_S + \frac{\lambda}{n}I_d)^{-1}\beta^\star\|^2 \\
&\geq \frac{n^{3/2}}{n^2}\|(\widehat{\Sigma}_S + \frac{\lambda}{n}I_d)^{-1}\beta^\star\|^2 \\
&\geq \frac{1}{\sqrt{n}}\lambda_{\min}^2((\widehat{\Sigma}_S + \frac{\lambda}{n}I_d)^{-1})\|\beta^\star\|^2 \\
&\geq \frac{\|\beta^\star\|^2}{9\sqrt{n}}.
\end{aligned}
$$

When $\lambda > n$, $\lambda_{\max}(\widehat{\Sigma}_S + \frac{\lambda}{n}I_d) \leq 2 + \lambda/n \leq \frac{3\lambda}{n}$, which means $\lambda_{\min}((\widehat{\Sigma}_S + \frac{\lambda}{n}I_d)^{-1}) \geq \frac{n}{3\lambda}$ This implies

$$
\begin{aligned}
B &= \frac{\lambda^2}{n^2}\|(\widehat{\Sigma}_S + \frac{\lambda}{n}I_d)^{-1}\beta^\star\|^2 \\
&\geq \frac{\lambda^2}{n^2}\lambda_{\min}^2((\widehat{\Sigma}_S + \frac{\lambda}{n}I_d)^{-1})\|\beta^\star\|^2 \\
&\geq \frac{\lambda^2}{n^2}\frac{n^2}{9\lambda^2}\|\beta^\star\|^2 \\
&\geq \frac{\|\beta^\star\|^2}{9}.
\end{aligned}
$$

$\square$

### C.2 UPPER BOUND FOR PCR

In this subsection, we will give the following upper bound for Principal Component Regression.

**Theorem 29.** When $n \gtrsim \sigma^8(r + \log\frac{1}{\delta})(\frac{\lambda_1}{\lambda_k - \lambda_{k+1}})^2\frac{\lambda_1^2 k^2\|\Sigma_T\|^2}{\lambda_k^4 \operatorname{tr}((\Sigma_{S,k})^{-1}\Sigma_{T,k})^2}$,

$$
\begin{aligned}
\mathbb{E}_\epsilon\|\widehat{\beta} - \beta^\star\|_{\Sigma_T}^2 &\leq \mathcal{O}(\sigma^8(\frac{\lambda_1}{\lambda_k - \lambda_{k+1}})^2(\frac{\lambda_1}{\lambda_k})^2\|\Sigma_T\|(\frac{r + \log\frac{1}{\delta}}{n})\|\beta_k^\star\|^2 + \frac{1}{n}v^2\operatorname{tr}((\Sigma_{S,k})^{-1}\Sigma_{T,k}) \\
&\quad + \frac{\|\Sigma_{T,k}\|\|\beta_{-k}^\star\|^2\|\Sigma_{S,-k}\|}{\lambda_k} + \beta_{-k}^{\star T}\Sigma_{T,-k}\beta_{-k}^\star)
\end{aligned}
$$

where $r = \frac{\sum_{i=1}^d \lambda_i}{\lambda_1}$.

*Proof.* For simplicity, we assume we have a sample size of $2n$, and in the first step we obtain an estimator $\widehat{U} \in \mathbb{R}^{d\times k}$ of the top-k subspace $U = \begin{pmatrix} I_k \\ 0 \end{pmatrix} \in \mathbb{R}^{d\times k}$, by using principal component analysis on the sample covariance matrix $\widehat{\Sigma}_S := \frac{1}{n}X^TX = \frac{1}{n}\sum_{i=1}^n x_i x_i^T$, namely $\widehat{U} = (\widehat{u}_1, \cdots, \widehat{u}_k)$ where $\widehat{u}_i$ is the $i$-th eigenvector of $\widehat{\Sigma}_S$. We denote the distance between the estimated subspace and the original one by $\Delta := \operatorname{dist}(U, \widehat{U}) = \|UU^T - \widehat{U}\widehat{U}^T\|$. For controlling $\Delta$, we have the following lemma (Lemma 6):

**Lemma 30.** With probability at least $1 - \delta$,

$$
\Delta \leq C\sigma^4\left(\sqrt{\frac{r + \log\frac{1}{\delta}}{n}} + \frac{r + \log\frac{1}{\delta}}{n}\right)\frac{\lambda_1}{\lambda_k - \lambda_{k+1}}
$$

where $r = \frac{\sum_{i=1}^n \lambda_i}{\lambda_1}$.

In the second step, we do linear regression on the projected (second half) data. With a little abuse of notation, we still use $X \in \mathbb{R}^{n\times d}$ to denote the data matrix indexed from $n+1$ to $2n$. The data here is independent from the data in step 1, and therefore independent of $\Delta$. If we let $Z := X\widehat{U} \in \mathbb{R}^{n\times k}$ be the projected data matrix, the estimator $\widehat{\beta}$ we obtained is given by

$$
\begin{aligned}
\widehat{\beta} &= \widehat{U}(Z^TZ)^{-1}Z^TY \\
&= \widehat{U}(\widehat{U}^TX^TX\widehat{U})^{-1}\widehat{U}^TX^TY. \tag{14}
\end{aligned}
$$

We aim to bound the excess risk on target, which is given by $\|\widehat{\beta} - \beta^\star\|_{\Sigma_T}^2 := \|\Sigma_T^{\frac{1}{2}}(\widehat{\beta} - \beta^\star)\|^2$. We introduce the following notations: suppose $\beta^\star = (\beta_1^\star, \cdots, \beta_d^\star)^T$. We let $\beta_U^\star := (\beta_1^\star, \cdots, \beta_k^\star, 0, \cdots, 0)^T$, $\beta_\perp^\star := (0, \cdots, 0, \beta_{k+1}^\star, \cdots, \beta_d^\star)^T = \beta^\star - \beta_U^\star$. Here we present an intermediate result for bounding the excess risk:

**Lemma 31.** Assume $\Delta \le \frac{\lambda_k^2 \operatorname{tr}((\Sigma_{S,k})^{-1}\Sigma_{T,k})}{4\lambda_1 k \|\Sigma_T\|}$. When $n \gtrsim \frac{\sigma^4 \lambda_1^2 \|\Sigma_T\|^2 k^3 \log(1/\delta)}{\lambda_k^4 \operatorname{tr}((\Sigma_{S,k})^{-1}\Sigma_{T,k})^2}$, then with probability $1 - \delta$,

$$\mathbb{E}_\epsilon \|\widehat{\beta} - \beta^\star\|_{\Sigma_T}^2 \le \mathcal{O}(\|\beta_U^\star\|^2 \Delta^2 (\frac{\lambda_1}{\lambda_k})^2 \|\Sigma_T\| + \frac{1}{n} v^2 \operatorname{tr}((\Sigma_{S,k})^{-1}\Sigma_{T,k})$$
$$+ \frac{\|\Sigma_{T,k}\| \|\beta_{-k}^\star\|^2 \|\Sigma_{S,-k}\|}{\lambda_k} + \beta_{-k}^{\star T} \Sigma_{T,-k} \beta_{-k}^\star)$$

If further $n \gtrsim \sigma^4 \Delta^{-2} k \log(1/\delta)$,

$$\mathbb{E}_\epsilon \|\widehat{\beta} - \beta^\star\|_{\Sigma_T}^2 \le \mathcal{O}(\|\beta_U^\star\|^2 (\Delta^4 (\frac{\lambda_1}{\lambda_k})^2 \|\Sigma_T\| + \Delta^2 \|\Sigma_{T,-k}\| + \Delta^3 \|\Sigma_T\|)$$
$$+ \frac{1}{n} v^2 \operatorname{tr}((\Sigma_{S,k})^{-1}\Sigma_{T,k}) + \frac{\|\Sigma_{T,k}\| \|\beta_{-k}^\star\|^2 \|\Sigma_{S,-k}\|}{\lambda_k} + \beta_{-k}^{\star T} \Sigma_{T,-k} \beta_{-k}^\star)$$

From Lemma 30, when $n \ge r + \log \frac{1}{\delta} = \frac{\sum_{i=1}^n \lambda_i}{\lambda_1} + \log \frac{1}{\delta}$, we have

$$\Delta \le 2C \frac{\lambda_1}{\lambda_k - \lambda_{k+1}} \sigma^4 \sqrt{\frac{r + \log \frac{1}{\delta}}{n}}$$

Therefore when $n \gtrsim (r + \log \frac{1}{\delta}) \sigma^8 (\frac{\lambda_1}{\lambda_k - \lambda_{k+1}})^2 \frac{\lambda_1^2 k^2 \|\Sigma_T\|^2}{\lambda_k^4 \operatorname{tr}((\Sigma_{S,k})^{-1}\Sigma_{T,k})^2}$, the assumption for $\Delta$ and $n$ in Lemma 31 will be both satisfied. We can thus apply Lemma 31 to get

$$\mathbb{E}_\epsilon \|\widehat{\beta} - \beta^\star\|_{\Sigma_T}^2 \le \mathcal{O}(\sigma^8 (\frac{\lambda_1}{\lambda_k - \lambda_{k+1}})^2 (\frac{\lambda_1}{\lambda_k})^2 \|\Sigma_T\| \frac{r + \log \frac{1}{\delta}}{n} \|\beta_U^\star\|^2 + \frac{1}{n} v^2 \operatorname{tr}((\Sigma_{S,k})^{-1}\Sigma_{T,k})$$
$$+ \frac{\|\Sigma_{T,k}\| \|\beta_{-k}^\star\|^2 \|\Sigma_{S,-k}\|}{\lambda_k} + \beta_{-k}^{\star T} \Sigma_{T,-k} \beta_{-k}^\star)$$

where $r = \frac{\sum_{i=1}^d \lambda_i}{\lambda_1}$.

$\square$

### C.3 PROOFS FOR LEMMA 31

In the following we will prove Lemma 31.

*Proof for Lemma 31.* The proof idea is similar to (Ge et al., 2023, Theorem 4.4) and (Tripuraneni et al., 2021b, Theorem 4).

We can decompose $\widehat{\beta} - \beta^\star$ as

$$\widehat{\beta} - \beta^\star = \widehat{U}(\widehat{U}^T X^T X \widehat{U})^{-1} \widehat{U}^T X^T Y - \beta^\star$$
$$= \widehat{U}(\widehat{U}^T X^T X \widehat{U})^{-1} \widehat{U}^T X^T (X\beta^\star + \epsilon) - \beta^\star$$
$$= \widehat{U}(\widehat{U}^T X^T X \widehat{U})^{-1} \widehat{U}^T X^T (X\beta_U^\star + X\beta_\perp^\star + \epsilon) - (\beta_U^\star + \beta_\perp^\star)$$
$$= A_1 + A_2 + A_3 - \beta_\perp^\star,$$

where $A_1 := \widehat{U}(\widehat{U}^T X^T X \widehat{U})^{-1} \widehat{U}^T X^T X \beta_U^\star - \beta_U^\star$, $A_2 := \widehat{U}(\widehat{U}^T X^T X \widehat{U})^{-1} \widehat{U}^T X^T X \beta_\perp^\star$, $A_3 := \widehat{U}(\widehat{U}^T X^T X \widehat{U})^{-1} \widehat{U}^T X^T \epsilon$. Therefore

$$\|\widehat{\beta} - \beta^\star\|_{\Sigma_T}^2 \le \|A_1\|_{\Sigma_T}^2 + \|A_2\|_{\Sigma_T}^2 + \|A_3\|_{\Sigma_T}^2 + \|\beta_\perp^\star\|_{\Sigma_T}^2 \tag{15}$$

We give three lemmas for bounding the related terms. The first lemma considers the bias term $A_1$:

**Lemma 32.** If $\Delta \le \frac{\lambda_k}{4\lambda_1}$ and $n \gtrsim \max\{\sigma^4 (\frac{\lambda_1}{\lambda_k})^2 k \log(1/\delta), \sigma^4 k \log(1/\delta)\}$, then with probability at least $1 - \delta$,

$$\|A_1\|_{\Sigma_T}^2 \leq \mathcal{O}(\|\beta_U^\star\|^2 \Delta^2 (\frac{\lambda_1}{\lambda_k})^2 \|\Sigma_T\|)$$

If we further have $n \gtrsim \sigma^4 \Delta^{-2} k \log(1/\delta)$, then with probability at least $1 - \delta$,

$$\|A_1\|_{\Sigma_T}^2 \leq \mathcal{O}(\|\beta_U^\star\|^2 (\Delta^4 (\frac{\lambda_1}{\lambda_k})^2 \|\Sigma_T\| + \Delta^2 \|\Sigma_{T,-k}\| + \Delta^3 \|\Sigma_T\|)) \leq \mathcal{O}(\|\beta_U^\star\|^2 \Delta^2 \|\Sigma_T\|)$$

The second lemma considers the variance term $A_3$:

**Lemma 33.** If $\Delta \leq \frac{\lambda_k^2 \operatorname{tr}((\Sigma_{S,k})^{-1} \Sigma_{T,k})}{4\lambda_1 k \|\Sigma_T\|}$ and $n \gtrsim \frac{\sigma^4 \|\Sigma_S\|^2 \|\Sigma_T\|^2 k^3 \log(1/\delta)}{\lambda_k^4 \operatorname{tr}((\Sigma_{S,k})^{-1} \Sigma_{T,k})^2}$, then with probability at least $1 - \delta$,

$$\mathbb{E}_\epsilon[\|A_3\|_{\Sigma_T}^2] \leq \mathcal{O}(\frac{1}{n} v^2 \operatorname{tr}((\Sigma_{S,k})^{-1} \Sigma_{T,k})).$$

For bounding $A_2$, we actually have a similar result to bounding $A_3$:

**Lemma 34.** If $n \gtrsim \sigma^4 (\frac{\lambda_1}{\lambda_k})^2 k \log(1/\delta)$ and $\Delta \leq \min\{\frac{\|\Sigma_{T,k}\|}{2\|\Sigma_T\|}, \frac{\lambda_k}{4\lambda_1}\}$, then with probability at least $1 - \delta$

$$\|A_2\|_{\Sigma_T}^2 \leq \mathcal{O}(\frac{\|\Sigma_{T,k}\| \|\beta_{-k}^\star\|^2 \|\Sigma_{S,-k}\|}{\lambda_k}) \tag{16}$$

By Lemma 32, 33, 34, together with the decomposition (15), we have with probability $1 - \delta$, when $n \gtrsim N_1$,

$$\mathbb{E}_\epsilon \|\widehat{\beta} - \beta^\star\|_{\Sigma_T}^2 \leq \mathcal{O}(\|\beta_U^\star\|^2 \Delta^2 (\frac{\lambda_1}{\lambda_k})^2 \|\Sigma_T\| + \frac{1}{n} v^2 \operatorname{tr}((\Sigma_{S,k})^{-1} \Sigma_{T,k}) \tag{17}$$

$$+ \frac{\|\Sigma_{T,k}\| \|\beta_{-k}^\star\|^2 \|\Sigma_{S,-k}\|}{\lambda_k} + \beta_{-k}^{\star T} \Sigma_{T,-k} \beta_{-k}^\star) \tag{18}$$

If further $n \gtrsim \sigma^4 \Delta^{-2} k \log(1/\delta)$,

$$\mathbb{E}_\epsilon \|\widehat{\beta} - \beta^\star\|_{\Sigma_T}^2 \leq \mathcal{O}(\|\beta_U^\star\|^2 (\Delta^4 (\frac{\lambda_1}{\lambda_k})^2 \|\Sigma_T\| + \Delta^2 \|\Sigma_{T,-k}\| + \Delta^3 \|\Sigma_T\|) \tag{19}$$

$$+ \frac{1}{n} v^2 \operatorname{tr}((\Sigma_{S,k})^{-1} \Sigma_{T,k}) + \frac{\|\Sigma_{T,k}\| \|\beta_{-k}^\star\|^2 \|\Sigma_{S,-k}\|}{\lambda_k} + \beta_{-k}^{\star T} \Sigma_{T,-k} \beta_{-k}^\star) \tag{20}$$

$\square$

## C.4 TECHNICAL PROOFS

In the sequel, we give the proofs of Lemma 32, 33, 34 and 30. We first prove some additional technical lemmas. The following lemma, which is a simple corollary of (Tripuraneni et al., 2021b, Lemma 20), shows the concentration property of empirical covariance matrix.

**Lemma 35.** Let $\{x_i\}_{i=1}^n$ be i.i.d. $d-$dimensional random vectors, satisfying: $x_i$ is mean zero, $\mathbb{E}[xx^T] = \Sigma$ such that $\sigma_{\max}(\Sigma) \leq C_{\max}$ and is $\sigma^2 \Sigma$-sub-gaussian, in the sense that

$$\mathbb{E}[\exp(v^T x_i)] \leq \exp\left(\frac{\|\sigma \Sigma^{1/2} v\|^2}{2}\right).$$

$X = (x_1, \cdots, x_n)^T \in \mathbb{R}^{n \times d}$. Then for any $A, B \in \mathbb{R}^{d \times k}$, we have with probability at least $1 - \delta$

$$\|A^T (\frac{X^T X}{n}) B - A^T \Sigma B\|_2 \leq \mathcal{O}(\sigma^2 \|A\| \|B\| \|\Sigma\| (\sqrt{\frac{k}{n}} + \frac{k}{n} + \sqrt{\frac{\log(1/\delta)}{n}} + \frac{\log(1/\delta)}{n}). \tag{21}$$

*Proof.* We write the SVD of $A$ and $B$: $A = U_1\Lambda_1 V_1^T$, $B = U_2\Lambda_2 V_2^T$, where $U_1, U_2 \in \mathbb{R}^{d\times k}$, $\Lambda_1, \Lambda_2, V_1, V_2 \in \mathbb{R}^{k\times k}$. Then

$$\|A^T(\frac{X^T X}{n})B - A^T\Sigma B\|_2 = \|V_1\Lambda_1 U_1^T(\frac{X^T X}{n})U_2\Lambda_2 V_2^T - V_1\Lambda_1 U_1^T\Sigma U_2\Lambda_2 V_2^T\|_2$$

$$\leq \|V_1\Lambda_1\|\|U_1^T(\frac{X^T X}{n})U_2 - U_1^T\Sigma U_2\|\|\Lambda_2 V_2^T\|$$

$$\leq \|A\|\|B\|\|U_1^T(\frac{X^T X}{n})U_2 - U_1^T\Sigma U_2\|. \tag{22}$$

Now since $U_1, U_2 \in \mathbb{R}^{d\times k}$ are projection matrices, we can apply Tripuraneni et al. (2021b) Lemma 20, therefore

$$\|U_1^T(\frac{X^T X}{n})U_2 - U_1^T\Sigma U_2\| \leq \mathcal{O}(\sigma^2\|\Sigma\|(\sqrt{\frac{k}{n}} + \frac{k}{n} + \sqrt{\frac{\log(1/\delta)}{n}} + \frac{\log(1/\delta)}{n})) \tag{23}$$

which gives what we want. $\qquad\square$

The following lemma is a basic matrix perturbation result (see Tripuraneni et al. (2021b) Lemma 25).

**Lemma 36.** Let $A$ be a positive definite matrix and $E$ another matrix which satisfies $\|EA^{-1}\| \leq \frac{1}{4}$, then $F := (A + E)^{-1} - A^{-1}$ satisfies $\|F\| \leq \frac{4}{3}\|A^{-1}\|\|EA^{-1}\|$.

With these two technical lemmas, we are able to prove Lemma 32, 33.

*Proof of Lemma 32.* Notice that by the definition of $U$ and $\beta_U^\star$, we have $UU^T\beta_U^\star = \beta_U^\star$. We denote $\alpha^\star := U^T\beta_U^\star$, then we also have $\beta_U^\star = U\alpha^\star$. Therefore

$$A_1 = \widehat{U}(\widehat{U}^T X^T X\widehat{U})^{-1}\widehat{U}^T X^T X\beta_U^\star - \beta_U^\star$$
$$= \widehat{U}(\widehat{U}^T X^T X\widehat{U})^{-1}\widehat{U}^T X^T XU\alpha^\star - U\alpha^\star$$
$$= (\widehat{U}(\widehat{U}^T X^T X\widehat{U})^{-1}\widehat{U}^T X^T XU - U)\alpha^\star$$

We consider $\widehat{U} \in \mathbb{R}^{d\times k}$ and $\widehat{U}_\perp^T \in \mathbb{R}^{d\times(d-k)}$ be orthonormal projection matrices spanning orthogonal subspaces which are rank $k$ and rank $d - k$ respectively, so that $\text{range}(\widehat{U}) \oplus \text{range}(\widehat{U}_\perp) = \mathbb{R}^d$. Then $\Delta = dist(\widehat{U}, U^\star) = \|\widehat{U}_\perp^T U^\star\|_2$. Notice that $I_d = \widehat{U}\widehat{U}^T + \widehat{U}_\perp\widehat{U}_\perp^T$, we have

$$\widehat{U}(\widehat{U}^T X^T X\widehat{U})^{-1}\widehat{U}^T X^T XU^\star - U^\star$$
$$= \widehat{U}(\widehat{U}^T X^T X\widehat{U})^{-1}\widehat{U}^T X^T X(\widehat{U}\widehat{U}^T + \widehat{U}_\perp\widehat{U}_\perp^T)U^\star - U^\star$$
$$= \widehat{U}(\widehat{U}^T X^T X\widehat{U})^{-1}\widehat{U}^T X^T X\widehat{U}\widehat{U}^T U^\star + \widehat{U}(\widehat{U}^T X^T X\widehat{U})^{-1}\widehat{U}^T X^T X\widehat{U}_\perp\widehat{U}_\perp^T U^\star - U^\star$$
$$= \widehat{U}(\widehat{U}^T X^T X\widehat{U})^{-1}\widehat{U}^T X^T X\widehat{U}_\perp\widehat{U}_\perp^T U^\star + \widehat{U}\widehat{U}^T U^\star - U^\star$$
$$= \widehat{U}(\widehat{U}^T X^T X\widehat{U})^{-1}\widehat{U}^T X^T X\widehat{U}_\perp\widehat{U}_\perp^T U^\star - \widehat{U}_\perp\widehat{U}_\perp^T U^\star \tag{24}$$

Thus

$$\|A_1\|_{\Sigma_T}^2 = A_1^T\Sigma_T A_1$$
$$= \alpha^{\star T}(\widehat{U}(\widehat{U}^T X^T X\widehat{U})^{-1}\widehat{U}^T X^T XU - U)^T\Sigma_T(\widehat{U}(\widehat{U}^T X^T X\widehat{U})^{-1}\widehat{U}^T X^T XU - U)\alpha^\star$$
$$= \alpha^{\star T}(\widehat{U}(\widehat{U}^T X^T X\widehat{U})^{-1}\widehat{U}^T X^T X\widehat{U}_\perp\widehat{U}_\perp^T U^\star - \widehat{U}_\perp\widehat{U}_\perp^T U^\star)^T\Sigma_T$$
$$\qquad(\widehat{U}(\widehat{U}^T X^T X\widehat{U})^{-1}\widehat{U}^T X^T X\widehat{U}_\perp\widehat{U}_\perp^T U^\star - \widehat{U}_\perp\widehat{U}_\perp^T U^\star)\alpha^\star$$
$$\leq \|\alpha^\star\|^2\|\widehat{U}(\widehat{U}^T X^T X\widehat{U})^{-1}\widehat{U}^T X^T X\widehat{U}_\perp\widehat{U}_\perp^T U^\star - \widehat{U}_\perp\widehat{U}_\perp^T U^\star\|_{\Sigma_T}^2$$
$$\leq \|\alpha^\star\|^2(\|\widehat{U}(\widehat{U}^T X^T X\widehat{U})^{-1}\widehat{U}^T X^T X\widehat{U}_\perp\widehat{U}_\perp^T U^\star\|_{\Sigma_T}^2 + \|\widehat{U}_\perp\widehat{U}_\perp^T U^\star\|_{\Sigma_T}^2). \tag{25}$$

Here we use the notation $\|M\|_{\Sigma_T} := \sqrt{\|M^T\Sigma_T M\|}$ for matrix $M$.

For the second term,

$$\|\widehat{U}_\perp \widehat{U}_\perp^T U^\star\|_{\Sigma_T}^2 \leq \|\widehat{U}_\perp^T \Sigma_T \widehat{U}_\perp\|\|\widehat{U}_\perp^T U^\star\|^2 \leq \Delta^2 \|\widehat{U}_\perp^T \Sigma_T \widehat{U}_\perp\|. \tag{26}$$

For the first term,

$$\|\widehat{U}(\widehat{U}^T X^T X \widehat{U})^{-1}\widehat{U}^T X^T X \widehat{U}_\perp \widehat{U}_\perp^T U^\star\|_{\Sigma_T}^2$$

$$= \|\widehat{U}(\widehat{U}^T \frac{X^T X}{n} \widehat{U})^{-1}\widehat{U}^T \frac{X^T X}{n} \widehat{U}_\perp \widehat{U}_\perp^T U^\star\|_{\Sigma_T}^2$$

$$= \|\widehat{U}((\widehat{U}^T \Sigma_S \widehat{U})^{-1} + F)(\widehat{U}^T \Sigma_S \widehat{U}_\perp \widehat{U}_\perp^T U^\star + E_1)\|_{\Sigma_T}^2$$

$$= \|(\widehat{U}^T \Sigma_S \widehat{U}_\perp \widehat{U}_\perp^T U^\star + E_1)^T((\widehat{U}^T \Sigma_S \widehat{U})^{-1} + F)^T \widehat{U}^T \Sigma_T \widehat{U}((\widehat{U}^T \Sigma_S \widehat{U})^{-1} + F)(\widehat{U}^T \Sigma_S \widehat{U}_\perp \widehat{U}_\perp^T U^\star + E_1)\|$$

$$\leq \|\widehat{U}^T \Sigma_S \widehat{U}_\perp \widehat{U}_\perp^T U^\star + E_1\|^2\|(\widehat{U}^T \Sigma_S \widehat{U})^{-1} + F\|^2\|\widehat{U}^T \Sigma_T \widehat{U}\|$$

$$\leq (\|\widehat{U}^T \Sigma_S \widehat{U}_\perp \widehat{U}_\perp^T U^\star\| + \|E_1\|)^2(\|(\widehat{U}^T \Sigma_S \widehat{U})^{-1}\| + \|F\|)^2\|\widehat{U}^T \Sigma_T \widehat{U}\| \tag{27}$$

where $E_1 = \widehat{U}^T \frac{X^T X}{n} \widehat{U}_\perp \widehat{U}_\perp^T U^\star - \widehat{U}^T \Sigma_S \widehat{U}_\perp \widehat{U}_\perp^T U^\star$, $F = (\widehat{U}^T \frac{X^T X}{n} \widehat{U})^{-1} - (\widehat{U}^T \Sigma_S \widehat{U})^{-1}$. We aim to show that $\|E_1\| \leq \|\widehat{U}^T \Sigma_S \widehat{U}_\perp \widehat{U}_\perp^T U^\star\|$ and $\|F\| \leq \|(\widehat{U}^T \Sigma_S \widehat{U})^{-1}\| = C_{\min}^{-1}$ for sufficiently large $n$, therefore the term in (27) can be bounded well. First we need a careful analysis of $\|\widehat{U}^T \Sigma_S \widehat{U}_\perp \widehat{U}_\perp^T U^\star\|$. It is obvious that

$$\|\widehat{U}^T \Sigma_S \widehat{U}_\perp \widehat{U}_\perp^T U^\star\| \leq \|\widehat{U}^T \Sigma_S \widehat{U}_\perp\|\|\widehat{U}_\perp^T U^\star\| \leq \Delta\|\widehat{U}^T \Sigma_S \widehat{U}_\perp\|. \tag{28}$$

As for $\|\widehat{U}^T \Sigma_S \widehat{U}_\perp\|$, notice that if without the "hat", we have $U^T \Sigma_S U_\perp = 0$ by the definition of $U$ and $\Sigma_S$ is diagonal. By definition of distance between two subspaces, there exist $R \in \mathcal{O}^{k\times k}$ and $Q \in \mathcal{O}^{(d-k)\times(d-k)}$, such that $\|\widehat{U}R - U\| = \Delta = \|\widehat{U}_\perp Q - U_\perp\|$. Then we have

$$\|\widehat{U}^T \Sigma_S \widehat{U}_\perp\| = \|R^T \widehat{U}^T \Sigma_S \widehat{U}_\perp Q\|$$

$$= \|U^T \Sigma_S U_\perp + R^T \widehat{U}^T \Sigma_S \widehat{U}_\perp Q - U^T \Sigma_S U_\perp\|$$

$$= \|R^T \widehat{U}^T \Sigma_S \widehat{U}_\perp Q - U^T \Sigma_S U_\perp\|$$

$$= \|R^T \widehat{U}^T \Sigma_S \widehat{U}_\perp Q - U^T \Sigma_S \widehat{U}_\perp Q + U^T \Sigma_S \widehat{U}_\perp Q - U^T \Sigma_S U_\perp\|$$

$$\leq \|R^T \widehat{U}^T \Sigma_S \widehat{U}_\perp Q - U^T \Sigma_S \widehat{U}_\perp Q\| + \|U^T \Sigma_S \widehat{U}_\perp Q - U^T \Sigma_S U_\perp\|$$

$$\leq \|R^T \widehat{U}^T - U^T\|\|\Sigma_S \widehat{U}_\perp Q\| + \|U^T \Sigma_S\|\|\widehat{U}_\perp Q - U_\perp\|$$

$$\leq 2\Delta\|\Sigma_S\|. \tag{29}$$

Combine (28) and (29), we have

$$\|\widehat{U}^T \Sigma_S \widehat{U}_\perp \widehat{U}_\perp^T U^\star\| \leq \mathcal{O}(\Delta^2\|\Sigma_S\|) \tag{30}$$

In order to bound $\|F\|$, let $E = \widehat{U}^T \frac{X^T X}{n} \widehat{U} - \widehat{U}^T \Sigma_S \widehat{U}$, then by Lemma 35, with probability at least $1 - \delta$,

$$\|E\| \leq \mathcal{O}(\sigma^2\|\Sigma_S\|(\sqrt{\frac{k}{n}} + \frac{k}{n} + \sqrt{\frac{\log(1/\delta)}{n}} + \frac{\log(1/\delta)}{n})). \tag{31}$$

Therefore,

$$\|E(\widehat{U}^T \Sigma_S \widehat{U})^{-1}\| \leq \|E\|\|(\widehat{U}^T \Sigma_S \widehat{U})^{-1}\|$$

$$\leq \|E\|C_{\min}^{-1}$$

$$\leq \mathcal{O}(\sigma^2 C_{\min}^{-1}\|\Sigma_S\|(\sqrt{\frac{k}{n}} + \frac{k}{n} + \sqrt{\frac{\log(1/\delta)}{n}} + \frac{\log(1/\delta)}{n})), \tag{32}$$

where $C_{\min} := \lambda_{\min}(\widehat{U}^T \Sigma_S \widehat{U})$. Notice that $n \gtrsim \sigma^4 C_{\min}^{-2}\|\Sigma_S\|^2 k \log(1/\delta)$ implies $\sqrt{\frac{k}{n}} + \frac{k}{n} + \sqrt{\frac{\log(1/\delta)}{n}} + \frac{\log(1/\delta)}{n} \lesssim \sigma^{-2} C_{\min}\|\Sigma_S\|^{-1}$. Thus, we show that when $n$ is large enough, we have $\|E(\widehat{U}^T \Sigma_S \widehat{U})^{-1}\| \leq \frac{1}{4}$. Therefore we can apply Lemma 36, which gives

$$\|F\| \leq \frac{4}{3}\|E(\widehat{U}^T \Sigma_S \widehat{U})^{-1}\|\|(\widehat{U}^T \Sigma_S \widehat{U})^{-1}\|$$

$$\leq \frac{4}{3} \times \frac{1}{4} \|(\widehat{U}^T \Sigma_S \widehat{U})^{-1}\|$$

$$\leq \frac{1}{3} C_{\min}^{-1}. \tag{33}$$

As for $\|E_1\|$, directly applying Lemma 35, when $n \gtrsim \sigma^4 \Delta^{-2} k \log(1/\delta)$ we get

$$\|E_1\| \leq \mathcal{O}(\sigma^2 \|\Sigma_S\| \|\widehat{U}_\perp \widehat{U}_\perp^T U^\star\| (\sqrt{\frac{k}{n}} + \frac{k}{n} + \sqrt{\frac{\log(1/\delta)}{n}} + \frac{\log(1/\delta)}{n}))$$

$$\leq \mathcal{O}(\sigma^2 \|\Sigma_S\| \Delta (\sqrt{\frac{k}{n}} + \frac{k}{n} + \sqrt{\frac{\log(1/\delta)}{n}} + \frac{\log(1/\delta)}{n})) \tag{34}$$

when $n \gtrsim \sigma^4 k \log(1/\delta)$ we have

$$\|E_1\| \leq \mathcal{O}(\Delta \|\Sigma_S\|) \tag{35}$$

, if further we have $n \gtrsim \sigma^4 \Delta^{-2} k \log(1/\delta)$, then

$$\|E_1\| \leq \mathcal{O}(\Delta^2 \|\Sigma_S\|). \tag{36}$$

Combining (27), (30), (33) and (36), we have

$$\|\widehat{U}(\widehat{U}^T X^T X \widehat{U})^{-1} \widehat{U}^T X^T X \widehat{U}_\perp \widehat{U}_\perp^T U^\star\|_{\Sigma_T}^2$$
$$\leq (\|\widehat{U}^T \Sigma_S \widehat{U}_\perp \widehat{U}_\perp^T U^\star\| + \|E_1\|)^2 (\|(\widehat{U}^T \Sigma_S \widehat{U})^{-1}\| + \|F\|)^2 \|\widehat{U}^T \Sigma_T \widehat{U}\|$$
$$\leq \mathcal{O}(\Delta^4 \|\Sigma_S\|^2 C_{\min}^{-2} \|\widehat{U}^T \Sigma_T \widehat{U}\|)$$
$$\leq \mathcal{O}(\Delta^4 \|\Sigma_S\|^2 C_{\min}^{-2} \|\Sigma_T\|) \tag{37}$$

Combining (25),(26) and (37), we get

$$\|A_1\|_{\Sigma_T}^2 \leq \|\alpha^\star\|^2 (\|\widehat{U}(\widehat{U}^T X^T X \widehat{U})^{-1} \widehat{U}^T X^T X \widehat{U}_\perp \widehat{U}_\perp^T U^\star\|_{\Sigma_T}^2 + \|\widehat{U}_\perp \widehat{U}_\perp^T U^\star\|_{\Sigma_T}^2)$$
$$\leq \mathcal{O}(\|\alpha^\star\|^2 (\Delta^4 \|\Sigma_S\|^2 C_{\min}^{-2} \|\Sigma_T\| + \Delta^2 \|\widehat{U}_\perp^T \Sigma_T \widehat{U}_\perp\|)) \tag{38}$$

with probability at least $1 - \delta$. Also, similar to (29), we have

$$\|\widehat{U}_\perp^T \Sigma_T \widehat{U}_\perp\| = \|Q^T \widehat{U}_\perp^T \Sigma_T \widehat{U}_\perp Q\|$$
$$\leq \|U_\perp^T \Sigma_T U_\perp\| + \|Q^T \widehat{U}_\perp^T \Sigma_T \widehat{U}_\perp Q - U_\perp^T \Sigma_T U_\perp\|$$
$$\leq \|U_\perp^T \Sigma_T U_\perp\| + 2\Delta \|\Sigma_T\| \tag{39}$$

Similarly, we can further know that $C_{\min}$ is close to $\lambda_k$:

$$C_{\min} = \lambda_k(\widehat{U}^T \Sigma_S \widehat{U})$$
$$= \lambda_k(R^T \widehat{U}^T \Sigma_S \widehat{U} R)$$
$$= \lambda_k(U^T \Sigma_S U + R^T \widehat{U}^T \Sigma_S \widehat{U} R - U^T \Sigma_S U)$$
$$\geq \lambda_k(U^T \Sigma_S U) - \|R^T \widehat{U}^T \Sigma_S \widehat{U} R - U^T \Sigma_S U\|$$
$$\geq \lambda_k(U^T \Sigma_S U) 2\Delta \|\Sigma_S\|$$
$$\geq \lambda_k - 2\lambda_1 \Delta$$
$$\geq \frac{1}{2} \lambda_k, \tag{40}$$

where the last inequality holds when $\Delta \leq \frac{\lambda_k}{4\lambda_1}$. Finally, combining (38), (39), (40), we have

$$\|A_1\|_{\Sigma_T}^2 \leq \mathcal{O}(\|\alpha^\star\|^2 (\Delta^4 (\frac{\lambda_1}{\lambda_k})^2 \|\Sigma_T\| + \Delta^2 \|U_\perp^T \Sigma_T U_\perp\| + \Delta^3 \|\Sigma_T\|))$$

$$\leq \mathcal{O}(\|\beta_U^\star\|^2 (\Delta^4 (\frac{\lambda_1}{\lambda_k})^2 \|\Sigma_T\| + \Delta^2 \|U_\perp^T \Sigma_T U_\perp\| + \Delta^3 \|\Sigma_T\|)) \tag{41}$$

when $\Delta \leq \frac{\lambda_k}{4\lambda_1}$ and $n \gtrsim \max\{\sigma^4(\frac{\lambda_1}{\lambda_k})^2 k \log(1/\delta), \sigma^4 \Delta^{-2} k \log(1/\delta)\}$. If in the previous proofs we replace (36) by (35), we have

$$\|A_1\|_{\Sigma_T}^2 \leq \mathcal{O}(\|\beta_U^\star\|^2 (\Delta^2 (\frac{\lambda_1}{\lambda_k})^2 \|\Sigma_T\| + \Delta^2 \|U_\perp^T \Sigma_T U_\perp\| + \Delta^3 \|\Sigma_T\|)) \tag{42}$$

$$\leq \mathcal{O}(\|\beta_U^\star\|^2 \Delta^2 (\frac{\lambda_1}{\lambda_k})^2 \|\Sigma_T\|) \tag{43}$$

when $\Delta \leq \frac{\lambda_k}{4\lambda_1}$ and $n \gtrsim \max\{\sigma^4(\frac{\lambda_1}{\lambda_k})^2 k \log(1/\delta), \sigma^4 k \log(1/\delta)\}$. Notice that by definition of $U$, $U_\perp^T \Sigma_T U_\perp = \Sigma_{T,-k}$, therefore the result is exactly what we want. $\qquad\square$

*Proof of Lemma 33.* Recall $A_3 := \widehat{U}(\widehat{U}^T X^T X \widehat{U})^{-1} \widehat{U}^T X^T \epsilon$. Therefore

$$\begin{aligned}
\|A_3\|_{\Sigma_T}^2 &= \epsilon^T X \widehat{U}(\widehat{U}^T X^T X \widehat{U})^{-1} \widehat{U}^T \Sigma_T \widehat{U}(\widehat{U}^T X^T X \widehat{U})^{-1} \widehat{U}^T X^T \epsilon \\
&= \mathrm{tr}(\epsilon^T X \widehat{U}(\widehat{U}^T X^T X \widehat{U})^{-1} \widehat{U}^T \Sigma_T \widehat{U}(\widehat{U}^T X^T X \widehat{U})^{-1} \widehat{U}^T X^T \epsilon) \\
&= \mathrm{tr}(\epsilon \epsilon^T X \widehat{U}(\widehat{U}^T X^T X \widehat{U})^{-1} \widehat{U}^T \Sigma_T \widehat{U}(\widehat{U}^T X^T X \widehat{U})^{-1} \widehat{U}^T X^T)
\end{aligned}$$

Taking expectation with respect to $\epsilon$, using $\mathbb{E}[\epsilon \epsilon^T] = v^2 I_n$, we have

$$\begin{aligned}
\mathbb{E}_\epsilon[\|A_3\|_{\Sigma_T}^2] &= \mathbb{E}[\mathrm{tr}(\epsilon \epsilon^T X \widehat{U}(\widehat{U}^T X^T X \widehat{U})^{-1} \widehat{U}^T \Sigma_T \widehat{U}(\widehat{U}^T X^T X \widehat{U})^{-1} \widehat{U}^T X^T)] \\
&= v^2 \mathrm{tr}(X \widehat{U}(\widehat{U}^T X^T X \widehat{U})^{-1} \widehat{U}^T \Sigma_T \widehat{U}(\widehat{U}^T X^T X \widehat{U})^{-1} \widehat{U}^T X^T) \\
&= v^2 \mathrm{tr}((\widehat{U}^T X^T X \widehat{U})^{-1} \widehat{U}^T \Sigma_T \widehat{U}(\widehat{U}^T X^T X \widehat{U})^{-1} \widehat{U}^T X^T X \widehat{U}) \\
&= v^2 \mathrm{tr}((\widehat{U}^T X^T X \widehat{U})^{-1} \widehat{U}^T \Sigma_T \widehat{U}) \\
&= \frac{1}{n} v^2 \mathrm{tr}(((\widehat{U}^T \Sigma_S \widehat{U})^{-1} + F) \widehat{U}^T \Sigma_T \widehat{U}) \tag{44}
\end{aligned}$$

Here we actually need a bound stronger than (33) for $\|F\|$: recall (32), we have with probability $1 - \delta$

$$\|E(\widehat{U}^T \Sigma_S \widehat{U})^{-1}\| \leq \mathcal{O}(\sigma^2 C_{\min}^{-1} \|\Sigma_S\|(\sqrt{\frac{k}{n}} + \frac{k}{n} + \sqrt{\frac{\log(1/\delta)}{n}} + \frac{\log(1/\delta)}{n})). \tag{45}$$

Applying Lemma 36, which gives

$$\begin{aligned}
\|F\| &\leq \frac{4}{3} \|E(\widehat{U}^T \Sigma_S \widehat{U})^{-1}\| \|(\widehat{U}^T \Sigma_S \widehat{U})^{-1}\| \\
&\leq \mathcal{O}(\sigma^2 C_{\min}^{-2} \|\Sigma_S\|(\sqrt{\frac{k}{n}} + \frac{k}{n} + \sqrt{\frac{\log(1/\delta)}{n}} + \frac{\log(1/\delta)}{n})) \\
&\leq \mathcal{O}(\frac{1}{k \|\Sigma_T\|} \mathrm{tr}((U^T \Sigma_S U)^{-1} U^T \Sigma_T U)) \tag{46}
\end{aligned}$$

when $n \gtrsim \sigma^4 C_{\min}^{-4} \|\Sigma_S\|^2 \|\Sigma_T\|^2 \mathrm{tr}((U^T \Sigma_S U)^{-1} U^T \Sigma_T U)^{-2} k^3 \log(1/\delta)$. Therefore we have

$$\begin{aligned}
\mathbb{E}_\epsilon[\|A_3\|_{\Sigma_T}^2] &= \frac{1}{n} v^2 \mathrm{tr}(((\widehat{U}^T \Sigma_S \widehat{U})^{-1} + F) \widehat{U}^T \Sigma_T \widehat{U}) \\
&= \frac{1}{n} v^2 (\mathrm{tr}((\widehat{U}^T \Sigma_S \widehat{U})^{-1} \widehat{U}^T \Sigma_T \widehat{U}) + \mathrm{tr}(F \widehat{U}^T \Sigma_T \widehat{U})) \\
&\leq \frac{1}{n} v^2 (\mathrm{tr}((\widehat{U}^T \Sigma_S \widehat{U})^{-1} \widehat{U}^T \Sigma_T \widehat{U})) + \frac{1}{n} v^2 \|F\| \mathrm{tr}(\widehat{U}^T \Sigma_T \widehat{U}) \\
&\leq \frac{1}{n} v^2 (\mathrm{tr}((\widehat{U}^T \Sigma_S \widehat{U})^{-1} \widehat{U}^T \Sigma_T \widehat{U})) + \frac{1}{n} v^2 k \|F\| \|\Sigma_T\| \\
&\leq \frac{1}{n} v^2 (\mathrm{tr}((\widehat{U}^T \Sigma_S \widehat{U})^{-1} \widehat{U}^T \Sigma_T \widehat{U})) + \frac{1}{n} v^2 \mathcal{O}(\mathrm{tr}((U^T \Sigma_S U)^{-1} U^T \Sigma_T U)) \tag{47}
\end{aligned}$$

The remaining thing is to show that indeed $\mathrm{tr}((\widehat{U}^T \Sigma_S \widehat{U})^{-1} \widehat{U}^T \Sigma_T \widehat{U})$ is close to $\mathrm{tr}((U^T \Sigma_S U)^{-1} U^T \Sigma_T U)$. In fact, $\mathrm{tr}((\widehat{U}^T \Sigma_S \widehat{U})^{-1} \widehat{U}^T \Sigma_T \widehat{U}) = \mathrm{tr}((R^T \widehat{U}^T \Sigma_S \widehat{U} R)^{-1} R^T \widehat{U}^T \Sigma_T R \widehat{U})$. Notice that

$$\|R^T\widehat{U}^T\Sigma_T\widehat{U}R - U^T\Sigma_T U\| \le 2\|\Delta\|\|\Sigma_T\|,$$

we have

$$\text{tr}((R^T\widehat{U}^T\Sigma_S\widehat{U}R)^{-1}R^T\widehat{U}^T\Sigma_T\widehat{U}R) \tag{48}$$

$$\le \text{tr}((R^T\widehat{U}^T\Sigma_S\widehat{U}R)^{-1}U^T\Sigma_T U) + \|R^T\widehat{U}^T\Sigma_T\widehat{U}R - U^T\Sigma_T U\|\,\text{tr}((\widehat{U}^T\Sigma_S\widehat{U})^{-1})$$

$$\le \text{tr}((R^T\widehat{U}^T\Sigma_S\widehat{U}R)^{-1}U^T\Sigma_T U) + 2\|\Delta\|\|\Sigma_T\|\,\text{tr}((\widehat{U}^T\Sigma_S\widehat{U})^{-1})$$

$$\le \text{tr}((R^T\widehat{U}^T\Sigma_S\widehat{U}R)^{-1}U^T\Sigma_T U) + 2\|\Delta\|\|\Sigma_T\|kC_{\min}^{-1}$$

$$\le \text{tr}((R^T\widehat{U}^T\Sigma_S\widehat{U}R)^{-1}U^T\Sigma_T U) + \text{tr}((U^T\Sigma_S U)^{-1}U^T\Sigma_T U) \tag{49}$$

when $\Delta \le \frac{\lambda_k\,\text{tr}((U^T\Sigma_S U)^{-1}U^T\Sigma_T U)}{4k\|\Sigma_T\|}$. Also, we have

$$\|(R^T\widehat{U}^T\Sigma_S\widehat{U}R)^{-1} - (U^T\Sigma_S U)^{-1}\| \le \|(R^T\widehat{U}^T\Sigma_S\widehat{U}R)^{-1}\|\|(U^T\Sigma_S U)^{-1}\|\|R^T\widehat{U}^T\Sigma_S\widehat{U}R - U^T\Sigma_S U\|$$

$$\le 4\lambda_k^{-2}\lambda_1\Delta,$$

therefore

$$\text{tr}((R^T\widehat{U}^T\Sigma_S\widehat{U}R)^{-1}U^T\Sigma_T U) \le \text{tr}((U^T\Sigma_S U)^{-1}U^T\Sigma_T U) + \|(R^T\widehat{U}^T\Sigma_S\widehat{U}R)^{-1} - (U^T\Sigma_S U)^{-1}\|\,\text{tr}(U^T\Sigma_T U)$$

$$\le \text{tr}((U^T\Sigma_S U)^{-1}U^T\Sigma_T U) + 4\lambda_k^{-2}\lambda_1\Delta\,\text{tr}(U^T\Sigma_T U)$$

$$\le 2\,\text{tr}((U^T\Sigma_S U)^{-1}U^T\Sigma_T U), \tag{50}$$

if $\Delta \le \frac{\lambda_k^2\,\text{tr}((U^T\Sigma_S U)^{-1}U^T\Sigma_T U)}{4\lambda_1\,\text{tr}(U^T\Sigma_T U)}$. Combining (47), (48) and (50) we have

$$\mathbb{E}_\epsilon[\|A_3\|_{\Sigma_T}^2] \le \mathcal{O}(\frac{1}{n}v^2\,\text{tr}((U^T\Sigma_S U)^{-1}U^T\Sigma_T U)),$$

whenever $\Delta \le \frac{\lambda_k^2\,\text{tr}((U^T\Sigma_S U)^{-1}U^T\Sigma_T U)}{4\lambda_1 k\|\Sigma_T\|} \le \min\{\frac{\lambda_k^2\,\text{tr}((U^T\Sigma_S U)^{-1}U^T\Sigma_T U)}{4\lambda_1\,\text{tr}(U^T\Sigma_T U)}, \frac{\lambda_k\,\text{tr}((U^T\Sigma_S U)^{-1}U^T\Sigma_T U)}{4k\|\Sigma_T\|}\}$ and $n \gtrsim \sigma^4 C_{\min}^{-4}\|\Sigma_S\|^2\|\Sigma_T\|^2\,\text{tr}((U^T\Sigma_S U)^{-1}U^T\Sigma_T U)^{-2}k^3\log(1/\delta)$, with probability at least $1-\delta$. Notice that $U^T\Sigma_S U = \Sigma_{S,k}$ and $U^T\Sigma_T U = \Sigma_{T,k}$, therefore the result is exactly what we want. $\qquad\square$

*Proof of Lemma 34.* Recall $A_2 := \widehat{U}(\widehat{U}^T X^T X\widehat{U})^{-1}\widehat{U}^T X^T X\beta_\perp^\star$. Also we have

$$\|\widehat{U}^T\Sigma_T\widehat{U}\| = \|R^T\widehat{U}^T\Sigma_T\widehat{U}R\|$$

$$\le \|U^T\Sigma_T U\| + \|R^T\widehat{U}^T\Sigma_T\widehat{U}R - U^T\Sigma_T U\|$$

$$\le \|U^T\Sigma_T U\| + 2\Delta\|\Sigma_T\| \tag{51}$$

Therefore

$$\|A_2\|_{\Sigma_T}^2 = \|\beta_\perp^{\star T} X^T X\widehat{U}(\widehat{U}^T X^T X\widehat{U})^{-1}\widehat{U}^T\Sigma_T\widehat{U}(\widehat{U}^T X^T X\widehat{U})^{-1}\widehat{U}^T X^T X\beta_\perp^\star\|$$

$$\le \|X\widehat{U}(\widehat{U}^T X^T X\widehat{U})^{-1}(\widehat{U}^T X^T X\widehat{U})^{-1}\widehat{U}^T X^T\|\|\widehat{U}^T\Sigma_T\widehat{U}\|\|X\beta_\perp^\star\|^2$$

$$\le \|A\|(\|U^T\Sigma_T U\| + 2\Delta\|\Sigma_T\|)\|X\beta_\perp^\star\|^2$$

$$\le 2\|A\|\|U^T\Sigma_T U\|\|X\beta_\perp^\star\|^2 \tag{52}$$

when $\Delta \le \frac{\|U^T\Sigma_T U\|}{2\|\Sigma_T\|}$, where we let $A = \frac{1}{n}\frac{X\widehat{U}}{\sqrt{n}}(\widehat{U}^T\frac{X^T X}{n}\widehat{U})^{-2}\frac{\widehat{U}^T X^T}{\sqrt{n}}$. If we define $B = \frac{X\widehat{U}}{\sqrt{n}} \in \mathbb{R}^{n\times r}$, then $A = \frac{1}{n}B(B^T B)^{-2}B^T$. Let the SVD of $B$ be $B = PMO^T$, where $P \in \mathbb{R}^{n\times k}$, $M, O \in \mathbb{R}^{k\times k}$, then

$$\|A\|_2 = \frac{1}{n}\|B(B^T B)^{-2}B^T\|_2$$

$$= \frac{1}{n}\|PMO^T(OM^2 O^T)^{-2}OMP^T\|_2$$

$$= \frac{1}{n}\|PM^{-2}P^T\|_2$$

$$\leq \frac{1}{n}\|M^{-2}\|_2$$
$$= \frac{1}{n}\|(B^T B)^{-1}\|_2 \tag{53}$$

Let $F = (\widehat{U}^T \frac{X^T X}{n} \widehat{U})^{-1} - (\widehat{U}^T \Sigma \widehat{U})^{-1}$. Recall (33), which states that with probability at least $1 - \delta$, we have $\|F\| \leq \frac{1}{3} C_{\min}^{-1} \leq \frac{2}{3} \lambda_k^{-1}$ when $n \gtrsim \sigma^4 C_{\min}^{-2} \|\Sigma_S\|^2 k \log(1/\delta)$ and $\Delta \leq \frac{\lambda_k}{4\lambda_1}$. Therefore

$$\|A\| \leq \frac{1}{n}\|(\widehat{U}^T \frac{X^T X}{n} \widehat{U})^{-1}\|$$
$$= \|(\widehat{U}^T \Sigma_S \widehat{U})^{-1} + F\|$$
$$\leq \frac{1}{n}\|(\widehat{U}^T \Sigma_S \widehat{U})^{-1}\| + \|F\|$$
$$\leq \mathcal{O}(\frac{1}{n}\lambda_k^{-1}). \tag{54}$$

Thus $\|A\| \leq \mathcal{O}(\lambda_k^{-1})$. As for $\|X\beta_\perp^\star\|^2$, notice that the first-$k$ entries of $\beta_\perp^\star$ are zero, therefore $X\beta_\perp^\star = X_{-k}\beta_{-k}^\star$. by Lemma 35,

$$\|\beta_{-k}^{\star T}(\frac{X_{-k}^T X_{-k}}{n})\beta_{-k}^\star - \beta_{-k}^{\star T}\Sigma_{S,-k}\beta_{-k}^\star\| \leq \mathcal{O}(\sigma^2 \|\beta_{-k}^\star\|^2 \|\Sigma_{S,-k}\|(\sqrt{\frac{1}{n}} + \frac{1}{n} + \sqrt{\frac{\log(1/\delta)}{n}} + \frac{\log(1/\delta)}{n}). \tag{55}$$

Therefore we have

$$\|X\beta_\perp^\star\|^2 = n\beta_{-k}^{\star T}(\frac{X_{-k}^T X_{-k}}{n})\beta_{-k}^\star$$
$$\leq n(\beta_{-k}^{\star T}\Sigma_{S,-k}\beta_{-k}^\star + \|\beta_{-k}^{\star T}(\frac{X_{-k}^T X_{-k}}{n})\beta_{-k}^\star - \beta_{-k}^{\star T}\Sigma_{S,-k}\beta_{-k}^\star\|)$$
$$\leq \mathcal{O}(n\|\beta_{-k}^\star\|^2 \|\Sigma_{S,-k}\|). \tag{56}$$

Combining (52)(54) and (56), we have

$$\|A_2\|_{\Sigma_T}^2 \leq \mathcal{O}(\frac{\|U^T \Sigma_T U\|\|\beta_{-k}^\star\|^2 \|\Sigma_{S,-k}\|}{\lambda_k}) \tag{57}$$

when $n \gtrsim \sigma^4 C_{\min}^{-2} \|\Sigma_S\|^2 k \log(1/\delta)$ and $\Delta \leq \min\{\frac{\|U^T \Sigma_T U\|}{2\|\Sigma_T\|}, \frac{\lambda_k}{4\lambda_1}\}$. $\qquad\square$

Finally we prove Lemma 30 in the following.

*Proof of Lemma 30.* In the first step, we obtain $\widehat{U} \in \mathbb{R}^{d \times k}$ by selecting the top$-k$ eigenvectors of the sample covariance matrix $\widehat{\Sigma}_S := \frac{1}{n}XX^T = \frac{1}{n}\sum_{i=1}^n x_i x_i^T$ using PCA. Then by Davis-Kahan theorem (Chen et al., 2021, Corollary 2.8),

$$\Delta \leq \frac{2\|\widehat{\Sigma}_S - \Sigma_S\|}{\lambda_k - \lambda_{k+1}}. \tag{58}$$

Therefore it remains to bound $\|\widehat{\Sigma}_S - \Sigma_S\|$. Applying Lemma 28, we immediately have

$$\|\widehat{\Sigma}_S - \Sigma_S\| \leq C\sigma^4 \left(\sqrt{\frac{r + \log\frac{1}{\delta}}{n}} + \frac{r + \log\frac{1}{\delta}}{n}\right)\lambda_1$$

where $r = \frac{\sum_{i=1}^n \lambda_i}{\lambda_1}$. Together with (58), we have with probability at least $1 - \delta$,

$$\Delta \leq C\sigma^4 \left(\sqrt{\frac{r + \log\frac{1}{\delta}}{n}} + \frac{r + \log\frac{1}{\delta}}{n}\right)\frac{\lambda_1}{\lambda_k - \lambda_{k+1}}.$$

$\qquad\square$

