# OpenReview forum: "Benign Overfitting in Out-of-Distribution Generalization of Linear Models"
_ICLR.cc/2025/Conference — ICLR 2025 Poster_

### Official Review · Reviewer_MgYk · 2024-10-28

**Soundness:** 3
**Presentation:** 4
**Contribution:** 2
**Rating:** 6
**Confidence:** 3

**Summary:**

The authors study the over-paramterized ridge regression under covariate shift assumption. Specifically,

1. The manuscript shows the ridge regression exhibits “benign overfitting” given the shifts in minor direction for target domain is smaller than source domains’ counterpart.
2. When there is significant components in the minor directions, the manuscript shows ridge regression only achieves a lower rate, e.g. $n^{-\frac{1}{2}}$. It also show using the PCR can achieve the fast rate $n^{-1}$ as in-distribution learning.

**Strengths:**

1. The authors provide extensive context to introduce the background and problem, making the logic very easy to follow.
2. Clear discussion about the role of $\mathcal{T}$ and the overall magnitude of $\Sigma_{T,-K}$.
3. The instance-specified lower bound for ridge regression for large shift in minor direction scenario.

**Weaknesses:**

##

1. For the large shifts in minor direction case, while I really appreciate the instance-specified lower bound for ridge regression, the result for PCR is more like “Shoot The Arrow, Then Draw The Target.” Given the assumption that the true signal primarily lies in the major directions of the source, it is not surprising that PCR works well as one has priorly excluded those un-handleable/irreducible parts in covariate shift, which eventually works like the process in Section 3.
2. The lower bound is instance specified. Although this is already a good example to show ridge regression is not a generic good choice for large shifts, but having the lower bound for general case make the contribution of the manuscript more convincing. Can authors elaborate more on the main obstacle for the lower bound in general case?
3. The fast rate in PCR is not adaptive. Similar to (1), if you know the true cut-off $k$, you pick the correct number of eigenvectors, this degrades the technical contribution of this manuscript. It is desired to have adaptive estimator/learner to adapt to the unknown $k$ and achieve fast rate.
4. (Potentially) Reviewer is not coming from the research area that working on over-parameterized models. Therefore, the reviewer cannot assert the technical contribution or novelty of this manuscript (although I see a detailed discussion on OOD in the over-parameterized model in related work ). I would like to leave this judgment to my peer reviewers, who are more familiar with this area.

**Questions:**

1. As the comment I raised in W3, is it possible to provide adaptive rate for PCR?
2. While the lower bound for ridge regression is instance-specified, the general intuition is that ridge regression typically fit the space in the whole scale. However, I would like to know if there is any special case that ridge regression can attain fast rate in large shifts.

---

> ### Author Response · Authors · 2024-11-19
> **Reply to Reviewer MgYk**
>
> Thank the reviewer for your positive response and valuable suggestions. The following is our response to your comments and suggestions.
>
>
> Q1: The result for PCR is more like “Shoot The Arrow, Then Draw The Target.”
>
> A1: We want to clarify that, considering the scenario where “the true signal primarily lies in the major directions” is necessary, and PCR appears to be a natural algorithm under this scenario. As we discussed at the beginning of Sec 4.2,  the signal in the minor directions is nearly lost since the eigenvalues of $\\Sigma_S$ in those directions are so small. In other words, learning the true signal from the minor directions is essentially impossible. Therefore assuming the true signal primarily lies in the major directions is necessary (actually in prior work of benign overfitting [Tsigler & Bartlett 2023], this is also implicitly assumed, as you can see there is a term regarding $\\beta^{\\star}\_{-k}$ in the upper bound).
> As long as the true signal primarily lies in the major directions, it is natural to consider PCR. We want to further clarify that we assume $\\beta^\\star_{-k}=0$ in Theorem 5 just for presentation clarity. We mentioned in Remark 5 that Lemma 31 provides the generic results without assuming $\\beta^{\\star}_{-k}=0$. One can see that in this case there will be a term in the upper bound regarding $\beta^{\*}\_{-k}$ as in ridge regression.
>
> Q2: Can authors elaborate more on the main obstacle for the lower bound in general case?
>
> A2: The proof techniques in prior work [Tsigler & Bartlett 2023] for the lower bound for in-distribution case crucially rely on the fact that they can express the variance term as a function of many independent subGaussian vectors, while this relies on the source and target covariance share the same eigenvectors (i.e., they are simultaneously diagonalizable). Here we do not assume such a strong assumption, therefore establishing a matching lower bound is much more difficult.
>
>
> Q3: The fast rate in PCR is not adaptive.
>
> A3: PCR is not adaptive due to its nature of applying principal component analysis. We need to choose a $k$ prior to applying PCR. There are some heuristics when choosing $k$: from our upper bound, we can see that we need to choose a $k$ such that $\lambda_k-\lambda_{k+1}$ is large. And we also want $\lambda_k$ not to be too small, since we want $tr(\mathcal{T})$ to be small. Such a $k$ can be obtained by drawing a “scree plot” which plots the eigenvalues of the sample covariance matrix in a descending order, and picking the “elbow” points.
> Also we want to clarify that, there is not a “correct” $k$. Our bound applies to any $k$. How to find a good $k$ (for example, using the “scree plot”) for the algorithm is another interesting direction.
>
> Q4: I would like to know if there is any special case that ridge regression can attain fast rate in large shifts.
>
> A4: For large shifts in minor directions, where the overall magnitude of $\Sigma_{T, -k}$ is large, we believe that in general ridge regression can not attain a fast rate. Intuitively when $\Sigma_{S, -k}$ has small eigenvalues, ridge regression will induce large variances on these directions, and this will cause a large excess risk when the overall magnitude of $\Sigma_{T, -k}$ is large.

---

> > ### Comment · Reviewer_MgYk · 2024-11-26
> >
> > I thank the authors for the detailed response. I'm happy with most of the answers that the authors provided except for Q1. I still think if you have already put some (either explicit or implicit) assumptions on the data (so as the model), then employing PCR is like working under well-specified model scenarios, i.e., you use the priorly known info about data or models to pick your estimation procedure. For me, it is more interesting to see what kind of misspecified models can be used and also achieve similar performance as PCR. But I think it would be out of the scope of this paper. I will keep my positive evaluation on this paper.

---

### Official Review · Reviewer_ZYgW · 2024-10-29

**Soundness:** 3
**Presentation:** 3
**Contribution:** 2
**Rating:** 8
**Confidence:** 3

**Summary:**

This paper considers a model to study benign overfitting in the context of out-of-distribution (OOD) generalisation for overparameterized linear models. The authors focus on covariate shift.

The authors prove the following results
* The paper derives an excess risk upper bound the above setting. This bound generalizes directly the one in [Tsigler&Bartlett 2023].
* The authors consider specific choices of covariance matrices to consider two opposite cases:
  * If the major directions of the shifted model are the same and the minor directions remain small the begning overfitting still is present reaching a $O(1/n)$ bound for the excess risk.
  * For any ridge parameter $\lambda >0$, under a different covariance model, the authors show a lower bound on the excess risk as $O(1/\sqrt{n})$.
* To mitigate this problem the authors also consider principal component regression and show that under specific conditions it always achieves the $O(1/n)$ rate.

**Strengths:**

* First work to provide non-asymptotic guarantees for benign overfitting under general covariate shift
* Provides practical insights into when ridge regression vs PCR should be used.
* Results generalize and recover previous known bounds as special cases fitting in nicely with the literature on the topics.

**Weaknesses:**

* The clarity of the paper could be improved. While it is very clear the discussion about previous results and how the current result generalise the old ones I feel that the assumptions for the original contributions are hidden in the appendix. The paper would greatly benefit from a clear statement of the assumptions at the beginning or just a discussion of them. A specific instance of this Theorem 5 where the result depends on the additional assumptions that $\beta^{\star}_{-k} = 0$ and the fact that the overlap gap between major and minor directions is large.
* Another problem that I find, connected to the previous one, is the lack of simulation studies to illustrate the theoretical findings. As the setting of linear regression with general covariances is generic enough one could back up the claim with experimental validation on real datasets.

**Questions:**

* Are the conditions considered in Section 3.1 the most general ones for which the rate of the excess risk is slow?

---

> ### Author Response · Authors · 2024-11-19
> **Reply to Reviewer ZYgW**
>
> Thank the reviewer for your positive feedback and valuable comments and suggestions. The following is our response to your comments.
>
> Q1: Numerical experiments.
>
> A1: We thank the reviewer for suggesting the consolidation of theory with simulations. We have included two simulation experiments in Appendix A. The first experiment takes the benign overfitting setup proposed in the paper, involving small shifts in the minor directions. It confirms the $\mathcal O(1/n)$ rate for ridge regression. Data is generated from a multivariate normal distribution, with target covariance matrices randomly generated. We validate the influence of two factors $\\|\\mathcal T\\|$, $ \\mathrm{tr}[U]/\\mathrm{tr}[V] $, identified in the paper as measures of covariate shifts in major and minor directions, respectively. The experiment results show that, for each combination of $\\|\\mathcal T\\|$ and $ \\mathrm{tr}[U]/\\mathrm{tr}[V] $, the excess risk of ridge regression decays at nearly 1/n. For a fixed sample size, the excess risk increases with larger values of $\\|\\mathcal T\\|$ or $ \\mathrm{tr}[U]/\\mathrm{tr}[V] $.
>
> The second experiment compares ridge regression with Principal Component Regression (PCR) under large shifts in the minor directions. The setup follows the instance in Theorem 4, where the excess risk of ridge regression is lower bounded by $\mathcal O(1/ \sqrt n)$, while PCR achieves an excess risk of $\mathcal O(1/n)$. This result is confirmed by the experiment, where the excess risk of PCR is compared against ridge regression with various regularization strengths: $\lambda = 0, n^{0.5}, n^{0.75}, n$. The findings show that the excess risk of ridge decays optimally at the rate of $n^{-0.48}$ for $\lambda = n^{0.75}$, consistent with the $\mathcal O(1/ \sqrt n)$ lower bound in Theorem 4. The optimal regularization in the experiment also aligns with that derived in the theoretical proof. In contrast, PCR achieves a superior decay rate of $n^{-0.99}$.
>
> Q2: While it is very clear the discussion about previous results and how the current result generalise the old ones I feel that the assumptions for the original contributions are hidden in the appendix.
>
> A2: We want to emphasize that we have included all assumptions in the main text and do not hide our assumptions in the appendix. Here, we would like to clarify our main assumptions again. In ridge regression, actually our assumptions are the same as the prior in-distribution results [Tsigler & Bartlett 2023]. All the assumptions are listed in the setup section and in the beginning of Section 3. We would also like to point out that our upper bound for ridge regression does not require assumptions that go beyond prior works. It is one of our core contributions that we generalize the in-distribution benign overfitting results without introducing additional assumptions. This stands in contrast to related work in section 1.1 which poses strong assumptions on target distributions.
> As for the PCR results in section 4, we use the same setup in section 2, and Assumption 1 [Tsigler & Bartlett 2023]  in section 3 is no longer required. Actually we assume $\beta^{\*}_{-k}=0$ in Theorem 5 just for presentation clarity. We mentioned in Remark 5 that Lemma 31 provides the generic results without assuming $\\beta^{*}\_{-k}=0$. Regarding the gap between major and minor directions, it is not an assumption because our result holds regardless of this gap. Following the theorem, we discuss the impact of this gap on our bound.
> We thank the reviewer for suggestions to further improve clarity of assumptions. We modified our presentation in the beginning of section 4 to include the above statements.
>
> Q3: Are the conditions considered in Section 3.1 the most general ones for which the rate of the excess risk is slow?
>
> A3: We believe that the conditions in Sec 3.1 are general for Theorem 2 to hold. Regarding the slow rate for ridge regression, we further need the overall magnitude of the minor components on target to be small.

---

> ### Comment · Reviewer_ZYgW · 2024-11-21
>
> I sincerely thank the authors for the time to produce the numerical simulations. I believe that they give a more visual backing to the results.
>
> I also better understand the results and the different assumptions used for the various results of the paper. I want to thank the authors for their clarity and patience.
>
> I have thus decided to increase my score.

---

### Official Review · Reviewer_NDnK · 2024-10-31

**Soundness:** 3
**Presentation:** 4
**Contribution:** 3
**Rating:** 6
**Confidence:** 2

**Summary:**

This manuscript investigate benign overfitting in out-of-distribution (OOD) generalization, focusing on over-parameterized linear models under covariate shift. It extends the concept of benign overfitting from in-distribution cases to settings where the target distribution differs from the training (source) distribution. The authors provide non-asymptotic excess risk upper bounds for ridge and principal component regression under specific structural conditions on the target covariance. More precisely, the main contribution is an instance dependent upper bound on the bias and variance terms of the excess risk of ridge regression under OOD. In particular, this upper bound shows that benign overfitting transfers from the in-distribution to the OOD setting when the target distribution’s covariance along the high-variance (or "major") directions aligns well with the source distribution’s major directions.

The authors also provide a discussion of an example with significant shifts in the minor directions, showing that in this case ridge regression incur a high excess risk, despite overfitting being benign in-distribution. Finally, the authors show that doing principal component regression (i.e. ridge regression only on the major directions) can mitigate this phenomena, since it avoids the excess error contributions from misaligned minor directions.

**Strengths:**

The manuscript is well written and easy to follow. It contains a nice balance of formal and intuitive discussion. The review of the in-distribution benign overfitting results is a nice addition that helps highlighting the contributions. Overall, I think it is a good contribution on a topic of interest to the theoretical community at ICLR.

**Weaknesses:**

The lack of a matching lower bound as in the in-distribution case of [Tsigler & Bartlett, 2023] is a weak point of the manuscript. Another minor weakness is that the technical contribution is also somehow limited, as it mostly relies mostly on extending existing results.

**Questions:**

- What are the main challenges in proving a matching lower bound for ODD similar to the in-distribution case?
- The fact that the analysis of OOD boils down only to the alignements of the covariance of the source and target distribution is closely to the square loss and linear estimator. How much should we expect this to transfer to other tasks, such as linear classification for instance?

- L39-42:
> "*However, over-parameterized models, such as deep neural networks and large language models (LLMs), which have more parameters than training samples, are widely used in modern machine learning.*"

This sentence is misleading and overly general. In fact, most of modern LLMs use more tokens than parameters for training. See for example Table 2.1 in [1].

- L11 in the abstract: "over-paramterized"

**References**

- [1] Brown et al. [Language Models are Few-Shot Learners](https://arxiv.org/pdf/2005.14165). NeurIPS 2020

---

> ### Author Response · Authors · 2024-11-19
> **Reply to Reviewer NDnK**
>
> Thank the reviewer for your positive feedback and valuable comments and suggestions. Regarding your concerns and suggestions, we write our response as the following:
>
>
> Q1: What are the main challenges in proving a matching lower bound for OOD similar to the in-distribution case? Also what is the technical contribution of this work?
>
> A1: The proof techniques in prior work [Tsigler & Bartlett 2023] for both the upper bound and lower bound crucially rely on the source and target distribution are the same. To be specific, in proving the lower bound for the in-distribution case, they can express the variance term as a function of many independent subGaussian vectors, while this relies on the condition that the source and target covariance share the same eigenvectors (i.e., they are simultaneously diagonalizable). In our work, we do not assume such a strong assumption. Therefore, establishing a matching lower bound is much more difficult.
> Also, regarding the technical contribution, when establishing the upper bound, we found that if naively applying previous techniques, the derived upper bound is loose. As an example, by previous techniques, the variance of the first $k$ components will scale as $\frac{tr(\Sigma_S \Sigma_T^{-1})}{n\mu_k(\Sigma_S \Sigma_T^{-1})}$. This quantity depends on $\Sigma_S \Sigma_T^{-1}$ which is extremely loose because it indicates smaller target covariance causes larger excess risk in some cases. In fact, it is inconsistent with previous art [Ge et al. 2024] demonstrating that $\Sigma_S^{-1} \Sigma_T$ captures the covariate shift in under-parameterized linear regression. To deal with this issue, we establish a new technique for deriving a tight bound which reflects our intuition.
>
> Q2: The fact that the analysis of OOD boils down only to the alignments of the covariance of the source and target distribution is closely to the square loss and linear estimator. How much should we expect this to transfer to other tasks, such as linear classification for instance?
>
> A2: We can extend the current results beyond the linear model by considering kernel ridge regression. In practice, nonlinear estimators can be well approximated by RKHS. As long as a proper kernel is chosen, the kernel ridge regression becomes the linear regression on the transformed feature space. In fact, our results also hold for infinite-dimensional linear regression. Therefore, our bound can also be applied to kernel ridge regression.
> As for classification problems, it remains unknown whether “benign overfitting” might happen. This can be an interesting future direction.
>
>
> Q3: Misleading sentence in L39-42 and typo in L11
>
> A3: Thank you very much for pointing out an improper sentence and a typo in our writing. The typo is addressed. And we modify the sentence in L39-42, mentioning that LLM can be viewed as over-parameterized during the fine-tuning stage.

---

> ### Comment · Reviewer_NDnK · 2024-11-26
>
> I thank the authors for their rebuttal, which addressed my questions. I am keeping my positive evaluation.

---

### Official Review · Reviewer_R8Hj · 2024-11-01

**Soundness:** 3
**Presentation:** 4
**Contribution:** 4
**Rating:** 8
**Confidence:** 3

**Summary:**

The paper provides an analysis of benign overfitting in the out-of-distribution regime. Both ridge regression and PCR are analyzed and have rates $O(\frac{1}{\sqrt{n}})$ and $O(\frac{1}{n})$.

**Strengths:**

The paper is very well written. It presents complex results with clarity and cohesion. The results are novel to the best of my knowledge and the setup under analysis is very interesting.

I didn't thoroughly check all the proofs. But the steps in the main text seem logical to me.

The differences between rates for Principal Components Regression and Ridge Regression are surprising and interesting.

**Weaknesses:**

I believe the paper could benefit from some simulation experiments confirming the theoretical results.

For instance, verifying the rates  $O(\frac{1}{\sqrt{n}})$ and $O(\frac{1}{n})$ hold for a small Gaussian example would strengthen the claim, and help to support that the mathematical proofs are correct.

**Questions:**

- Does the result in PCR requires the number of relevant components k to be known in advance? What is the effect if the number is mispecified

---

> ### Author Response · Authors · 2024-11-19
> **Reply to Reviewer R8Hj**
>
> Thank the reviewer for your positive feedback and valuable suggestions and questions. The following is our response to your questions.
>
>
> Q1: Numerical experiments.
>
> A1: We thank the reviewer for suggesting the consolidation of theory with simulations. We have included two simulation experiments in Appendix A. The first experiment takes the benign overfitting setup proposed in the paper, involving small shifts in the minor directions. It confirms the $\mathcal O(1/n)$ rate for ridge regression. Data is generated from a multivariate normal distribution, with target covariance matrices randomly generated. We validate the influence of two factors $\\|\mathcal T\\|$, $ \\mathrm{tr}[U]/ \\mathrm{tr}[V] $, identified in the paper as measures of covariate shifts in major and minor directions, respectively. The experiment results show that, for each combination of $\\|\\mathcal T\\|$ and $ \\mathrm{tr}[U]/\\mathrm{tr}[V] $, the excess risk of ridge regression decays at nearly 1/n. For a fixed sample size, the excess risk increases with larger values of $\\|\\mathcal T\\|$ or $ \\mathrm{tr}[U]/\\mathrm{tr}[V] $.
>
> The second experiment compares ridge regression with Principal Component Regression (PCR) under large shifts in the minor directions. The setup follows the instance in Theorem 4, where the excess risk of ridge regression is lower bounded by $\mathcal O(1/ \sqrt n)$, while PCR achieves an excess risk of $\mathcal O(1/n)$. This result is confirmed by the experiment, where the excess risk of PCR is compared against ridge regression with various regularization strengths: $\lambda = 0, n^{0.5}, n^{0.75}, n$. The findings show that the excess risk of ridge decays optimally at the rate of $n^{-0.48}$ for $\lambda = n^{0.75}$, consistent with the $\mathcal O(1/ \sqrt n)$ lower bound in Theorem 4. The optimal regularization in the experiment also aligns with that derived in the theoretical proof. In contrast, PCR achieves a superior decay rate of $n^{-0.99}$.
>
> Q2: Does the result in PCR require the number of relevant components k to be known in advance? What is the effect if the number is misspecified?
>
> A2: Our theory does not require k to be known in advance, because our upper bound holds for any $k$ as long as the assumptions are satisfied (Lemma 31 provides the generic result where we do not require $\\beta^{\\star}\_{-k} = 0$), though when applying PCR, a specific $k$ needs to be chosen. The effect of $k$ is discussed in Sec. 4.2, where we show that PCR works well when the eigengap $\lambda_k - \lambda_{k+1}$ is large and $\beta^{*}_{-k}$ is small.

---

> > ### Comment · Reviewer_R8Hj · 2024-11-28
> >
> > I thank the authors for addressing my questions. I think this is a very good paper and it should be accepted.

---

### Author Response · Authors · 2024-11-19
**Reply to reviewers**

Thank you all for your insightful comments and suggestions! We have revised our paper to incorporate several of your recommendations, with the changes highlighted in red. In response to the primary concern regarding the inclusion of numerical experiments to support our findings, we have added a simulation study, detailed in Appendix A.

---

### Meta-Review · Area_Chair_sKcV · 2024-12-20

**Metareview:**

(a) Summary of Scientific Claims and Findings
The paper provides a theoretical exploration of benign overfitting in the out-of-distribution (OOD) regime for over-parameterized linear models. Building on prior work limited to in-distribution settings, this submission demonstrates that benign overfitting persists under specific covariate shift scenarios. The authors derive non-asymptotic upper bounds for the excess risk in ridge regression and principal component regression (PCR), showing that ridge regression generalizes well under aligned source and target covariance. They highlight cases where ridge regression incurs high excess risk, emphasizing that PCR achieves faster statistical rates in such scenarios. The work generalizes existing results and offers insights into the conditions under which benign overfitting occurs OOD.

(b) Strengths

Novelty and Generalization: Extends the theoretical framework of benign overfitting to OOD settings, addressing a key gap in the literature.
Clarity and Rigor: The paper is well-written and systematically builds intuition alongside rigorous theoretical results.
Experimental Validation: The authors included numerical simulations that confirm their theoretical claims, enhancing the paper's credibility.
Connections to Existing Work: The results recover and generalize prior in-distribution findings and are contextualized within broader literature on ridge regression and PCR.
Practical Implications: The insights on when to use ridge regression versus PCR under covariate shift have real-world applicability.
(c) Weaknesses

Lower Bound Challenges: While the paper provides strong upper bounds, the lack of matching lower bounds (as seen in prior in-distribution results) is a limitation.
Assumption Clarity: Some reviewers noted that the assumptions for original contributions could be more prominently and clearly stated in the main text rather than the appendix.
Adaptiveness of PCR: The results for PCR rely on prior knowledge of the number of significant components, and an adaptive method would strengthen the contribution.
Generality of Numerical Results: While the simulations effectively validate the theoretical claims, more exploration of real-world datasets could further substantiate the findings.

(d) Decision Rationale
This paper addresses a significant theoretical challenge in machine learning by extending benign overfitting analyses to OOD scenarios. The rigorous theoretical contributions, supported by numerical experiments, provide strong evidence for the validity of the claims. While the absence of matching lower bounds and adaptiveness of PCR are noted limitations, they do not detract from the overall value of the paper. The contributions align with the interests of the ICLR theoretical community and offer a solid foundation for future work. Thus, I recommend acceptance as a spotlight presentation to highlight the importance of this topic.

**Additional Comments On Reviewer Discussion:**

The reviewers raised concerns about the lack of lower bounds and the adaptiveness of PCR. The authors addressed these issues by clarifying the technical challenges in proving lower bounds under relaxed assumptions and by proposing practical heuristics for selecting the number of components in PCR. Reviewers also suggested numerical experiments, which the authors provided, validating the theoretical results and enhancing the paper's empirical relevance. One reviewer expressed concerns about the assumptions being less explicit in the main text; the authors revised the paper to address this, further improving clarity.

Overall, the discussion strengthened the paper, with all reviewers maintaining or improving their positive evaluations. Each concern was adequately addressed, and the additional experiments and clarifications solidified the case for acceptance. I weighed the theoretical contributions and the robustness of the rebuttal responses heavily in my decision to recommend this paper.

---

### Decision · Program_Chairs · 2025-01-22

Accept (Poster)